



# Chemical characteristics of size resolved atmospheric aerosols in Iasi, north-eastern Romania. Nitrogen-containing inorganic compounds controlling aerosols chemistry in the area

Alina Giorgiana Galon-Negru[1], Romeo Iulian Olariu[1,2], Cecilia Arsene[1,2]

[1]"Alexandru Ioan Cuza" University of Iasi, Faculty of Chemistry, Department of Chemistry, 11 Carol I, 700506 Iasi, Romania

[2]"Alexandru Ioan Cuza" University of Iasi, Integrated Centre of Environmental Science Studies in the North Eastern Region, 11 Carol I, Iasi 700506, Romania

*Correspondence to*: Cecilia Arsene (carsene@uaic.ro); Phone number: +40-232-201354; Fax: +40-232-201313; Postal address: Cecilia Arsene, "Alexandru Ioan Cuza" University of Iasi, Faculty of Chemistry, Department of Chemistry, 11 Carol I, 700506 Iasi, Romania

**Abstract.** This study assesses the atmospheric aerosol load and behaviour (size and seasonal dependent) of the major inorganic and organic aerosol ionic components (i.e., acetate, ($C_2H_3O_2^-$), formate, ($HCO_2^-$), fluoride, ($F^-$), chloride, ($Cl^-$), nitrite, ($NO_2^-$), nitrate, ($NO_3^-$), phosphate, ($PO_4^{3-}$), sulfate, ($SO_4^{2-}$), oxalate, ($C_2O_4^{2-}$), sodium, ($Na^+$), potassium, ($K^+$), ammonium, ($NH_4^+$), magnesium, ($Mg^{2+}$) and calcium, ($Ca^{2+}$), in Iasi urban area, north-eastern Romania. Continuous measurements were carried out over 2016 by means of a cascade Dekati Low-Pressure Impactor (DLPI) performing aerosol size classification in 13 specific fractions evenly distributed over the 0.0276 up to 9.94 μm size range. Fine particulate $Cl^-$, $NO_3^-$, $NH_4^+$ and $K^+$ exhibited clear minima during the warm seasons and clear maxima over the cold seasons, mainly controlled by corroboration between factors such as enhancement in the emission sources, changes in the mixed layer depth and specific meteorological conditions. Fine particulate $SO_4^{2-}$ did not show much variation with respect to seasons. Particulate $NH_4^+$ and $NO_3^-$ ions were identified as critical parameters controlling aerosols chemistry in the area. The measured concentrations of particulate $NH_4^+$ and $NO_3^-$ in fine mode ($PM_{2.5}$) aerosols were found to be in reasonable good agreement with modelled values for winter but not for summer, an observation reflecting actually the susceptibility of $NH_4NO_3$ aerosols to be lost due to volatility over the warm seasons. Clear evidences have been obtained for the fact that in Iasi, north-eastern Romania, $NH_4^+$ in $PM_{2.5}$ is primarily associated with $SO_4^{2-}$ and $NO_3^-$ but not with $Cl^-$. However, indirect ISORROPIA-II estimations showed that the atmosphere in the investigated area might be ammonia-rich during both the cold and warm seasons, such as enough $NH_3$ to be present to neutralize $H_2SO_4$, $HNO_3$ and $HCl$ acidic components and to generate fine particulate ammonium salts, in the form of $(NH_4)_2SO_4$, $NH_4NO_3$ and $NH_4Cl$. ISORROPIA-II runs allowed us estimating that over the warm seasons ~ 35 % of the total analyzed samples presented pH values in the very strong acidity fraction (0–3



pH units range) while over the cold seasons the contribution in this pH range was of ~ 43 %. Moreover, while over the warm seasons ~ 24–25 % of the acidic samples were in the 1–2 pH range, reflecting mainly contributions from very strong inorganic acids, over the cold seasons an increase to ~ 40 %, brought by the 1–3 pH range, would reflect possible contributions from other acidic type species (i.e., organics), changes in aerosols acidity impacting most probably the gas–particle partitioning of semi-volatile organic acids. In overall, it has been estimated that within the aerosol mass concentration the ionic mass brings contribution as high as 40.6 % with the rest being unaccounted yet.

**Keywords:** urban aerosols, ionic chemical composition, ammonium, nitrate, pH, long range transport, Iasi, Romania





# 1 Introduction

Despite dramatic progress made to improve air quality, at global level air pollution continues to harm people's health and the environment. In Europe and at worldwide level, the aerosols or particulate matter (PM) problem is still of a great concern of interest (Olariu et al., 2015) as Europe and many other areas in the world (e.g., China, India, USA) are some of the most
important sources for anthropogenic aerosols.

Atmospheric aerosols, described as complex mixtures of liquid and/or solid particles suspended in a gas (Olariu et al., 2015), are mainly originating from anthropogenic and natural sources (Querol et al., 2004). Fine $PM_{2.5}$ particles (airborne particles with an equivalent aerodynamic diameter < 2.5 μm) are assigned as air pollutants with significant effects on human health (Pope et al., 2004; Dominici et al., 2006; WHO, 2006a; Directive 2008/50/EC, 2008; Sicard et al., 2011; Ostro et al., 2014),
air quality (Directive 2008/50/EC, 2008; Freney et al., 2014), visibility (Tsai and Cheng, 1999; Directive 2008/50/EC, 2008), ecosystems, weather and climate (Ramanathan şi colab., 2001; Directive 2008/50/EC, 2008; IPCC, 2013) states. However, aerosols are also known to play a significant role within the chemistry of the atmosphere (Prinn, 2003), by providing appropriate surfaces for heterogeneous chemical reactions to occur (Ravishankara, 1997).

Safety threshold values for both the $PM_{2.5}$ (10 or 17 μg m$^{-3}$ air, as annual mean) and the $PM_{10}$ (airborne particles with an
equivalent aerodynamic diameter < 10 μm; 20 or 28 μg m$^{-3}$ air, as annual mean) fractions are addressed by the WHO (2006b) or by the 2008/50/EC Directive of the European Parliament and the Council of 21 May 2008 on ambient air quality and cleaner Europe (Directive 2008/50/EC, 2008). With regard to the $PM_{2.5}$ fraction the EEA Report 5 (2015) clearly shows that, for instance, in 2013 the EU daily limit value for $PM_{10}$ was exceeded in 22 out of the 28 EU member states while for the $PM_{2.5}$ the target value was exceeded in 7 states (a decreasing trend is observed when compared with the data in the WHO
Report (2006b)). Moreover, it has been recently shown that on a global scale exposure to fine $PM_{2.5}$ leads to about 3.3 million premature deaths per year worldwide (i.e., predominantly in Asia), with an estimate susceptible to double by 2050 (Lelieved et al., 2015). It was also clearly shown that PM air pollution imparts a tremendous burden to the global public health, ranking it as the 13[th] leading cause of mortality (Brook, 2008).

Up to now chemical composition of atmospheric aerosols is reported for various urban European sites (Bardouki et al., 2003;
Hitzenberger et al., 2006; Tursic et al., 2006; Gerasopoulos et al., 2007; Schwarz et al., 2012; Laongsri and Harrison, 2013; Wonaschutz et al., 2015; Sandrini et al., 2016) but at eastern European sites such information is very scarce (Arsene et al., 2011). Sulfate ($SO_4^{2-}$), nitrate ($NO_3^-$) and ammonium ($NH_4^+$) ions are often assigned as significantly contributing inorganic species to the total aerosol mass (Wang et al., 2005; Bressi et al., 2013; Hasheminassab et al., 2014; Voutsa et al., 2014). Ammonium aerosols, with atmospheric lifetime of 1–15 days and clear tendency to deposit at larger distances from their
emission sources (Aneja et al., 2000), seem to play a very important role in the atmospheric chemistry. In urban air, fine particulate $NO_3^-$ abundance seems to be mainly controlled by the reaction between $HNO_3$ and $NH_3$ (Stockwell et al., 2000), while at a global scale $HNO_3$ heterogeneous induced reaction on the mineral dust and sea salt particles surface might be the predominant source for particulate $NO_3^-$ (Athanasopoulou et al., 2008; Karydis et al., 2011). In the atmosphere, through





particulate $NH_4NO_3$, ammonia currently contributes to effects on human health since there are evidences that this has become gradually more important relative to emissions of oxidized nitrogen (Sutton et al., 2011).

Relative humidity (RH) related reactions (Jang et al., 2002) and gas–particle partitioning processes of semi-volatile species (i.e., ammonia ($NH_3$), nitric ($HNO_3$), hydrochloric ($HCl$) and some organic acids) highly controlled by hydronium ion

($H_3O^+$) concentrations (Keene et al., 2004; Guo et al., 2016) are responsible on aerosol formation and their chemical composition. Ion balance method (Trebs et al., 2005; Metzger et al., 2006; Arsene et al., 2011), phase partitioning (Meskhidze et al., 2003; Keene et al., 2004) and thermodynamic equilibrium models (e.g., E-AIM, (Clegg et al., 1998; Wexler and Clegg, 2002) or ISORROPIA-II (Nenes et al., 1999; Fountoukis and Nenes, 2007; Hennigan et al., 2015; Guo et al., 2015, 2016; Fang et al., 2017)) often are used to estimate aerosols acidity. Particles acidity might influence transition

metals solubility and enhance aerosols toxicity and atmospheric nutrient delivered through atmospheric deposition in marine areas (Meskhidze et al., 2003; Fang et al., 2017).

Apart the scientific progress made in aerosols chemical composition characterization, the aerosols role in the global atmospheric system is not yet sufficiently understood. Aerosols treatment is particularly still very challenging mainly due to the existence of multiple sources (e.g., soil erosion, sea spray, biogenic emission, volcano eruptions, soot from combustion,

condensation of precursor gases, and other anthropogenic sources) or due to the complexity of interactions with other atmospheric constituents (Zhang et al., 2015). Nowadays, sources, distribution and behaviour of natural and anthropogenic aerosols are still subjected to much scientific debates even in research groups with a relatively good understanding of the interest matters. However, it is still agreed on the scarcity of aerosols related work for eastern EU countries (EEA Report 5, 2015) but also on the existent discrepancies between theoretically predicted values and those generated from real

measurements. The uncertainties are strongly related to secondary inorganic aerosols mainly controlled by the availability of atmospheric sulfuric acid ($H_2SO_4$), nitric acid ($HNO_3$) and ammonia ($NH_3$) (Ianniello et al., 2011). Lack of sufficient detailed observations and emissions data bring high uncertainty in the estimated radiative forcing but it is clearly stated that warming of the climate system is unequivocal, and also that since the 1950s, many of the observed changes are unprecedented (IPCC, 2013).

Despite a growing recognition at international level on the importance related to the air pollution and air quality problems, assessment of the air pollution patterns in Romania remains very scarce. For the north-eastern Romania, there is only limited number of publications mainly concerning the chemical characteristics of the ambient air pollutants (i.e., water soluble ionic constituents of aerosols (Arsene et al., 2011) and rainwater (Arsene et al., 2007)). Recent work performed by Arsene's group in the field of atmospheric chemistry clearly show that in Iasi urban environment (north-eastern Romania), exploring

aerosols chemical composition and the chemical processes taking place still involves many unknowns and many challenges (e.g., ~ 59 % of the total aerosol mass concentration unaccounted yet). The present work reports for the first time detailed information concerning the chemical composition and seasonal variation of size segregated water soluble ions in aerosol samples collected for a year in 2016 in Iasi urban area (north-eastern Romania). Consideration is given also on the potential



ongoing chemistry in the area toward the potential contributions brought by critical driving forces as meteorological factors (i.e., relative humidity, temperature), mixed layer depth and emission sources intensity.

## 2 Experimental

### 2.1 Measurement site

Measurements were performed in Iasi, north-eastern Romania, at the Air Quality Monitoring Station (AMOS, 47°9' N latitude and 27°35' E longitude) of the Integrated Centre of Environmental Science Studies in the North Eastern Region, "Alexandru Ioan Cuza" University of Iasi, CERNESIM-UAIC, Romania. AMOS is located north-east from the city centre, on the rooftop of the highest University building (~ 35 m above the ground level). As AMOS is located in a totally open area the sampling site is assigned as an urban receptor point most probably influenced by well-mixed air masses. A

comprehensive demo-geographical characterization of Iasi is described in detail by Arsene et al. (2007, 2011). However, according to a more recent estimate of the Romanian National Institute of Statistics, in 2016 the population in Iasi reached about 362,142 inhabitants (Ichim et al., 2016).

### 2.2 Field measurements

Size resolved atmospheric aerosols were collected on 25 mm diameter ungreased aluminum filters using a cascade Dekati

Low-Pressure Impactor (DLPI) operating at a flow rate of 29.85 L min$^{-1}$. Similar devices have been successfully used also in other studies (Kocak et al., 2007; Wonaschutz et al., 2015). The DLPI unit performs aerosol size classification in the 0.0276 up to 9.94 μm range using 13 specific fractions evenly distributed (with 0.0276, 0.0556, 0.0945, 0.155, 0.260, 0.381, 0.612, 0.946, 1.60, 2.39, 3.99, 6.58, 9.94 μm at 50 % calibrated aerodynamic cut-point diameters). Sampling performance of the DLPI unit was verified through comparison with simultaneous measurements performed with the stacked filter unit (SFU)

system previously used by Arsene et al. (2011). During sampling neither denuders nor backup-filters were used which would mean that sampling artifacts of semi-volatile species (e.g., $NH_4NO_3$ and $NH_4Cl$) cannot be completely excluded. Before each reuse the sampler's components were cleaned with ultra pure water and methanol. Dispensable polyethylene gloves were always used to avoid hand contact with sampling components. The sampler's components were assembled and dissembled in a Labguard Class II Safety Biological Cabinet, NuAire, disposed in one of CERNESIM's location. The DLPI sampler was

transported to and from the field in tightened polyethylene bags.

Sampling was performed over the entire 2016 time period, twice a week, on weekend and working days. A total of 84 sampling events (41 during the cold seasons covering October to March time-interval, and 43 during the warm seasons covering April to September time-interval), generating 1092 size resolved aerosol samples, were collected and analyzed over the entire period. Sampling was carried out on a 36 h basis, with each sampling event starting at 18:00 local time. A volume

of 64.33 ± 0.85 m³ (given as an average of the 84 sampling events) was collected per each 36-hour based event. At least two field blanks (consisting of loaded sampler taken to and from the field but never removed from its tightened polyethylene



bag) were generated and simultaneously analyzed with the laboratory blanks in order to determine either or not contamination occurred during the sampler loading, transport, or analysis.

Meteorological parameters including atmospheric temperature (AT), relative humidity (RH), wind speed (WS), wind direction and global radiation were accessed from the data base available for the Hawk GSM-240 CERNESIM's weather

station, running at the AMOS site. Information about the mixed layer depth (the atmospheric boundary layer) and its ability to dilute atmospheric pollutants at the investigated site was obtained from the NOAA Air Resources Laboratory (ARL) website (Stein et al., 2015; Rolph et al., 2017).

## 2.3 Sample analyses

Aerosols mass for both the $PM_{2.5}$ and $PM_{10}$ fractions was gravimetrically determined using a Sartorius microbalance

(MSU2.7S-000-DF, ± 0.2 µg sensitivity) by weighing aluminium filters before and after sampling. Prior undertaking the weighing procedure, filters stored in Petri dishes were kept for at least 3 days in a conditioned room at a relative humidity (RH) of 40 ± 2 %, and a temperature of 20 ± 2 °C. In between different procedure, Petri dishes containing unloaded and loaded filters were appropriately stored in zipped plastic bags.

Chemical analyses for 9 water soluble anions (i.e., acetate, $(C_2H_3O_2^-)$, formate, $(HCO_2^-)$, fluoride, $(F^-)$, chloride, $(Cl^-)$,

nitrite, $(NO_2^-)$, nitrate, $(NO_3^-)$, phosphate, $(PO_4^{3-})$, sulfate, $(SO_4^{2-})$ and oxalate, $(C_2O_4^{2-})$) and 5 water soluble cations (i.e., sodium, $(Na^+)$, potassium, $(K^+)$, ammonium, $(NH_4^+)$, magnesium, $(Mg^{2+})$ and calcium $(Ca^{2+})$) were performed by ion chromatography (IC) on a ICS-5000$^+$ DP DIONEX ion chromatographic system (Thermo Scientific, SUA). Analyses were performed within a week after sampling, in aqueous extracts of the collected samples. For chemical analysis, steps in the analytical procedures were strictly quality-controlled to avoid any contamination of the samples.

After sampling and all other required preparative steps, one half of each collected filter was ultrasonically extracted for 45 minutes in 5 mL of deionized water (resistivity of 18.2 MΩ.cm) produced by a Milli-Q Advantage A10 system (Millipore). Filtered extracts (0.2 µm pore size cellulose acetate filters, ADVANTEC) were analyzed on an IonPac CS12A (4×250 mm) analytical column for cations and on an IonPac AS22 (4×250 mm) column for anions, running simultaneously on the DIONEX ICS-5000$^+$ DP system. The chromatographic instrumental setup was completed by CSRS 300×4 mm and AERS

500×4 mm electrochemical suppressors and conductivity detectors. Ions analyses were performed under isocratic elution mode, using $CO_3^{2-}/HCO_3^-$ (4.5/1.4 mM, 1.2 mL min$^{-1}$) as mobile phase for anions and methane sulfonic acid (20 mM, 1.0 mL min$^{-1}$) as mobile phase for cations.

Dionex Seven Anions II and Dionex Six Cations II traceable standard solutions were used to generate calibration curves for each species of interest. Quantitative estimates of the analyzed ions were obtained by interpolation on the constructed linear

calibration curves that all had correlation coefficients $R^2$ well above 0.995. For major analyzed cations and anions, the detection limits (defined as 3 times the standard deviation of blank measurements relative to the methods sensitivity) on a 36 h measurement period were of 0.0003 µg m$^{-3}$ for $NH_4^+$ (3.2 µg L$^{-1}$), 0.001 µg m$^{-3}$ for $Na^+$ (13.4 µg L$^{-1}$), 0.0015 µg m$^{-3}$ for $K^+$ (19.8 µg L$^{-1}$), 0.0002 µg m$^{-3}$ for $Mg^{2+}$ (3.2 µg L$^{-1}$), 0.0016 µg m$^{-3}$ for $Ca^{2+}$ (20.6 µg L$^{-1}$), 0.0010 µg m$^{-3}$ for $SO_4^{2-}$ (13.2



µg L$^{-1}$), 0.0012 µg m$^{-3}$ for NO$_3^-$ (15.1 µg L$^{-1}$) and 0.0019 µg m$^{-3}$ for Cl$^-$ (24.9 µg L$^{-1}$). Ions concentrations in analyzed blank filters (laboratory and field) were subtracted from their corresponding concentrations in the analyzed aerosol samples. The sum (of detected ions or of the gravimetrically determined mass concentration) over all DLPI stages is termed "PM$_{10}$ fraction" while the sum over impactor stages from 1 to 10 are termed "PM$_{2.5}$ fraction". Modal diameters of the size segregated aerosols particles or of the analyzed ionic components (individual or as a sum) were determined by fitting lognormal distributions.

## 2.4 Estimation of the aerosols acidity

The thermodynamic model proposed by Fountoukis and Nenes (2007), i.e. ISORROPIA-II (http://isorropia.eas.gatech.edu/), was used to get an estimate of the in situ potential acidity in the PM$_{2.5}$ fraction in Iasi, north-eastern Romania. ISORROPIA-II thermodynamic equilibrium model calculates the gas/liquid/solid equilibrium partitioning of the K$^+$, Ca$^{2+}$, Mg$^{2+}$, NH$_4^+$, Na$^+$, SO$_4^{2-}$, NO$_3^-$, Cl$^-$, aerosol water content and can predict particles pH. Up to now the model has been actually used for various field campaigns data analysis (Nowak et al., 2006; Fountoukis et al., 2009).

To obtain the best predictions of aerosols pH, ISORROPIA-II model was run in the "forward mode" for metastable aerosol state. Preliminary runs of the experimental data in the "reverse mode" did not supplied suitable information. In the "metastable" mode the aerosol is assumed to be present only in the aqueous phase, either supersaturated or not (Fountoukis and Nenes, 2007). As input data into the model, we have used just aerosol-phase ions concentrations measured by the IC, along with relative humidity and temperature data from the Hawk GSM-240 weather station. However, there is suggestion that without accompanying gas-phase data required to constrain the thermodynamic models, a more accurate representation of aerosol pH can be obtained by using the aerosol concentrations as input in "forward mode" calculations (Guo et al., 2015; Hennigan et al., 2015). Under these conditions the model seems to be less sensitive to measurement error than the "reverse mode". Where required, predicted NH$_3$ data (as output of ISORROPIA-II thermodynamic equilibrium model) have been used for the interpretation of the results reported in the present work.

## 2.5 Air mass back trajectories and air mass origin

Air mass back trajectories were calculated using the HYSPLIT 4 model of the NOAA Air Resources Laboratory (Stein et al., 2015; Rolph et al., 2017). 48 h back trajectories, arriving at the investigated site at 18:00 local time (15:00 UTC), were computed at 500, 1000 and 2000 m altitudes above the ground level. Four major sectors of air masses origin were distinguished and their contributions are shown in Fig. 1. The north-eastern (N-E) sector (36.6 %) seems to be the most prevalent while the south-south-eastern (S-SE) sector is the least frequent (11.4 %). The W-SW sector is mainly prevailing during winter, while the N-E sector is most common during summer. North-westerly type sector seems to present a slightly enhanced frequency in winter and summer and, according to James (2007), this would reflect a possible European monsoon circulation. The events from the S-SE sector, prevailing mainly in spring, carried out marine chemical features highly influenced by the Black Sea. Air masses travelling above large (long range transport) or short (local) continental areas,



undertaking faster vertical transport most probably due to the locally/continentally driven buoyancy (Holton, 1979; Seinfeld and Pandis, 1998), were also identified (e.g., in April 2016, 5 sampled events out of the total 8 were highly influenced by fast vertical air masses transport). Quite often, for these events, the air masses from both 500 and 1000 m altitude ceased down in their elevation to below 500 meters (air masses brushing the ground surface) with strong impact on the chemical composition

of the collected particles (see also discussion later in the text).

## 3 Results and discussion

### 3.1 Variability of $PM_{10}$ and $PM_{2.5}$ mass concentrations

Table 1 shows summary statistics (median, geometric mean, arithmetic mean, standard deviation, minimum and maximum) for $PM_{10}$ and $PM_{2.5}$ fractions mass concentrations (gravimetrically determined) at the AMOS site for both working days and

weekends. Analyses by statistical tests were applied to determine either or not the differences among the mean of working days (more intense local anthropogenic activities) and weekends are significant. Shapiro-Wilk normality test, applied for both $PM_{2.5}$ (p = 0.546, at the 95 % confidence level) and $PM_{10}$ (p = 0.682, at the 95 % confidence level) fractions indicated that the entire data-base sets were normally distributed. Since the difference in the mean values of the two groups was not great enough to reject the possibility that the difference is due to random sampling variability it was assumed that there is not

a statistically significant difference between the input groups (p = 0.987, at the 95 % confidence level for $PM_{2.5}$ fraction; p = 0.998, at the 95 % confidence level for $PM_{10}$ fraction). Since not statistically significant differences were observed, the result allowed us suggesting that AMOS site is mainly influenced by long range transport rather than local contributions. Moreover, it seems that at this site, the $PM_{10}$ annual mean mass concentration of 18.95 µg m$^{-3}$ do not exceeds the 20 µg m$^{-3}$ air annual mean WHO set value, while for the $PM_{2.5}$ fraction the annual mean mass concentration of 16.92 µg m$^{-3}$ exceeds

the 10 µg m$^{-3}$ air annual mean WHO set value (WHO, 2006b).

Table 2 presents the annual and/or seasonal arithmetic means of the $PM_{10}$ and $PM_{2.5}$ fraction mass concentrations in Iasi, north-eastern Romania, and other various European sites (mean ± stdev). As seen in Table 2 the annual averages obtained in the present work (i.e., 16.9 ± 9.1 for $PM_{2.5}$ and 18.9 ± 9.3 µg m$^{-3}$ for $PM_{10}$) are slightly different from those reported by Arsene et al. (2011) (i.e., 10.5 ± 11.5 µg m$^{-3}$ for $PM_{1.5}$ and 38.3 ± 25.4 µg m$^{-3}$ for $PM_{>1.5}$). Possible explanations for the

differences between afore mentioned studies might be seen as results either of different sampling altitudes (35 m in the present work vs. 25 m in Arsene et al. (2011)) or due to an enhanced sampling efficiency of the DLPI unit with regard to fine and ultrafine particles (DLPI unit in the present work vs. SFU unit in Arsene et al. (2011)). Parallel DLPI and SFU sampling runs (performed within January–July 2016) showed that the DLPI unit could collect in average with ~ 6.7 µg m$^{-3}$ more particles than the SFU system. However, in a study from 2016, Alastuey et al. (2016) report for $PM_{10}$ concentrations in

Moldova (the closest point to our sampling site), values as high as ~ 25 µg m$^{-3}$ over summer and of ~ 25–30 µg m$^{-3}$ over winter period, yielding an annual averaged value of ~ 27.5 µg m$^{-3}$ which is much higher than the 18.9 µg m$^{-3}$ value reported in the present work, for Iasi, north-eastern Romania. However, the elevated concentrations observed at the eastern sites were



attributed to regional or local sources (Alastuey et al., 2016). The difference observed in comparison with other European sites might actually reflect that altitude would be an important controlling factor to the atmospheric aerosol burden at a site.

In Iasi, north-eastern Romania, mass concentrations in the 5 to 10 µg m$^{-3}$ range, for the PM$_{10}$ and PM$_{2.5}$ fractions were especially observed in samples collected after raining events (i.e., May, June, August, and October). Such behaviour would

be expected since particles from the atmosphere might be efficiently removed by precipitation (Arsene et al., 2011). However, during events with strong natural or anthropogenic contributions the PM$_{10}$ and PM$_{2.5}$ mass concentrations exceeded the averages observed at AMOS. For example, an event collected in April 2016, from 9[th] to 11[th], with 34.3 µg m$^{-3}$ in PM$_{2.5}$ and 43.9 µg m$^{-3}$ in PM$_{10}$, was actually highly influenced by the long range transport phenomena of African dust and marine aerosols from the Black and Aegean/Mediterranean seas (i.e., event described in detail later in the text). For other

events, the exceeding fine fraction mass concentrations are believed to be a result of more variable sources (combustion, biogenic, local mineral dust, meteorological factors, etc.).

Figure 2 shows monthly arithmetic mean mass concentrations of the PM$_{10}$ and PM$_{2.5}$ fractions (accompanied by standard deviations) in Iasi, north-eastern Romania. The distribution of the PM$_{2.5}$/PM$_{10}$ ratio is presented in the same figure. The relative contribution of the PM$_{2.5}$ toward the PM$_{10}$ particles indicates low variability amongst the months of the year with

ratios ranging from ~ 0.75 to ~ 1.0. For urban background and/or traffic sites PM$_{2.5}$/PM$_{10}$ ratios in the 0.42 to 0.82 range are given by the WHO (WHO, 2006a). Clear seasonal pattern with maxima during the cold seasons and minima during the warm seasons are attributable to the PM$_{2.5}$, PM$_{10}$ and PM$_{2.5}$/PM$_{10}$ profiles. The observed seasonal pattern might be the result of combined effects from emissions seasonal variations, local and long range air masses transport and dispersion, chemical production and loss, and deposition (Wang et al., 2016). The higher PM$_{2.5}$/PM$_{10}$ ratios, occurring especially during cold

seasons, are most probably due to contributions from combustion processes. During such periods, i.e. coal/petroleum burning for heating purposes, an enhancement of secondary aerosols generation (sulfate and organic compounds) is expected (Li et al., 2012). The lower PM$_{2.5}$/PM$_{10}$ ratios occurring during warm seasons can be due to the increased frequency of dust and also due to more intense anthropogenic activities in the neighbourhood of the sampling site (excavation, construction, build renewing and not only) which would result in higher loading of coarse particles in the atmosphere. According to Zhang et al.

(2001) it might also be that during the warm seasons the growth of plants may enhance the dry deposition of PM$_{2.5}$ and hence the PM$_{2.5}$/PM$_{10}$ ratio.

However, over the investigated period clear seasonal pattern was observed for the gravimetrically determined mass concentration size distribution (Fig. 3). The average size distribution over the cold seasons presents a clear monomodal feature with maximum at 381 nm. The warm seasons are characterized by the presence of a dominant fine mode, with

maxima at 381 nm, but also by the occurrence between 1.60–2.39 µm of a supermicrone mode. Most probably, the observed seasonal variations cannot be attributed solely to changes in sources contributions, but also probably to variations of meteorological conditions.



### 3.2 Ionic balance, seasonality of water soluble ions and stoichiometry of $(NH_4)_2SO_4$ and $NH_4NO_3$

#### 3.2.1 Ionic balance and potential aerosols acidity

Water soluble inorganic anions (i.e., $F^-$, $Cl^-$, $NO_2^-$, $NO_3^-$, $PO_4^{3-}$, $SO_4^{2-}$), organic anions (i.e., $HCO_2^-$, $C_2H_3O_2^-$, $C_2O_4^{2-}$), and inorganic cations (i.e., $Na^+$, $K^+$, $NH_4^+$, $Mg^{2+}$, $Ca^{2+}$) were analyzed by ion chromatography. The completeness degree of the

ionic balance for the identified and quantified species in both the $PM_{10}$ and $PM_{2.5}$ fractions was checked. For raw ion chromatography data slopes lower than unity in the $\sum_{cations}$ vs. $\sum_{anions}$ dependences were observed in both the $PM_{2.5}$ (total period, averaged ratio of 0.69, Pearson coefficient of 0.94, p < 0.001, at the 99.9 % confidence level; cold seasons, ratio of 0.67, Pearson coefficient of 0.98, p < 0.001, at the 99.9 % confidence level; warm seasons, ratio of 0.84, Pearson coefficient of 0.87, p < 0.001, at the 99.9 % confidence level) and $PM_{10}$ (total period, averaged ratio of 0.70, Pearson coefficient of 0.94,

p < 0.001, at the 99.9 % confidence level; cold seasons, ratio of 0.68, Pearson coefficient of 0.98, p < 0.001, at the 99.9 % confidence level; warm seasons, ratio of 0.86, Pearson coefficient of 0.86, p < 0.001, at the 99.9 % confidence level) and these indicate important cation deficit in the ionic balance. It seems that over cold seasons the cation deficit is higher than that over warm seasons and it should be also noted that at the investigated site the RH values went even to ~ 82 % during cold seasons. However, per sampled events either cation or anion deficit has been observed in various stages.

Since ion balance method, thermodynamic equilibrium models, molar ratio, and phase partitioning of certain semi-volatile compounds (e.g., $NH_3/NH_4^+$, $HNO_3/NO_3^-$, $HCl/Cl^-$), are methods commonly employed for proton loading in atmospheric particles estimation, the generated data-base was under scrutiny in these regards. Predicting pH is suggested as the best method to analyze also particle acidity (Guo et al., 2015). Usually, the ion balance method is based upon the principle of electroneutrality. Any deficit in measured cationic charge compared to measured anionic charge is usually assumed to be due

to the presence of unmeasured protons ($H^+$) while the reverse is available for unmeasured hydroxyl ($OH^-$) (Hennigan et al., 2015) or bicarbonate/carbonate ($HCO_3^-/CO_3^{2-}$) (Fountoukis and Nenes, 2007).

In the present work, however, $HCO_3^-/CO_3^{2-}$ was assigned as the missing anion while $NH_4^+$ as the major missing cation. The missing $HCO_3^-/CO_3^{2-}$ has been estimated as suggested by Arsene et al. (2007) while the rationale previously proposed by Arsene et al. (2011) has been used to estimate the missing $NH_4^+$. Within the $NH_4^+$(total) fraction (defined as the sum between

that derived from raw IC data and the part estimated by using the rationale of Arsene et al. (2011)), the correction for the missing $NH_4^+$ accounted for about 21.65 ± 25.70 % for the warm seasons while for the cold seasons the correction accounted for about 46.05 ± 18.43 %. However, when estimated missing $NH_4^+$ has been taken into account a significant improvement in the overall ionic balance ($\sum_{cations}$ vs. $\sum_{anions)}$ was observed for both the $PM_{2.5}$ (total period, ratio of 0.95, Pearson coefficient of 0.98, p < 0.001, at the 99.9 % confidence level; cold seasons, ratio of 0.97, Pearson coefficient of 0.99, p < 0.001, at the

99.9 % confidence level; warm seasons, ratio of ~ 0.87, Pearson coefficient of 0.96, p < 0.001, at the 99.9 % confidence level) and the $PM_{10}$ (total period, ratio of 0.95, Pearson coefficient of 0.98, p < 0.001, at the 99.9 % confidence level; cold seasons, ratio of 0.98, Pearson coefficient of 0.99, p < 0.001, at the 99.9 % confidence level; warm seasons, ratio of 0.86, Pearson coefficient of 0.96, p < 0.001, at the 99.9 % confidence level) fractions.



Since in the literature there are suggestions that in the ion balance any deficit in inorganic cations relative to anions is mainly due to the presence of H$^+$ (Hennigan et al., 2015), in an attempt to investigate whether or not the H$^+$ species would bring an important contribution within the ionic balance, ISORROPIA-II thermodynamic equilibrium model proposed by Fountoukis and Nenes (2007) has been used in the present work. The model helps deriving (even under constrains) either information

related to potential H$^+$ concentration in the aerosol particles (implicitly also on aerosols pH) or to the liquid water content (if a liquid phase exists), and computed concentrations of gas–phase semi-volatile compounds in equilibrium with the aerosol (e.g., $NH_3$, $HNO_3$, and $HCl$) (Hennigan et al., 2015). For the present data-base, ISORROPIA-II model was run in "forward mode" for metastable aerosol. Runs were performed both for $NH_4^+$ derived from raw IC data and also for the situation when missing ionic species have been indirectly estimated and taken into account ($NH_4^+$(total) and $HCO_3^-$/$CO_3^{2-}$ estimated as

previously presented). Although use of ISORROPIA-II model runs in forward mode with only aerosol-phase input is highly susceptible for debate, Guo et al. (2015) report also about use of the model under these specific conditions. It seems that under these circumstances the model is less sensitive to measurement error than the reverse mode. However, in Guo's et al. (2015) work, pHs reported for SCAPE were corrected for the identified bias and the values were increased by 1 to simplify the correction.

Although there is suggestion that in thermodynamic model calculation (a region where the thermodynamic predictions, and assumption of equilibrium, become more accurate) only data with RH exceeding 60 % might be considered (Moya et al., 2002), in the present work ISORROPIA-II was applied over the entire set of data-base regardless RH values (in 2016, at the investigated site, July, August and September were the months with RH < 40 %). However, the model was also run at additional 5 to 10 % RH (to that experimentally recorded) in order to check RH potential influence on the envisaged

estimations. It has been observed that a change in the RH value with 5 or 10 % would result actually in a pH change with about 2 and, respectively, 3 %. Since in the present work, organic species have been taken into account in the ionic balance it is believed that the bias in the estimated species (especially $NH_4^+$ and $HCO_3^-$/$CO_3^{2-}$) is minimized and implicitly that in inferred H$^+$ by ISORROPIA-II thermodynamic model.

For Iasi, north-eastern Romania, when ISORROPIA-II estimated H$^+$ has been taken into account the improvement in the

overall ionic balance (in comparison with that derived by considering the $NH_4^+$(total) fraction) was almost insignificant with an ~ 2 % increase on the $\sum_{cations}$ vs. $\sum_{anions}$ ratio, observed in both the PM$_{2.5}$ and PM$_{10}$ fractions. However, over the warm seasons it seems that neither $NH_4^+$(total) fraction option nor the one taking into account ISORROPIA-II estimated H$^+$ concentrations do not lead to a close to 1 ionic balance, with ratios varying between 0.84 and 0.88 for the PM$_{2.5}$ fraction and 0.84 to 0.87 for the PM$_{10}$ fraction. Such observation would not suggest high uncertainty in the accuracy of the ionic species

measured values but most probably that some volatile cation species (e.g., amines) were not measured.

However, the relationship between ISORROPIA-II predicted aerosol pH and the ionic balance has been investigated for the present data-base and the results are presented in Fig. 4a. Details in this figure (following features of a traditional titration curve) clearly show that many of the analyzed particles were neutral (dashed lines at 0). A very important fraction of the investigated samples were in the acidic range (particles pH below 3 if aerosol samples were in cation deficit mode) while the



remaining fraction was in the alkaline range (particles pH slightly above 7 if aerosol samples were in anion deficit mode). However, at neutrality it is observed that, indeed as suggested by Hennigan et al. (2015), small uncertainties in the ionic balance (coming mainly from the uncertainty in the measurements) may lead to shifts in pH that span over about 10 pH units. Moreover, the sensitivity in predicted aerosol pH under forward-mode calculation to changes in the aerosol $NH_4^+$

concentration has been checked and, as shown in Fig. 4b, it seems that the predicted pH might decrease by 2 % when the ionic balance takes into account $NH_4^+$(total) concentration. Actually, for about 19 % of the data the aerosol pH differed by less than 1.0 pH unit when $NH_4^+$(total) concentrations increased in comparison with $NH_4^+$ derived from raw IC data (a $(35.65 \pm 24.92)$ % increase in $NH_4^+$ concentration for the entire data-base, a $(21.65 \pm 25.7)$ % increase for the warm seasons and a $(46.05 \pm 18.43)$% (mean $\pm$ stdev) increase for the cold seasons). Such change in $NH_4^+$ concentrations is not unexpected

since this is assigned as a highly volatile compound, with high susceptibility to be lost by evaporation. As it will be shown later in the text, meteorological parameters (i.e., RH) could play an important role within $NH_4^+$ distribution in both during the warm and the cold seasons. In other studies, higher evaporation rates are reported for summer when compared to winter, but during winter low RH values were prevailing (Ianniello et al., 2011; Zhao et al., 2016). Ianniello et al. (2011) report for example that in Beijing, China, about 35 % of fine particulate $NH_4^+$ was susceptible to evaporate from the sampling filters

during winter while about 53 % of fine particulate $NH_4^+$ was susceptible to evaporate during summer. For the same location, Zhao et al. (2016) report $NH_4^+$ loss of 6 % and higher losses of $NO_3^-$ and $Cl^-$.

In Iasi, north-eastern Romania, an important fraction of the total analyzed samples was alkaline while the remaining part was acidic. A more detailed view of the samples pH distribution can be easily cached from the data presented in Fig. 4c-d. It seems that over the warm seasons about 55–56 % of the analyzed samples were alkaline (pH > 7) and about 44–45 % were

acidic (pH < 7), with the last fraction mainly distributed in the very strong acidity fraction (~ 35 % of the samples with pH in the 0–3 range and about 2 % with aerosol pH less than 0). Over the cold seasons only 47 % of the total analyzed samples were alkaline (pH > 7) and 53 % were acidic (pH < 7). The acidity was also mainly distributed in the very strong acidity fraction (~ 43 % of the acidic samples with pH in the 0–3 range). However, while during the warm seasons ~ 24–25 % of the acidic samples has had the pH in the 1-2 range (reflecting actually contributions from very strong inorganic acids), over the

cold seasons an increase to ~ 40 %, brought by the 1–3 pH range, would reflect possible contributions from other acidic type species (i.e., organics). Aerosols acidity might actually impact the gas–particle partitioning of semi-volatile organic acids. While under strong acidic conditions (pH in the 1–3 range) organic acids contributions to the $H^+$ (hence pH) are expected to be negligible (since low pH prevents dissociation of organic acids), at pHs above 3, and well approaching 7, it is believed that their contribution would significantly enhance (see also discussion later in the text). Under these circumstances, in Iasi,

north-eastern Romania, the contribution from formic acid, one of the strongest organic acid which has a $pK_a$ of 3.75 (Bacarella et al., 1955) ($pK_a = - \log K_a$, with $K_a$ referring to acid dissociation constant) might become important over the cold seasons.

Figures 5a,b present the size distribution of averaged aerosol mass, $NO_3^-$, $SO_4^{2-}$ and $NH_4^+$ concentrations while Fig. 5c,d present ISORROPIA-II estimated pH and $H^+$ mass concentration distributions for both the cold (Fig. 5.a,c) and the warm





(Fig. 5b,d) seasons. Clear monomodal distribution seems to be specific for the cold seasons while for the warm seasons the second mode mass concentration distribution seems to be predominated by $NO_3^-$. For the 155–612 nm size range, from details presented in Fig. 5c,d, it is quite clear that the pH values were at 2 or bellow 2 pH units. Moreover, in the present work it has been observed that the aerosol $H^+$ levels, inferred indirectly from the ion balance concept proposed by Hennigan

et al. (2015), showed statistically significant correlation with the $H^+$ loadings predicted by ISORROPIA-II in the forward mode (Pearson coefficient of 0.72, p < 0.001, at the 99.9 % confidence level). However, there are discrepancies between the two estimated $H^+$ levels, with those from thermodynamic model in orders of magnitude lower than those from the ionic balance (most probably due to the fact that the models account only for partial dissociation). It is also believed that estimation of the aerosol $H^+$ loading from the ionic balance was affected by the uncertainty coming from the propagation of

measurement error, especially under conditions where a slight anion deficit balance was inferred as an $H^+$ loaded system. Under these circumstances this trace could also represent an uncertainty source in $H^+$ levels estimated from the ion balance. However, overall in our study, conditions of increasing $H^+$ loadings were corresponding to decreasing aerosol pH and this observation would actually imply that most probably in the 155–612 nm particles size range, the aerosol particles were characterized by strong acidity (i.e., $H^+$ contributed mainly by completely dissociated strong acids, mainly in the $H_2SO_4$ and

$HNO_3$ form). Contributions from free acidity (dissociated $H^+$) or total acidity (free $H^+$ and undissociated $H^+$ bound to weak acids) is expected to be more important in all other remaining fraction, and especially in the 27.6–94.5 nm particles size range (see discussion also later in the text). However, it is strongly believed that highly confident estimated aerosol particles pH would allow better prediction in the chemical behaviour of organics especially if taken into account the fact that, at relatively low acidities, organic acids will dissociate, greatly contributing to the ion balance.

In the literature there is suggestion that the molar ratio approach is also often used as a proxy for aerosol pH estimation (Hennigan et al., 2015). However, the procedure is also highly susceptible to bias the results either due to exclusion of minor ionic species or due to the fact that it does not take account for the effects of aerosol water or species activities on particle acidity. In the present work, even under the limiting case when the aerosol regime was inferred to be highly acidic (i.e., samples with $NH_4^+/(Cl^- + NO_3^- + 2 \times SO_4^{2-})$ molar ratio less than 0.75), no statistically significant correlation has been

observed neither between the cation/anion molar ratio and $[H^+]$ from the ion balance nor between the molar ratio and forward-mode predictions of aerosol pH. In such conditions it was assumed that the molar ratio would not be a suitable tool to infer the acidity of atmospheric particles at the interest site but it could be a good parameter reliable to distinguish between alkaline and acidic particles.

Although measurements on gaseous $NH_3$ were not available the potential relationship between $NH_3/NH_4^+$ phase partitioning

approach (with ISORROPIA-II gaseous $NH_3$ predicted values) and predicted aerosol particles pH has also been investigated. For further interpretation, equilibrium gas-aerosol system was assumed to exist due to the fact that the length of sampling time-interval (36 h) was significantly exceeding the equilibrium time for submicron particles, which is on the order of seconds to minutes (Meng et al., 1995). As previously mentioned, in the present work it has been determined that only limited number of samples possess pH values < 0 while a significant fraction possess pH greatly higher than 0. Moreover,




the ISORROPIA-II thermodynamic model predicted that in the 94.5 and 612 nm size range not all of the $NH_3$ partitioned in the aerosol phase and also that a significant fraction was present in the gas phase.

For Iasi, north-eastern Romania, using data from ISORROPIA-II runs, predicted $NH_3$ concentrations suggested values as high as $0.52 \pm 0.28$ (0.46) µg m$^{-3}$ at RH < 40 %, $0.61 \pm 0.26$ (0.49) µg m$^{-3}$ at RH in the 40–60 % range and $0.96 \pm 0.54$

(0.92) µg m$^{-3}$ at RH > 60 % (mean ± stdev (median)). These values specific for the warm seasons are smaller than those reported in a modelling study by Backes et al. (2016), which by the reference case and the political NEC 2020-national emission ceilings coming into force 2020/30 scenario predict $NH_3$ abundances as high as 1.6 to 2.4 µg m$^{-3}$ (data extracted from $NH_3$ concentration for the reference case, i.e., fig. 3 in Backes et al., 2016, for north-eastern Romania). The $0.96 \pm 0.54$ (0.92) µg m$^{-3}$ value at RH > 60 %, predominating mainly during cold seasons, seems to be in reasonable agreement with the

$\leq 0.8$ µg m$^{-3}$ value modelled by Backes et al. (2016) over winter-time period. Moreover, from ISORROPIA-II runs performed at RH < 40 %, it was estimated that $(77.6 \pm 28.4)$ % or $(79.3 \pm 26.2)$ % (mean ± stdev) of the ISORROPIA-II predicted $NH_3$ value (both for $NH_4^+$ derived from raw IC data and, respectively, for the $NH_4^+$(total) fraction) could be present in the gaseous form. Nitric acid ($HNO_3$) and hydrochloric acid (HCl) distributions were in average of $(14.9 \pm 23.1)$ % (when $NH_4^+$ derived from raw IC data) or $(13.9 \pm 21.4)$ % (for $NH_4^+$(total) fraction) and, respectively, $(7.4 \pm 12.8)$ % (when $NH_4^+$

derived from raw IC data) or $(6.8 \pm 11.4)$ % (for $NH_4^+$(total) fraction). In each run an opposite trend was observed between the distribution of the $NH_3/NH_4^+$ partition and those specific for the $HNO_3/NO_3^-$ and $HCl/Cl^-$ systems.

By increasing the RH, the $NH_3/NH_4^+$ partition enhanced, and at RH in the 40–60 % range it could be observed that $(76.5 \pm 30.9)$ % or $(78.6 \pm 28.2)$ % (mean ± stdev) of the ISORROPIA-II predicted $NH_3$ value (both for $NH_4^+$ derived from raw IC data and, respectively, for the $NH_4^+$(total) fraction) could be present in the gaseous form. The nitric acid distribution in the

gas phase could account for $(17.5 \pm 26.9)$ % or $(16.1 \pm 24.5)$ % while that for hydrochloric acid was accounted by $(6.0 \pm 11.7)$ % or $(5.3 \pm 10.0)$ % for both $NH_4^+$ concentrations derived from raw IC data and, respectively, for the $NH_4^+$ (total) fraction.

At RH > 60 % only $(68.3 \pm 36.7)$ % or $(74.5 \pm 29.7)$ % (mean ± stdev) of the ISORROPIA-II predicted $NH_3$ value (both for $NH_4^+$ derived from raw IC data and, respectively, for the $NH_4^+$(total) fraction) was susceptible to be present in the gaseous

form. The nitric acid distribution in the gas phase could account for $(25.6 \pm 32.5)$ % or $(20.3 \pm 25.8)$ % while that for hydrochloric acid was accounted by $(6.1 \pm 10.6)$ % or $(5.2 \pm 9.0)$ % for $NH_4^+$ concentrations derived both from raw IC data and, respectively, for the $NH_4^+$(total) fraction. In other studies, thermodynamic equilibrium calculations predicted that all of the $NH_3$ was mainly susceptible for partitioning to the particle phase at the equilibrium and also that > 44 or 51 % of the investigated samples presented aerosols pH less than 0 (Hennigan et al., 2015). It is however believed that at the investigated

Romanian site (i.e., Iasi, north-eastern Romania), the atmosphere was enough rich in $NH_3$ such as to promote its partition into the particle phase and to exist also into gaseous form.

Using data from the ISORROPIA-II thermodynamic model, it has been observed that in the 155 and 612 nm size range (regardless the RH values) the aerosol ammonium fraction ($NH_4^+/(NH_3 + NH_4^+)$), calculated both for $NH_4^+$ derived from raw IC data or for the $NH_4^+$(total) fraction, was exceeding 0.20 value. Clear maxima of the ($NH_4^+/(NH_3 + NH_4^+)$) ratio were





observed at 381 nm (0.71 at RH < 40 %, 0.63 at RH in the 40–60 % range and 0.76 at RH > 60 % with $NH_4^+$ derived from raw IC data or 0.66 at RH < 40 %, 0.56 at RH in the 40–60 % range and 0.65 at RH > 60 % with $NH_4^+$(total) fraction. As seen in Fig. 5c,d, in the 155–612 nm size range, aerosols pH size distribution presents a clear cessation while in the 27.6–94.5 and 612–9940 nm size ranges, aerosol particles pH is often slightly below 8. Although not presented, in the same size

range (i.e., 155–612 nm) the aerosol ammonium fraction ($NH_4^+/(NH_3 + NH_4^+)$) shows a significant increase with values as high as almost 1, while in other two size ranges the aerosol ammonium fraction ($NH_4^+/(NH_3 + NH_4^+)$) was very low or close to 0 suggesting to some extent also the existence of gaseous $NH_3$. However, such behaviour would actually suggest that at the estimated pH values, enough $NH_3$ was always present in the gaseous phase such as an important fraction to partition also in the particle phase.

### 3.2.2 Seasonality of major water soluble ions

Table 3 shows at the investigated site (i.e., Iasi, north-eastern Romania) summary statistics (monthly based) for the meteorological variables, $PM_{10}$ and $PM_{2.5}$ mass concentrations and for major water soluble ions mass concentrations in the $PM_{2.5}$ fraction. In the $PM_{10}$ fraction, in comparison with the $PM_{2.5}$ fraction, increases within 13–80 % (35 %) (min–max (mean) in the mass concentration of $Cl^-$, 1–107 % (32 %) for $NO_3^-$, 1–170 % (17 %) for $SO_4^{2-}$, 38–185 % (63 %) for $HCO_3^-$,

14–171 % (41 %) for acetate, 4–136 % (22 %) or formate, 0–294 % (27 %) for oxalate, 16–48 % (32 %) for $Na^+$, 6–58 % (20 %) for $K^+$, 1–105 % (12 %) for $NH_4^+$(total), 28–83 % (46 %), for $Mg^{2+}$ and 33–123 % (61 %) for $Ca^{2+}$ were observed. However, the $PM_{10}$ and $PM_{2.5}$ mass concentrations fractions show statistically significant correlation with a ratio of 1.1 (Pearson coefficient of 0.99, p < 0.001, at the 99.9 % confidence level). Higher mass concentrations of specific water soluble ions (i.e., $Cl^-$, $NO_3^-$, $K^+$, $NH_4^+$ and to some extent also $SO_4^{2-}$) during the cold seasons than during the warm seasons are most

probable the result either of the combination of increased strength of pollution sources and meteorological effects (inducing most probable lower mixing heights or even temperature inversion) or of different chemical/photochemical processing. In a study at Kanpur, India, the authors report also higher abundances of particulate $NO_3^-$, $SO_4^{2-}$, $NH_4^+$, $K^+$ in winter than in summer but the authors claim that $CaCO_3$ could be mainly responsible on $NO_3^-$ abundance (Sharma et al., 2007).

Seasonal variations for selected water-soluble ionic components in the $PM_{2.5}$ fraction are shown in Fig. 6a-h while Fig. 6i

presents the variation of the mixed layer depth at the investigated site. Fine particulate $Cl^-$, $NO_3^-$, $K^+$, $NH_4^+$(total), and to some extent even $SO_4^{2-}$, seem to exhibit distinct seasonal variations with maxima during the cold seasons and minima over the warm seasons. However, it might be that changes in the mixed layer depth contributed also to the observed seasonality. The summer minima observed for both $NO_3^-$ and $NH_4^+$ ions is not extremely surprising since their most predominant form, i.e. $NH_4NO_3$, is volatile and tends to dissociate and stay in gas phase at high temperatures. Coarse particulate $C_2O_4^{2-}$, $Ca^{2+}$

and $Na^+$ did not show much variation with respect to seasons. However, $SO_4^{2-}$ and $C_2O_4^{2-}$ show similar pattern (implying most probably common sources) and $Ca^{2+}$ variation reflects possible prevalent contribution from soil dust. Higher winter than summer ion concentrations are reported by Sharma et al. (2007) for Kanpur (India) while Ianniello et al. (2011) report opposite trends for Beijing (China).





Particulate $Cl^-$ mass concentrations show clear seasonal pattern with higher values during the cold seasons than during the warm seasons (Fig. 6a). In both $PM_{2.5}$ and $PM_{10}$ fractions $Cl^-$ mass concentration correlated statistically significant with RH (Pearson coefficient higher than 0.71, p = 0.010, at the 95.4 % confidence level, positive slope), temperature (only for $PM_{10}$ fraction, Pearson coefficient higher than 0.59, p = 0.045, at the 95.4 % confidence level, negative slope), particle loading

(Pearson coefficient higher than 0.60, p = 0.038, at the 95.4 % confidence level, positive slope) and mixed layer depth (Pearson coefficient higher than 0.69, p = 0.012, at the 95.4 % confidence level, negative slope). Its maxima during the cold seasons might be the result of increased coal burning activities for heating purposes or due to the use of NaCl in winter to avoid the effects of slippery roads. These observations are in agreement with those of other studies conducted at eastern European sites (Arsene et al., 2011; Alastuey et al., 2016). However, $Cl^-$ mass concentration follows similar pattern with that

of $K^+$ (tracer of biomass burning) allowing us suggesting that over the cold seasons wood burning might become an important heating source (Christian et al., 2010; Akagi et al., 2011).

As seen in both Table 3 and Fig. 6b, $NO_3^-$ ion presents clear temporal pattern with maxima during the cold seasons and minima during the warm seasons. The inset distribution presented within $NO_3^-$ seasonal variation (Fig. 6b) is clearly reflecting that over the warm seasons the coarse fraction, in comparison with fine fraction, might also bring significant

contributions to the aerosol atmospheric burden. In our work fine particulate $NO_3^-$ mass concentrations varied from 0.31 to 3.62 µg m$^{-3}$ (Table 3) and these data are in very good agreement with those predicted for Europe in a modelling study performed by Backes et al. (2016). The data measured in the present work over the cold seasons (3.62 ± 1.10 µg m$^{-3}$ over January, February, December as the coldest months of the year and 2.65 ± 0.38 µg m$^{-3}$ over January, February, March, October, November, December; averaged data from Table 3) seem to be in reasonable good agreement with those predicted

for Europe in the modelling study performed by Backes et al. (2016) (abundances predicted by the reference case and the political scenario NEC 2020-national emission ceilings coming into force 2020/30 in the 2.4 to 3.2 or even higher µg m$^{-3}$ range over winter; data extracted from $NO_3^-$ concentration in $PM_{2.5}$, i.e., fig. 5 in Backes et al., 2016, for north-eastern Romania). The 0.59 ± 0.30 µg m$^{-3}$ $NO_3^-$ measured concentration in $PM_{2.5}$ over warm seasons (including April month characterized by predominant air masses buoyancies events; 0.44 ± 0.12 µg m$^{-3}$ $NO_3^-$ measured concentration without April

month) is lower than that predicted by Backes et al., (2016) model (abundances as high as 0.8 µg m$^{-3}$ over the summer) and, as in $NH_4^+$ case, this might reflect the susceptibility of $NO_3^-$ to be mainly transferred in the gas phase over the warm seasons. From the data presented in Table 3 it has been also observed that $NO_3^-$ mass concentrations, in both $PM_{2.5}$ and $PM_{10}$ fractions, correlated statistically significant with RH values (Pearson coefficient higher than 0.84, p < 0.001, at the 99.9 % confidence level, positive slope), temperature (Pearson coefficient higher than 0.92, p < 0.001, at the 99.9 % confidence

level, negative slope), mixing layer depth (Pearson coefficient higher than 0.84, p < 0.001, at the 99.9 % confidence level, negative slope) and particles loading (Pearson coefficient higher than 0.65, p = 0.021, at the 95.4 % confidence level, positive slope) in both $PM_{2.5}$ and, respectively, $PM_{10}$ fractions. However, as previously presented, since highly acidic aerosols are expected over all seasons this will most probably affect a variety of processes and definitely the partitioning of $HNO_3$ to the gas phase, resulting in low nitrate aerosol levels. Guo et al. (2015) report also low nitrate aerosol levels during





summer. Moreover, since $NO_3^-$ heterogeneous formation (i.e., condensation or absorption of $NO_2$ in moist aerosols or $N_2O_5$ oxidation and $HNO_3$ condensation) generally relates to RH and the particulate loading (Wang et al., 2006; Ianniello et al., 2011) it is believed that at the investigated site this process might be of similar importance as gas–particle conversion (implying mainly oxidation by photochemical processes of precursor gases, such as $NO_x$, to nitrate via $HNO_3$ formation).

The high concentration of $NO_3^-$ during the cold seasons might also be due to higher concentrations of $NH_3$ in the atmosphere, from unaccounted yet sources, available to neutralize $H_2SO_4$ and $HNO_3$ (as will be later shown in the text). Moreover, the high relative humidity conditions over the cold seasons could offer suitable conditions such as significant fractions of $HNO_3$ and $NH_3$ to be dissolved in humid particles, therefore enhancing fine particulate $NO_3^-$ and $NH_4^+$ in the atmosphere (Pathak et al., 2009, 2011; Ianniello et al., 2010; Sun et al., 2010). Kai et al. (2007), from measurements
performed in October 2004 in Beijing, China, concluded also that higher RH, higher stable atmosphere structure and higher concentration of $NH_3$ can lead to a higher transformation ratio of $NO_x$ to $NO_3^-$.

For particulate $SO_4^{2-}$ maxima are observable during the cold seasons but also during the warm seasons. However, $SO_4^{2-}$ mass concentrations didn't show statistically significant correlation with any of the measured meteorological parameters. From the data presented in Fig. 6c it can be observed that its monthly mean mass concentrations seasonality seems to be not
that clear as that of $NO_3^-$ and $NH_4^+$ suggesting also a possible regional characteristic of $SO_4^{2-}$ formation (Wang et al., 2016). In Iasi, north-eastern Romania, higher $SO_4^{2-}$ concentrations during the cold seasons may be due to increased coal and wood burning combined with poor dispersion (low mixed layer depth) and high RH values. High relative humidity is seen as important factor responsible for the conversion of $SO_2$ to $SO_4^{2-}$ (Kadowaki, 1986) with a significant enhancement of $SO_4^{2-}$ production rate in aqueous phase (Sharma et al., 2007). Over the warm seasons the temporal trend observed for $SO_4^{2-}$ is
thought to be induced by the high rate of photochemical activity (although not shown, higher solar radiation intensity and temperature measured at AMOS) and atmospheric oxidation (most probably high oxidant concentration, such as OH radicals) which increases the oxidation of $SO_2$ and its conversion rate to $SO_4^{2-}$ (Stelson and Seinfeld, 1982; Stockwell and Calvert, 1983; Kadowaki, 1986; Wang et al., 2005).

As for particulate $NH_4^+$(total), both the data in Table 3 and Fig. 6f show that this chemical species presents clear seasonal
pattern with maxima during the cold seasons and minima over the warm seasons, observation in agreement with reports at other European (Schwarz et al., 2012; Bressi et al., 2013; Tositti et al., 2014; Voutsa et al., 2014) or non-European sites (Sharma et al., 2007; Wang et al., 2016). The behaviour seems to be opposite to Ianniello et al. (2011) report and this might be due to the fact that changes in the mixed layer depth (as seen in Fig. 6i) played a very important role in its distribution. Although $NH_4^+$ is expected to present enhanced evaporation over warm seasons, it seems that in Iasi, north-eastern Romania,
an important fraction of the particulate $NH_4^+$ (~ 45 %) is lost during the cold seasons mainly due to the influence of meteorological parameters such as those inducing emissions from dew evaporation with fast temperature increase (see also discussion later in the text). Higher particulate $NH_4^+$ abundances over the cold seasons (although a maximum is expected over the warm seasons) are mainly related to changes in the mixed layer depth all around the year (i.e., mixed layer depth





development associated with dilution effect responsible for the warm seasons minima, while lower mixed layer depth associated with accumulation effect responsible for the cold seasons maxima).

In our work $NH_4^+$(total) measured concentration varied from 0.8 to 2.09 µg m$^{-3}$, a range which is much lower than that reported by Meng et al. (2011) for a more polluted site (i.e., Beijing, China, with concentrations varying between 4.73 to 9.04 µg m$^{-3}$ within various seasons). However, the data measured in the present work over the cold seasons (2.03 ± 0.30 µg m$^{-3}$ over January, February, December as the coldest months of the year and 1.65 ± 0.23 µg m$^{-3}$ over January, February, March, October, November, December; averaged data from Table 3) seem to be in reasonable good agreement with those predicted for Europe in a modelling study performed by Backes et al. (2016) (abundances predicted by various models in the 1.6 to 2.4 µg m$^{-3}$ range over winter; data extracted from $NH_3$ concentration for the reference case, i.e., fig. 4 in Backes et al., 2016, for north-eastern Romania). The 0.90 ± 0.09 µg m$^{-3}$ $NH_4^+$(total) measured concentration over warm seasons is much lower than that predicted by Backes et al., (2016) model (abundances as high as 1.6 µg m$^{-3}$ over the summer) but the discrepancy might actually reflect the limitation of the experimental measurement techniques in regard to $NH_4^+$ volatility.

Moreover, in Iasi, north-eastern Romania, it has been also observed that $NH_4^+$(total) mass concentration correlated statistically significant with RH values (Pearson coefficient higher than 0.80, p = 0.001, at the 95.4 % confidence level, positive slope), temperature (Pearson coefficient higher than 0.85, p < 0.001, at the 99.9 % confidence level, negative slope, temperature increase causing particulate $NH_4^+$(total) mass concentration decrease), mixed layer depth (Pearson coefficient higher than 0.80, p = 0.001, at the 95.4 % confidence level, negative slope, mixed layer depth increase causing particulate $NH_4^+$(total) mass concentration decrease), and particle loading (Pearson coefficient higher than 0.64, p = 0.023, at the 95.4 % confidence level, positive slope) in both $PM_{2.5}$ and, respectively, $PM_{10}$ fractions. $PM_{2.5}$ fraction showed statistically significant correlation also with the mixing depth (Pearson coefficient higher than 0.67, p = 0.016, at the 95.4 % confidence level).

The seasonal variation of particulate $NH_4^+$(total) follows especially those of particulate $NO_3^-$ and $Cl^-$, which would indicate that most probably $NH_4^+$(total) largely originates from the neutralization between $NH_3$ and acidic species of the afore mentioned ions (Wang et al., 2006) or that the species were likely internally mixed and came from similar gas-to-particle processes (Huang et al., 2010). Although in the present work gaseous $NH_3$ was not measured, ISORROPIA-II thermodynamic equilibrium model runs predicted that at the investigated site the atmosphere was often in gaseous ammonia-rich state regardless the RH values (the $[NH_3]/([HNO_3] + [HCl])$ strongly exceeding 1). Studies undertaken over various seasons (but with RH values predominantly below 50 % over the cold and > 70 % over the warm seasons) showed that temperature presented a significant positive correlation with $NH_3$, a behaviour suggesting actually that high temperature would facilitate $NH_3$ release from its sources while both low wind speed and low mixed layer depth would cause intensive atmospheric stability, limiting the dispersion of air pollutants (Zhao et al., 2016). ISORROPIA-II runs predicted for $NH_3$ in Iasi, north-eastern Romania, values as high as 0.96 ± 0.54 (0.92) µg m$^{-3}$ at RH > 60 % often prevailing over the cold season, which were significantly higher than those estimated for other RH values and higher temperatures. It might be that over cold seasons, at the interest location, the meteorological parameters would play a more important role than expected in the





physico-chemical processes controlling the atmospheric burden of chemical species. Since over the cold seasons RH values were as high as almost 69 % it is supposed that at this high RH value the dew point was closer to the current air temperature. Moreover, over the cold seasons at the investigated site, changes of about 10 °C in the ambient temperature from day-to night were quite often experienced and such a large driving force it is believed actually to enhance the evaporation rate

process which would become the limiting step controlling $NH_4^+$ abundance in both particulate and gaseous phase. In the atmosphere, gaseous $NH_3$ concentration, concentrations of atmospheric acidic gases, characteristics of pre-existing aerosols, air temperature, and humidity are supposed to play a significant role in generating particulate $NH_4^+$. In a study performed by Zhao et al. (2016), the authors report about the fact that $NH_3$ to $NH_4^+$ conversion was favoured under the conditions of low temperature and high relative humidity. It might be that also in our study, high $NH_3$ ISORROPIA-II estimated level (under

cold associated very high RH conditions) is the factor of utmost importance facilitating fine particulate ammonium salt formation during this period of the year. However, in overall particulate $NH_4^+$(total) associated mainly with $NO_3^-$ and $SO_4^{2-}$ in both $PM_{2.5}$ and $PM_{10}$ fraction and therefore it is suggested that enhanced fine particulate ammonium salt formation due to available $NH_3$ could be an important cause of $PM_{2.5}$ and $PM_{10}$ pollution in the urban atmosphere of Iasi, north-eastern Romania.

It has been observed that at the investigated site $C_2O_4^{2-}$ and $SO_4^{2-}$ ions present similar behaviour and $C_2O_4^{2-}$ maxima during summer may suggest photochemical and/or biogenic contribution to its abundance (Laongsri and Harrison, 2013). Sodium ion, tracer of sea-salt or NaCl aerosols, shows higher concentrations during spring, a season highly predominated by long range transport of air masses from S-SE sector with contributions from natural sources, especially sea-spray aerosols from the Black Sea. Particulate $K^+$ mass concentrations show also clear seasonal pattern with higher values during cold than

during the warm seasons (Fig. 6e), and this could be due to increased wood burning process corroborated with the height of the mixed layer depth. However, particulate $K^+$ mass concentrations revealed also some maxima during the months with intense agricultural biomass burning for field clearing (i.e., April, July, and September). It has been observed that $K^+$ mass concentrations follows similar pattern with that of $Cl^-$ (Pearson coefficient of 0.79, p = 0.002, at the 95.4 % confidence level) but not with that of $SO_4^{2-}$ allowing us to suggest that intense wood burning might be a possible common source for $K^+$ and

$Cl^-$ species (Christian et al., 2010; Akagi et al., 2011).
High mass concentrations of $Ca^{2+}$ and $Mg^{2+}$ (with $Mg^{2+}$ shown only in Table 3 but not in Fig. 6), as soil/dust tracers, were observed especially in spring and summer. Over these seasons, cessation in precipitation frequency and high wind speed contribute to the observed behaviour. During cold seasons, low wind speeds might prevent soil getting reborn in the atmosphere and hence lower values for these ions during these periods of the year. However, $Mg^{2+}$ and $Ca^{2+}$ as mineral ions

did not show any correlations neither with $PM_{2.5}$ nor with $PM_{10}$ fractions and these observations allows us suggesting that most probable $NH_3$ rather than mineral ions would play a more important role in inorganic secondary particle formation.





### 3.2.3 Stoichiometry of (NH$_4$)$_2$SO$_4$, NH$_4$NO$_3$ and NH$_4$Cl

Table 4 presents the correlation matrix (Pearson coefficients) for major ionic species (Cl$^-$, NO$_3^-$, SO$_4^{2-}$, CH$_3$COO$^-$, HCOO$^-$, C$_2$O$_4^{2-}$, HCO$_3^-$, Na$^+$, NH$_4^+$(total), K$^+$, Mg$^{2+}$, Ca$^{2+}$) in PM$_{2.5}$ aerosol particles both for the cold seasons (Table 4.a) and the warm seasons (Table 4.b). Although similar correlations have been observed also for the PM$_{10}$ fraction, for the correlation

matrix analysis PM$_{2.5}$ fraction has been selected as this is the most representative. For the cold seasons, data in Table 4.a shows significant correlations among many chemical pairs (at the 99.9 % confidence level). It might be that at lower temperatures chemical species such as (NH$_4$)$_2$SO$_4$, NH$_4$NO$_3$, (NH$_4$)$_2$C$_2$O$_4$, and other associations are formed in the atmosphere with significant contributions to PM$_{2.5}$ fraction. Over the warm seasons (NH$_4$)$_2$SO$_4$, Mg(NO$_3$)$_2$, and NaCl chemical associations seems to be the most important. However, Ca(HCO$_3$)$_2$ might play a role in both during the cold and

warm seasons. Over the cold seasons it has been also observed that K$^+$ ion showed statistically significant correlation with many inorganic (NO$_3^-$, Cl$^-$ and SO$_4^{2-}$) and organic (HCOO$^-$ and C$_2$O$_4^{2-}$) anions. Such observations allow us suggesting that these species might have common biomass burning sources known to consist mostly of internally mixed potassium salts (K$_2$SO$_4$, KNO$_3$, KCl and probably HCOOK and K$_2$C$_2$O$_4$) (Ianniello et al., 2011 and references therein).

In ambient atmosphere, inorganic ammonium salts such as ammonium bisulfate (NH$_4$HSO$_4$), ammonium sulfate

((NH$_4$)$_2$SO$_4$), ammonium nitrate (NH$_4$NO$_3$), and ammonium chloride (NH$_4$Cl) are known to be produced by gas-to-particle conversion processes. In the present work, from the ionic balance analysis NH$_4^+$ ion was assigned as the most critical parameter in the chemical composition analysis of aerosol particles in Iasi, north-eastern Romania (with 274 analysed samples representing 25 % from the total, i.e. 1092, highly deficient in cations). However, as previously presented, when the missing NH$_4^+$ (estimated by undertaking the procedure suggested by Arsene et al., 2011) has been taken into account the

ionic balance has significantly improved. Moreover, as a depth inside investigation procedure, NH$_4^+$ potential neutralization factor, defined as the molar ratio of NH$_4^+$ to the theoretical one, assuming almost complete conversion of acidic species (H$_2$SO$_4$, HNO$_3$ and HCl) to (NH$_4$)$_2$SO$_4$, NH$_4$NO$_3$ and NH$_4$Cl, was under scrutiny in the present work.

Figures 7a,b present the relationship between molar concentrations of fine particulate NH$_4^+$ (i.e., from raw IC data and also in the total form estimated under the assumptions from Arsene et al., 2011), and that of particulate SO$_4^{2-}$ for both the cold

(Fig. 7a) and warm (Fig. 7b) seasons. The correlations between molar concentrations of particulate NH$_4^+$ and SO$_4^{2-}$ are statistically significant in either situations (cold seasons, NH$_4^+$ from raw IC data, Pearson coefficient of 0.97, p < 0.001, at the 99.9 % confidence level, and for NH$_4^+$(total), Pearson coefficient of 0.96, p < 0.001, at the 99.9 % confidence level; warm seasons, NH$_4^+$ from raw IC data, Pearson coefficient of 0.98, p < 0.001, at the 99.9 % confidence level and for NH$_4^+$(total), Pearson coefficient of 0.97, p < 0.001, at the 99.9 % confidence level).

While for raw data, both during the cold and warm seasons, the molar ratios are either slightly higher or lower than 1, for NH$_4^+$(total) the ratio is 1.76 for the cold seasons and 1.02 for the warm seasons. These observations would allow us suggesting the existence of enough NH$_3$ for complete neutralization of H$_2$SO$_4$ and also a predominance of particulate (NH$_4$)$_2$SO$_4$ in agreement with Ianniello et al. (2011) observations. Moreover, as shown in Table 4, for particulate SO$_4^{2-}$ and





$NH_4^+$(total) ions pair the correlation was statistically significant (with Pearson coefficients of 0.96, p < 0.001 for cold and 0.97, p < 0.001 for warm seasons, at 99.9 % confidence level) suggesting that $(NH_4)_2SO_4$ could be formed by the reaction of $H_2SO_4$(g) with $NH_3$(g) in either situations. However, the 1.76 value for the $[NH_4^+]$(total)/$(2\times[SO_4^{2-}])$ molar ratio during the cold seasons will indicate that there should still be particulate $NH_4^+$ potentially available to combine with other anions (or

existence of enough excess ammonia to neutralize acidic species such as $HNO_3$ and $HCl$). Zhao et al. (2016) report $[NH_4^+]/(2\times[SO_4^{2-}])$ molar ratio of 1.54 ($R^2$ = 0.63) with an average ratio of 2.08, indicating the complete neutralization of $H_2SO_4$ and a predominance of $(NH_4)_2SO_4$ in sulfate salts during the cold seasons.

In the present work it has been observed that particulate $NO_3^-$ is the major inorganic species in the 0.155–0.612 μm fraction, especially during months with very high RH. Moreover, the relationship between $NO_3^-$ and $SO_4^{2-}$, in both the slopes and

correlation coefficient terms, was much better over the cold seasons than over the warm seasons. For $(NH_4^+, SO_4^{2-})$ ions pair the correlation was statistically significant only during the cold seasons. These observations would allow us suggesting that over these seasons heterogeneous formation of particulate $NO_3^-$ was probably more important than homogeneous formation route.

Unfortunately, at present, no measured $NH_3$ values are available for the interest site but it is still believed that in the

atmosphere of Iasi, north-eastern Romania, sufficient $NH_3$ in air exists such as significantly to promote both the homogeneous and heterogeneous formation of $NO_3^-$ in the collected aerosol particles. Its formation routes might involve either homogeneous reaction between gaseous $HNO_3$ and $NH_3$ (expected to predominate over the warm seasons) (Ianniello et al., 2011) or the heterogeneous reaction between $NH_3$ and the products formed through the hydrolysis of potential present $N_2O_5$ on the surface of the pre-existing moist aerosols under relatively high humidity (Pathak et al., 2011; Shon et al., 2013).

Actually, in the atmosphere gaseous $NH_3$ can influence both the aerosol phase inorganic ions but also the aqueous phase hydronium ion ($H^+$) distribution in aerosols. The concentration of $H^+$ in aqueous aerosols, or pH, is mainly determined by the balance of acidic ionic components with basic ones. During our measurements period, the atmosphere of Iasi, north-eastern Romania, most probably was frequently abundant with enough $NH_3$ since, during both the cold and warm seasons, almost half (or slightly higher) of the collected samples were found alkaline with pH values fluctuating between 7 and 8. The

remainder samples were acidic and presented pH values ranging mainly from 1 to 3. Zhao et al. (2016) report also about the fact that, on average, a ± 25 % perturbation in $NH_3$ level could lead to a 0.14 unit pH increase for 25 % perturbation above the measured value and a 0.23 unit pH decrease for a perturbation below the measured value (75 % case). The authors concluded that the relatively flat increasing tendency of pH with increasing $NH_3$ level reflected and supported their conclusion that sufficient $NH_3$ was frequently present in wintertime atmosphere and also that the fine collected particulates

were almost fully neutralized by $NH_3$.

Figures 7c,d,e,f show the relationship between molar concentrations of fine particulate $NH_4^+$ and sum of the molar concentrations of fine particulate $SO_4^{2-}$ and $NO_3^-$ (Fig. 7c,d) and, respectively, fine particulate $NH_4^+$ and the sum of the molar concentrations of fine particulate $SO_4^{2-}$, $NO_3^-$ and $Cl^-$ (Fig. 7e,f) for both the cold and warm seasons. From details given in Fig. 7c,d,e,f it can be easily observed that $NH_4^+$, in both derived from raw IC data and total forms, was in deficit





over the cold and warm seasons and under these circumstances it is believed that both particulate $NO_3^-$ and $Cl^-$ could be associated with other alkaline species or be part of acidic aerosol. Since the neutralizing capacity of $NH_4^+$ toward $SO_4^{2-}$, $NO_3^-$ and $Cl^-$ acidic species might give a rough indication about potential particles acidity (Li et al., 2015) from Fig. 7c,d,e,f it is quite clear that at AMOS site, if $NH_4^+$ in derived from raw IC data is used, the neutralization ratios in the investigated

particles are less than unity, suggesting that atmospheric particles are most likely acidic and also that a more complex chemistry is ongoing in regard with $HNO_3$ and $HCl$ species. However, when $NH_4^+$(total) (defined as the sum between that derived from raw IC data and the part estimated by using the rationale of Arsene et al. (2011)) the ratios approaches 1 suggesting a possible complete neutralization of particles acidity. From details in Fig. 7e,f it can be easily seen that even for the [$NH_4^+$] and ($2\times$[$SO_4^{2-}$] + [$NO_3^-$] + [$Cl^-$]) molar ratio, actually the available $NH_4^+$ is not being enough to compensate for

other species. However, it should be noted that the $Cl^-$ is not significantly influencing the neutralization of the $PM_{10}$ particles acidity. In a study performed by Zhao et al. (2016) the authors reported [$NH_4^+$]/($2\times$[$SO_4^{2-}$] + [$NO_3^-$]) molar ratio of 0.86 ($R^2$ = 0.78) with an average ratio of 1.05, and an [$NH_4^+$]/($2\times$[$SO_4^{2-}$] + [$NO_3^-$] + [$Cl^-$]) molar ratio of 0.60 ($R^2$ = 0.86) with an average ratio of 0.67). Details presented in Fig. 7c,d clearly show that when $NH_4^+$(total) is taken into account a complete neutralization of $H_2SO_4$ and $HNO_3$ can be achieved during the cold seasons (Fig. 7c) while during the warm seasons (Fig.

7d) the molar ratio is becoming slightly lower than 1 (i.e., 0.95). During the warm seasons, according to Seinfeld and Pandis (1998), high temperature and low relative humidity would be favourable for $NH_4^+$ to reach a minimum concentration since it will mainly be transformed into $NH_3$ and, actually, temperature > 25 °C, such as those often encountered at the investigated site over the warm seasons, are known to prevent formation of significant amount of particulate $NH_4NO_3$ (Adams et al., 1999). Under these circumstances, a cessation in the [$NH_4^+$](total)/([$NO_3^-$] + $2\times$[$SO_4^{2-}$]) molar ratio is to be expected since a

significant fraction of the available particulate $NH_4^+$ will be in equilibrium with gaseous $NH_3$. Moreover, at the investigated site, during the cold seasons temperature and relative humidity were of 5.3 ± 3.9 °C and, respectively, (65.3 ± 12.8) % and, according with details presented in Table 4, the ($NO_3^-$, $NH_4^+$(total)) pair presented significant correlation (with Pearson coefficient of 0.98, p < 0.001, at the 99.9 % confidence level) only during the cold seasons (low temperature and high relative humidity being favourable for particulate $NH_4NO_3$ formation, Stelson and Seinfeld, (1982)). During the warm

seasons (temperature and relative humidity of 18.9 ± 3.8 °C and, respectively, 40.5 ± 7.7 %) the ($NO_3^-$, $NH_4^+$(total)) pair correlation is of very poor significance most probably due to the influence of meteorological conditions which were not favourable for particulate $NH_4NO_3$ formation (increasing temperature and decreasing relative humidity limit the production of $NH_4NO_3$ aerosol (Matsumoto and Tanaka 1996; Utsunomiya and Wakamatsu 1996; Alastuey et al. 2004)).

As presented in Sect. 3.2.2, at the investigated site reductions in $NO_3^-$ and $SO_4^{2-}$ abundances were observed over warm

seasons and these may cause increases in gas phase $NH_3$ with potential more $NH_3$ deposited closer to emission sites, and therefore decreased sensitivity of inorganic $PM_{2.5}$ to $NH_3$ over these seasons. However, as an important aerosol fraction was initially acidic, reductions in $SO_4^{2-}$ is also expected to cause some $NH_3$ to become available for $NH_4NO_3$ formation, which could increase the sensitivity of $PM_{2.5}$ to $NH_3$ emissions. Actually in the atmosphere, $H_2SO_4$ and $HNO_3$ both are known to compete for reacting with $NH_3$ to form ($NH_4$)$_2SO_4$ and $NH_4NO_3$. The reaction rate constant of ($NH_4$)$_2SO_4$ aerosol formation





(by an irreversible reaction due to $H_2SO_4$ affinity for $NH_3$), known to be as high as $1.5 \times 10^{-4}$ sec$^{-1}$ (Harrison and Kitto, 1992), is almost similar with the reaction rate constants for $NH_4NO_3$ formation (by a balanced reaction due to $NH_3$ affinity for $HNO_3$), which is of the order of $1.59 \times 10^{-4}$ m$^3$ $\mu$mol$L^{-1}$ s$^{-1}$ (Pandolfi et al., 2012; Behera et al., 2013). However, both reaction rate constants are much higher than that between $NH_3$ and HCl ($5.16 \times 10^{-5}$ m$^3$ $\mu$mol$L^{-1}$ s$^{-1}$) (Behera and Sharma,

2012) and this is what most probably dictates the competition of any of the particulate $SO_4^{2-}$, $NO_3^-$ and Cl$^-$ for the available $NH_4^+$. Usually, when sufficient amount of $NH_3$ is available (for neutralization of $H_2SO_4$ and $HNO_3$), fine mode $(NH_4)_2SO_4$ and $NH_4NO_3$ will be formed via reactions R1 (Cziczo et al., 1997; Zhang et al., 2015) and R2 (Fountoukis and Nenes, 2007; Zhang et al., 2015),

$$2NH_{3(g)} + H_2SO_{4(aq)} = (NH_4)_2SO_{4(aq)} \tag{R1}$$

$$NH_{3(g)} + HNO_{3(g)} = NH_4NO_{3(s)} \tag{R2}$$

while, when $NH_3$ limited environment, coarse mode $NO_3^-$ will be formed through reaction R3 involving $Mg^{2+}$ (not $Ca^{2+}$) mineral ion,

$$MgCO_{3(aq)} + 2HNO_{3(g)} = Mg(NO_3)_{2(s)} + H_2O + CO_{2(g)} \tag{R3}$$

During the warm seasons, however, higher concentrations of $(NH_4)_2SO_4$ compared to $NH_4NO_3$ are expected since $(NH_4)_2SO_4$

is less volatile than $NH_4NO_3$ (Utsunomiya and Wakamatsu, 1996) and, moreover, $NH_4NO_3$ will form only when available excess $NH_3$ will react with $HNO_3$. Backes et al. (2016), from their modelling study, suggest that a reduction of $NH_3$ emissions by 50 % may lead to a 24 % reduction of the total $PM_{2.5}$ concentrations in northwest Europe, with the reduction mainly driven by reduced formation of $NH_4NO_3$. However, even in the case of assuming a drastic reduction of $NH_3$, the over Europe $NH_3$ concentration in the atmosphere seems to be high enough to saturate the reaction forming $SO_4^{2-}$ particles, but in

contrary to the reaction with $H_2SO_4$, the $NH_3$ concentration in the atmosphere will not be high enough to saturate the reaction with $HNO_3$ to form $NH_4NO_3$ particles. The reduced formation of $NH_4NO_3$ particles may leads to a shift towards gas phase $HNO_3$ which is intensified in winter. In our study, from ISORROPIA-II thermodynamic model higher shifts towards gas phase $HNO_3$ have been estimated at higher RH values which were mainly prevailing during cold seasons. The elevated concentration of gas phase $HNO_3$ may lead to an increased condensation onto existing particles such as sodium chloride

(NaCl), and the replacement of Cl$^-$ with $NO_3^-$ may result in an increasing gas phase concentration of HCl in the atmosphere (similar processes described also in Arsene et al. (2011)).

Since in the atmosphere, apart $NH_4NO_3$ and $NH_4Cl$ salts, other less volatile nitrate and chloride containing species might be also present, the potential of the fine particulate $NO_3^-$ and Cl$^-$ to be chemically bound as relatively non-volatile salts of $Ca^{2+}$, $Mg^{2+}$, $K^+$ or $Na^+$, has been also investigated. The free $NO_3^-$ and Cl$^-$ concentrations, defined as the fractions of nitrate and

chloride in excess which are not bound with the alkali or alkaline earth metals, have been estimated for both the cold and warm seasons according to the concept described in Ianniello et al. (2011). Zero or negative values of free $NO_3^-$ and Cl$^-$ imply that $NH_4NO_3$ and $NH_4Cl$ are not present. From the estimated free $NO_3^-$ and Cl$^-$ concentrations, with similar contributions in both the $PM_{2.5}$ and $PM_{10}$ fractions (i.e., over cold seasons 6.5E–03 $\pm$ 7.9E–03 $\mu$mol m$^{-3}$ (0.4 $\pm$ 0.5 $\mu$g m$^{-3}$) for $NO_3^-$ and negative values for free Cl$^-$; over the warm seasons 1.6E–03 $\pm$ 1.8E–03 $\mu$mol m$^{-3}$ (0.1 $\pm$ 0.1 $\mu$g m$^{-3}$) for $NO_3^-$





and negative values for free Cl⁻, mean ± stdev), allow us suggesting the potential presence of $NH_4NO_3$ especially during the cold seasons but not of $NH_4Cl$ neither during the cold nor during the warm seasons. During the cold seasons, particulate $NO_3^-$ didn't show correlation neither with $Ca^{2+}$ nor with $Mg^{2+}$, but it showed significant correlation with $K^+$ (r = 0.85, p < 0.001, at the 99.9 % confidence level from Table 4), indicating possible formation of non-volatile $KNO_3$ salt along with

$NH_4NO_3$. Over the warm seasons, fine particulate $NO_3^-$ didn't show correlation with $K^+$, but it showed significant correlation with $Na^+$ and $Mg^{2+}$ (r = 0.71, p < 0.001 and r = 0.86, p < 0.001 at the 99.9 % confidence level, from Table 4), indicating possible formation of non-volatile $NaNO_3$ and $Mg(NO_3)_2$ but not of $Ca(NO_3)_2$ salts. Since over the warm seasons fine particulate $NO_3^-$ didn't show correlation with $NH_4^+$ (expected at high temperatures) but it showed statistically significant correlation with $HCO_3^-$ (r = 63, p < 0.001 at the 95.4 % confidence level, from Table 4) these observations would actually

suggest possible formation of particulate $NO_3^-$ rather via mineral route than via homogeneous reactions. The significant correlation of fine particulate $NO_3^-$ with $Mg^{2+}$ during the warm seasons (r = 0.86, p < 0.001 at the 99.9 % confidence level, from Table 4) could increase as importance due to the fact that, as $NH_4^+$ was not available, neutralization of $HNO_3$ could occur on coarse soil-driven particle rich in $Mg^{2+}$ (Matsumoto and Tanaka, 1996; Utsunomiya and Wakamatsu, 1996; Alastuey et al., 2004).

In the present work moreover, the relative humidity at deliquescence (RHD, which is the relative humidity at which solid particles liquefy) was (65.6 ± 2.0) % over warm seasons, while that over the cold seasons was of (74.1 ± 2.7) %. From the literature it is already known that species such as $(NH_4)_2SO_4$ and $NH_4HSO_4$, at room temperature, uptakes water (deliquesces) at RH of (79 ± 1) % and, respectively, at RH of 39 % (Cziczo et al., 1997). While until a very low RH (33 ± 2) %, crystallization point) is reached $(NH_4)_2SO_4$ aerosol particles might remain in "metastable" or supersaturated state liquid

phase (only thereafter a solid might be formed), for $NH_4HSO_4$ it has been shown that solid phase is difficult to form. In the present work it might be that only over July (RH of 32.95 %), August (RH of 37.12 %) and September (RH of 31.56 %) months formation of solid $(NH_4)_2SO_4$ or $NH_4HSO_4$ could occur.

Pure $NH_4NO_3$ deliquesces at 62 % RH and there is suggestion that sometime even at 8 % RH the crystallization point is not reached (Dougle et al., 1998). However, according to suggestions from the literature, only during months for which the

ambient RH values are less than RHD values the $NH_4NO_3$ is considered as a solid (Seinfeld and Pandis, 1998). In the present work, from March to October, the ambient RH was always lower than the RHD and therefore, within this period, $NH_4NO_3$ is assumed to exist to some extend in equilibrium with the solid phase. Within all other months, from the estimated RHD values, there is a small possibility such as $NH_4NO_3$ to exist also in equilibrium with the aqueous phase and deliquescent particles. However, most probably in Iasi, north-eastern Romania, solid $NH_4NO_3$ could form almost all over the year (in both

during the cold and the warm seasons) either due to the very complex chemical composition of the collected particles or due to abundant contribution of organic carbon to the particles mass concentration (Dougle et al., 1998). Formation of $NH_4NO_3$ over the warm seasons has been reported also by Ianniello et al. (2011) but under different conditions. For deliquesced particles there is suggestion that most of the fine particulate $NO_3^-$ can exists as an internal mixture with $SO_4^{2-}$, and also that $HNO_3$ can easily be absorbed into the droplets (Huang et al., 2010). In specific circumstances, the fine particulate $NO_3^-$ can



be formed from HNO$_3$ and NH$_3$ through heterogeneous reactions on fully neutralized fine particulate SO$_4^{2-}$, which is abundantly present in an urban area (Stockwell et al., 2000). However, in the present work, during the cold seasons, in the PM$_{2.5}$ fraction a statistically significant correlation between SO$_4^{2-}$ and NO$_3^-$ was observed (r = 0.91, p < 0.001, at the 99.9 % confidence level). High concentrations of NO$_3^-$ were found at high levels of RH and SO$_4^{2-}$ concentrations were high over the

entire range of RH values. Significant correlation was observed between NO$_3^-$ and RH (r = 0.84, p < 0.001, at the 99.9 % confidence level) but that between sulfate and RH was not statistically significant (r = 0.44, p = 0.177). These results can be interpreted as nitrate being produced on pre-existing sulfate aerosols, which could provide sufficient surface area and aerosol water content for the heterogeneous reactions to occur. Although, formation of fine particulate NO$_3^-$ was suggested to occur via reaction (R2), in the previously presented situations and especially at high RH values, the amounts of the gaseous

precursors, such as NH$_3$ and HNO$_3$, are expected to have relatively little influence on the fine particulate NO$_3^-$ formation (Markovic et al., 2011).

As for particulate NH$_4$Cl this is known as usually showing a volatility which is 2–3 times higher than that of NH$_4$NO$_3$ (HCl is more volatile than HNO$_3$) and at humidity lower than 75–85 % the particulate NH$_4$Cl exists in the solid phase in equilibrium with the gaseous products (Ianniello et al., 2011 and references therein). During the cold seasons, particulate

NH$_4^+$(total) showed a statistically significant correlation with particulate Cl$^-$ (Pearson coefficient of 0.73, p < 0.001, at the 99.9 % confidence level) but during the warm seasons the correlation was of very poor significance (meteorological parameters might be responsible on this). Moreover, only during the cold seasons significant correlations have been observed between fine particulate Cl$^-$ and SO$_4^{2-}$ (Pearson coefficient of 0.59, p < 0.001, at the 95.4 % confidence level) and between fine particulate Cl$^-$ and RH (Pearson coefficient of 0.71, p = 0.010, at the 95.4 % confidence level) and, usually, high

concentrations of fine particulate Cl$^-$ and SO$_4^{2-}$ were found at high levels of RH (35–83 %). Under these circumstances, in all above presented situations, the amount of the gaseous precursors are believed to present relatively little influence on the formation of fine particulate Cl$^-$ and, if formed, NH$_4$Cl most probably is generated from HCl and NH$_3$ through heterogeneous reactions on neutralized sulfate particles. However, in the present work, estimation of free Cl$^-$ allow us suggesting that no Cl$^-$ was available to be bond with other chemical species (i.e., NH$_4^+$) apart with the alkali or alkaline earth

metals, and therefore NH$_4$Cl in significant concentrations was not expected to be formed (especially over the warm seasons).

### 3.3 Relative ions contribution in size resolved aerosol particles from Iasi and potential influence of long range transport phenomena on particles size distribution

Figures 8a,b,c,d present, as monthly based averages, the relative contributions of identified and quantified water soluble ions to total detected components in fractions grouped in four stages, i.e., 0.0276–0.0945 μm size range (Fig. 8a), 0.155–0.612

μm size range (Fig. 8b), 0.946–2.39 μm size range (Fig. 8c) and 3.99–9.94 μm size range (Fig. 8d). From details presented in Fig. 8a, for the 0.0276–0.0945 μm size range fraction it is obvious that an important contribution within the total detected components is brought by organic anions identified in the formate, acetate and oxalate form. The presence of such important contribution of organics within this submicron size range may actually indicate a possible important role of organic acids in





secondary organic aerosols formation. Higher values over the warm seasons may suggest an enhancement in the role of biogenic emission sources. Important contributions are brought by particulate $SO_4^{2-}$, $NH_4^+$(total), $K^+$ and unexpected high $HCO_3^-$. However, high particulate $HCO_3^-$ is also evident for the 0.946–2.39 µm size range (Fig. 8c) and 3.99–9.94 µm size range (Fig. 8d) fractions. The 0.155–0.612 µm size range (Fig. 8b) fraction seems to be mainly constituted by $SO_4^{2-}$, $NO_3^-$

and $NH_4^+$(total) with very small contributions of all other particulate analyzed ions.

The seasonal variation observed mainly for $SO_4^{2-}$ and $NO_3^-$ might suggest an enhancement of photo-oxidative processes over the warm seasons. The 0.946–2.39 µm (Fig. 8c) and 3.99–9.94 µm size range (Fig. 8d) fractions seem to present mainly a non-homogeneous chemical composition and are dominated mainly by $HCO_3^-$, $NO_3^-$ and organics. However, over the investigated period, among all analyzed chemical species, in the $PM_{10}$ fraction $SO_4^{2-}$ is the most abundant with (26.0 ± 4.3)

% contribution, followed by $NO_3^-$ with (26.0 ± 10.7) %, $NH_4^+$(total) (15.0 ± 3.4) %, organics (including acetate, formate and oxalate) (12.2 ± 3.7) % and $HCO_3^-$ (10 ± 7.7) %. Similarly, in the $PM_{2.5}$ fraction $SO_4^{2-}$ is the most abundant with (28.9 ± 5.6) % and it is followed by $NO_3^-$ (19.6 ± 12.1) %, $NH_4^+$(total) (16.6 ± 3.2) %, organics (including acetate, formate and oxalate) (11.4 ± 4.0) % and $HCO_3^-$ (8.0 ± 6.8) %. In both the $PM_{10}$ and the $PM_{2.5}$ fractions, the largest contribution of $SO_4^{2-}$ was observed in June 2016 with (34.6 ± 10.9) % and, respectively, (40.8 ± 11.0) %. During the cold seasons, particulate $SO_4^{2-}$

and $NO_3^-$ contributions to the $PM_{10}$ fraction are of (23.6 ± 2.3) % and respectively, (28.6 ± 4.9) %, while during warm seasons these are of (28.5 ± 4.7) % and, respectively, (10.1 ± 4.8) %. In the $PM_{2.5}$ fraction, over cold seasons, these contributions are of (25.5 ± 2.9) % and respectively, (30.1 ± 5.8) %, while during warm seasons these are of (32.4 ± 5.6) % and, respectively, (9.2 ± 5.4) %. Wonaschutz et al. (2015) report for Vienna (Austria) $NO_3^-$ contributions of 31.3 % during winter and of 6.9 % during summer.

Particulate $Ca^{2+}$ and $HCO_3^-$, as dust tracers' ions, brought a significant contribution especially over the warm seasons (e.g., contribution of ~ 43.8 ± 11.2 % in $PM_{10}$ and ~ 37.8 ± 12.3 % in $PM_{2.5}$ brought by the two ions in August 2016), and such contributions would actually reflect that $Ca^{2+}$ and $HCO_3^-$ mostly originate from soil dust re-suspension during dry seasons. In 2016, in Iasi, north-eastern Romania, spring distinguished as the season with predominant air masses undertaking long range transport phenomenon from S-SE sector and the highest contributions from sea-spray aerosols tracers (i.e., $Na^+$ and $Mg^{2+}$,

Masiol et al., (2012)) were recorded in April (5.8 % for $Na^+$ and 0.7 % for $Mg^{2+}$). Such observations would actually allow us suggesting the presence of sea-spray aerosols from the Black Sea or other marine areas in the size resolved aerosols from Iasi. Although in overall $SO_4^{2-}$, $NO_3^-$ and $NH_4^+$(total) ions (as secondary pollution products) are the most abundant, at the investigated site, organics (including acetate, formate and oxalate) might also bring significant contributions ((17.3 ± 4.4) % in $PM_{10}$ fraction and 18.2 ± 5.1 % in $PM_{2.5}$ fraction in July 2016, much higher than that reported by Arsene et al., (2011) in

Iasi). The difference might reflect either an inversion of the photochemistry taking place at the investigated location or differences in the sampling efficiency between the two studies.

Figures 9a,b,c,d show the size distributions of seasonal averaged mass concentrations for $Cl^-$, $NO_3^-$, $SO_4^{2-}$, $NH_4^+$ (Fig. 9a,b) and $K^+$, $Na^+$, $Mg^{2+}$, $Ca^{2+}$ (Fig. 9c,d) ions in atmospheric aerosols from Iasi, both during the cold and, respectively, the warm seasons. While during the cold seasons $NO_3^-$, $SO_4^{2-}$, $NH_4^+$ and $K^+$ reside mainly in the fine mode with maxima at ~ 381 nm,




all other ions (i.e., Cl⁻, Na⁺, Mg²⁺, Ca²⁺) seem to present major contributions in the supermicrone mode (maxima between 1.6–2.39 µm). For Cl⁻ distribution over the cold seasons clear evidences were obtained about its existence in a bimodal mode. During the warm seasons only $SO_4^{2-}$ and $K^+$ present clear maxima at 381 nm while all other identified/quantified species have more important contributions in the supermicrone mode. For sulfate, larger modal diameter over the cold than

over the warm seasons is most likely due to hygroscopic growth under high RH, and/or due to increased secondary aerosol production, lower temperatures facilitating condensation. Secondary aerosol mass from aqueous-phase reactions may also play a role (Wonaschutz et al., 2015). The maxima observed in the coarse mode could also be explained considering that heterogeneous chemistry occurring on dust particles could also act as a source for some particulate species (Wang et al., 2012).

While particulate $NO_3^-$ over the cold seasons presented monomodal distributions in the submicron size range (maxima at 381 nm), over the warm seasons this ion presented a second mode with maxima in the 1.60 to 2.39 µm size range. Such a size distribution would allow us suggesting that $NO_3^-$ during the warm seasons is most probably produced by adsorption of $HNO_3$ on sea salt and soil particles (Park et al., 2004). According to Karydis et al. (2016), particulate $NO_3^-$ is not associated only with $NH_4^+$ in the fine mode. In particular light-metallic ions, as $Ca^{2+}$, $Mg^{2+}$, $Na^+$, and $K^+$ mainly present in the coarse

mode, can be associated with $NO_3^-$ and affect its partitioning into the aerosol phase. Dust effects on the distribution of particulate species might include decreasing of $NH_4^+$ fine-mode and shifting of particulate $NO_3^-$ from the fine- to the coarse-mode (Wang et al., 2012). In addition, the presence of significant fractions of particulate $NO_3^-$, Cl⁻, $Mg^{2+}$, $Ca^{2+}$ and $Na^+$ ions in the coarse fraction, might suggest that $NO_3^-$ possibly originates from the reactions of $HNO_3$ with $MgCO_3$, $CaCO_3$ or $NaCl$. Similar patterns were identified in Vienna, Austria, (Wonaschutz et al., 2015) and Prague, Czech (Schwarz et al., 2012).

Significant amounts of particulate $NO_3^-$ formed through the reaction between $HNO_3$ with $CaCO_3$ on soil-derived particles have also been observed and reported (Yao et al., 2003; Sharma et al., 2007). Moreover, sea-salt aerosols may also undergo chemical transformation of $NaCl$ to $NaNO_3$ during their transport (Schwarz et al., 2012).

The size distributions of particulate $K^+$ reflect the existence of one dominant fine mode, which most likely reflects contributions from biomass burning all over the year (Schmidl et al., 2008; Pachon et al., 2013). For both $Ca^{2+}$ and $Mg^{2+}$

ions, clear monomodal mass concentration distributions, with maxima in the 1.6 to 2.39 µm size range, have been observed over the investigated period. Over the warm seasons $Ca^{2+}$ seems to account for $(7.0 \pm 2.9)$ % of the $PM_{10}$ fraction $((5.5 \pm 2.9)$ % of the $PM_{2.5}$ fraction) while over the cold seasons it accounts for only $(3.0 \pm 0.6)$ % of the $PM_{10}$ fraction $((2.2 \pm 0.5)$ % of the $PM_{2.5}$ fraction) and these observations indicate that the impact from soil dust re-suspension could be more important during the warm (dried) seasons. Mineral dust may also explain the higher coarse fraction of $Mg^{2+}$ (mineral source of

$MgCO_3$).

Clear evidences have been obtained in the present work about the fact that air mass origin might greatly influence aerosol chemical composition at the investigated site. Wonaschutz et al. (2015) suggested that in Vienna, Austria, air mass origin is the most important factor for bulk PM concentrations, chemical composition of the coarse fraction ($> 1.5$ µm) and the mass size distribution, and less important for chemical composition of the fine fraction ($< 1.5$ µm). Although Iasi is located far



from the Mediterranean or Black Sea, over the warm seasons sea-salt chloride contribution to the aerosol budget in the area is not entirely excluded (Arsene et al., 2011). Moreover, dust particles originating from the Sahara are acknowledged as travelling across the tropical Atlantic Ocean (10–90 μg m$^{-3}$) and across the Mediterranean, affecting air quality in southern Europe (10–60 μg m$^{-3}$) (Karydis et al., 2016). In the present work, particulate Na$^+$ and Cl$^-$ ions, as tracers of sea-salt aerosols

(Tositti et al., 2014), were mainly observed in sampled events predominated by contributions of air masses arriving in Iasi from S-SE directions. However, one of the most interesting collected event was the one conducted 9$^{th}$ to 11$^{th}$ April in 2016. For this event the PM$_{10}$ fraction mass concentration was as high as 43.9 μg m$^{-3}$, a value which is about two times higher than the average of the total events. This event was actually highly influenced by air masses originating from both the Saharan desert and also from Mediterranean Sea. As shown in Fig. 10a the size distributions of particulate Na$^+$, Ca$^{2+}$, Mg$^{2+}$, Cl$^-$ ions

and mass concentrations present a highly dominating mode with maxima at 2.39 μm. For this event (Ca$^{2+}$, Mg$^{2+}$) and (Na$^+$, Cl$^-$) pairs showed statistically significant correlations (i.e., r = 0.94, p < 0.001 and, respectively, r = 0.85, p < 0.001, at the 99.9 % confidence level) suggesting common contributions from mineral Saharan dust and, respectively, from sea-salt marine aerosols. Moreover, Fig. 10a clearly shows that Na$^+$, Ca$^{2+}$, Mg$^{2+}$, Cl$^-$ mass concentrations make a very significant contribution to the total aerosol mass in the supermicrone mode with the maxima at 2.39 μm.

For April 2016 interesting behaviour was also observed for averaged mass size distributions of particulate NH$_4^+$, NO$_3^-$, SO$_4^{2-}$ and Mg$^{2+}$ ions and mass gravimetrically determined in samples collected at AMOS (Fig. 10b). During this month (highly affected by the atmospheric air masses buoyancy phenomenon, as shown by trajectories analysis for selected events collected in April), while particulate NH$_4^+$ and SO$_4^{2-}$ were mainly residing in the fine mode with clear maxima at 381 nm, NO$_3^-$ and Mg$^{2+}$ presented also a predominant mode in the fraction with maxima between 1.6–2.39 μm. Such distributions,

corroborated with meteorological conditions, would actually suggest possible heterogeneous formation route for SO$_4^{2-}$ (Wang et al., 2012), while for NO$_3^-$ adsorption of HNO$_3$ on mineral dust and sea salt particles (Karydis et al., 2016) would become more important.

## 4 Conclusions

The atmospheric concentrations of particulate species including acetate, (C$_2$H$_3$O$_2^-$), formate, (HCO$_2^-$), fluoride, (F$^-$),

chloride, (Cl$^-$), nitrite, (NO$_2^-$), nitrate, (NO$_3^-$), phosphate, (PO$_4^{3-}$), sulfate, (SO$_4^{2-}$), oxalate, (C$_2$O$_4^{2-}$), sodium, (Na$^+$), potassium, (K$^+$), ammonium, (NH$_4^+$), magnesium, (Mg$^{2+}$) and calcium (Ca$^{2+}$), have been measured in Iasi urban site, north-eastern Romania, over the 2016 year. The measurements were carried out by means of a cascade Dekati Low-Pressure Impactor (DLPI) performing aerosol size classification in 13 specific fractions evenly distributed over the 0.0276 up to 9.94 μm size range.

The entire data set was analyzed such as to investigate the seasonal variations in fine particulate species and meteorological effects, and to examine the contribution of local and regional sources to fine particulate species. ISORROPIA-II





thermodynamic model runs were used to estimate the pH of collected atmospheric particles since on the present data-base it has been proved that this was the best method to analyze particles acidity.

Within the aerosol mass concentration the ionic mass brings contribution as high as 40.6 % with the rest being unaccounted yet. Fine particulate $Cl^-$, $NO_3^-$, $NH_4^+$ and $K^+$ exhibited clear seasonal variations (with minima during the warm seasons),

mainly controlled by corroboration between factors as enhancement in the emission sources, changes in the mixed layer depth and specific meteorological conditions (e.g., higher RH values prevailing during cold seasons). Fine particulate $SO_4^{2-}$ did not show much variation with respect to seasons. Particulate $NH_4^+$ and $NO_3^-$ measured concentration in fine mode ($PM_{2.5}$) aerosols were found to be in reasonable good agreement with modelled values for the cold seasons but not for the warm seasons, an observation reflecting actually the susceptibility of $NH_4NO_3$ aerosols to be lost due to volatility.

Clear evidences have been obtained for the fact that in Iasi, north-eastern Romania, $NH_4^+$ in $PM_{2.5}$ was primarily associated with $SO_4^{2-}$ and $NO_3^-$. However, indirect ISORROPIA-II estimations showed that the atmosphere in Iasi, north-eastern Romania, might be ammonia-rich during both the cold and warm seasons, such as enough $NH_3$ to be present to neutralize $H_2SO_4$, $HNO_3$ and $HCl$ acidic components and to generate fine particulate ammonium salts, in the form of $(NH_4)_2SO_4$, $NH_4NO_3$ and $NH_4Cl$. Significant amounts of fine particulate $NO_3^-$ have been found mainly during the cold seasons such as to

promote $NH_4NO_3$ formation. The presence of eventual large amounts of $NH_3$, the domination of $(NH_4)_2SO_4$ over $NH_4NO_3$ and $NH_4Cl$, the high relative humidity conditions (highly prevailing over the cold seasons) dissolving probably a significant fraction of atmospheric $HNO_3$ and $NH_3$, are among the most important driving forces enhancing fine particulate $NO_3^-$ and $NH_4^+$ distribution in the atmosphere of Iasi, north-eastern Romania.

Most probably in Iasi, north-eastern Romania, gaseous $NH_3$ is not acting as the sole precursor of $NH_4^+$ formation, but it

exerts an important role on $NO_3^-$ and eventually $Cl^-$ formation in $PM_{2.5}$ via neutralization process. The chemical composition data-base in $PM_{2.5}$ (and $PM_{10}$), combined with predictions from the thermodynamic model ISORROPIA-II in the forward mode, metastable, allow us suggesting that at the investigated site, most probably $NH_3$ was present in sufficiently high concentration at most time such as to promote fine particle acidity neutralisation both during the cold and during the warm seasons. Although it is already known that running ISORROPIA-II in the forward mode, but with only aerosol

concentrations as input may result in a bias in predicted pH due to repartitioning of ammonia in the model, this approach was the single one helping to interpret the obtained results in a more trustful way.

Over the warm seasons ~ 35 % of the total analyzed samples presented pH values in the very strong acidity fraction (0–3 pH units range) while over the cold seasons the contribution in this pH range was of ~ 43 %. However, while during the warm seasons ~ 24–25 % of the acidic samples were with pH values in the 1–2 range (reflecting mainly contributions from very

strong inorganic acids), over the cold seasons an increase to ~ 40 %, brought by the 1–3 pH range, would reflect possible contributions from other acidic type species (i.e., organics), changes in aerosols acidity impacting actually the gas–particle partitioning of semi-volatile organic acids.





**Acknowledgements**

The authors acknowledge the financial support provided by UEFISCDI within the PN-III-P4-ID-PCE-2016-0299 (AI-FORECAST), PN-II-PCE-2011-3-0471 (EVOLUTION-AIR) and PN-II-RU-TE-2014-4-2461 (SOS-AROMATIC) projects. European Union's Horizon 2020 research and innovation programme through the EUROCHAMP-2020 Infrastructure Activity under grant agreement No 730997 is also gratefully acknowledged. The authors acknowledge also the NOAA Air Resources Laboratory (ARL) for the provision of the HYSPLIT transport and dispersion model and/or READY website (http://www.ready.noaa.gov) used in this publication.

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



Table 1: Basic statistics for the $PM_{10}$ and $PM_{2.5}$ fractions mass concentrations determined over the investigated period (n = 84 sampling events) in Iasi, north-eastern Romania.

| Statistical parameter | $PM_{10}$ (µg m$^{-3}$) | | | $PM_{2.5}$ (µg m$^{-3}$) | | |
| --- | --- | --- | --- | --- | --- | --- |
| | Working day | Weekend | Annual | Working day | Weekend | Annual |
| Mean | 19.25 | 18.60 | 18.95 | 17.31 | 16.47 | 16.92 |
| Median | 16.05 | 17.26 | 16.35 | 14.01 | 15.43 | 14.51 |
| Geomean | 16.90 | 16.91 | 16.90 | 14.86 | 14.83 | 14.84 |
| Stdev | 10.05 | 8.62 | 9.35 | 9.92 | 8.11 | 9.07 |
| Min | 5.56 | 7.11 | 5.56 | 5.08 | 6.30 | 5.08 |
| Max | 42.65 | 44.84 | 44.84 | 41.57 | 43.91 | 43.91 |



Table 2: Annual and/or seasonal arithmetic means of the $PM_{10}$ and $PM_{2.5}$ fraction mass concentrations in Iasi, north-eastern Romania, and other various European sites (mean ± stdev).

| Site | Category | Sampling aagl[*] (m) | Sampling period | $PM_{2.5}$ (µg m$^{-3}$) | $PM_{10}$ (µg m$^{-3}$) | Reference |
|---|---|---|---|---|---|---|
| Iasi (Romania) | urban | 35 | 2016 | 16.9±9.1<br>14.0±7.1 (warm)<br>21.3±13.0 (cold) | 18.9±9.3<br>16.8±8.3 (warm)<br>22.6±13.1 (cold) | This work |
| Iasi (Romania) | urban | 25 | 2007–2008 | 10.5±11.2 | 38.3±25.4 | Arsene et al., 2011[a] |
| Paris (France) | urban background | 20 | 2009–2010 | 14.8±9.6 | – | Bressi et al., 2013 |
| Northern Europe (SE12) | EMEP and 4 regional background sites | – | 2012–2013 | – | 3–8 | Alastuey et al., 2016[b] |
| North-western Europe (IE321) | | | | | 10–15 (S); ~ 35 (W) | |
| Central western Europe (FR09) | | | | | 10–15 (S); ~ 25 (W) | |
| Central Europe (DE44) | | | | | 20–25 (S); 25–30 (W) | |
| Eastern Europe (SK06,HU02) | | | | | 10–15 (S); 15–25 (W) | |
| Eastern Europe (MD13) | | | | | ~ 25 (S); 25–30 (W) | |
| South western Europe (ES22) | | | | | 20–25 (S); 5–10 (W) | |
| Central southern Europe (IT01) | | | | | 25–35 (S); 20–25 (W) | |
| South eastern Europe (GR02) | | | | | ~ 25 (S); 35–40 (W) | |
| Thessaloniki (Greece) | urban | 7 | 2011–2012 | 37.7±15.7 | – | Tolis et al., 2015 |
| Thessaloniki (Greece) | urban | 3 | 2011–2012 | 21.5±8.3 (warm)<br>33.9±19.3 (cold) | – | Voutsa et al., 2014[c] |
| Finokalia (Greece) | remote coastal | ~ 5 | 2004–2006 | 18.2 | 30.8 | Gerasopoulos et al., 2007 |
| Bologna (Italy) | urban background | courtyard | 2005–2006 | 31.6±21.0 | 44.5±24.2 | Tositti et al., 2014 |
| Venice (Italy) | semi-rural coastal | – | 2007–2008 | – | 22.5±12.9 | Masiol et al., 2012 |
| Prague (Czech) | urban | 12-25 | 2004–2005 | – | 33±13 | Schwarz et al., 2008 |

Note: [*]sampling altitude above ground level, [a]the total (coarse + fine) and the fine fractions reported (coarse fraction - particles of AED > 1.5 µm and fine fractions - particles of AED < 1.5 µm); [b]S–summer (8 June–12 July 2012), W–winter (11 January–8 February 2013); [c]averaged value of warm and cold seasons data (error propagation method for uncertainty estimation).



Table 3: Monthly averages of meteorological variables, $PM_{10}$, $PM_{2.5}$ fractions and of water soluble ions mass concentrations ($\mu g\ m^{-3}$) in $PM_{2.5}$ aerosol particles from Iasi, north-eastern Romania. Data are presented as mean±stdev (median).

| Month | January | February | March | April | May | June | July | August | September | October | November | December |
|---|---|---|---|---|---|---|---|---|---|---|---|---|
| WS (m/s) | 6.41±4.89 (5.60) | 7.34±4.74 (6.80) | 6.49±4.79 (5.70) | 6.41±5.05 (5.60) | 5.45±4.44 (4.90) | 6.07±4.36 (5.70) | 6.28±5.00 (5.40) | 5.37±4.72 (4.70) | 4.67±3.73 (4.40) | 7.17±4.57 (6.80) | 5.72±4.92 (4.90) | 3.77±3.76 (3.20) |
| AT (°C) | -1.06±4.71 (-1.42) | 5.86±1.98 (6.61) | 6.50±3.49 (5.35) | 13.65±4.15 (14.48) | 15.65±3.25 (15.70) | 21.05±4.75 (20.46) | 23.27±3.05 (24.58) | 21.86±2.13 (21.21) | 18.20±5.57 (20.30) | 10.03±5.11 (7.86) | 7.41±5.86 (6.23) | 2.77±2.41 (2.56) |
| RH (%) | 71.27±15.86 (69.57) | 68.20±6.97 (68.27) | 46.30±27.50 (37.88) | 44.43±13.19 (47.27) | 50.32±15.88 (47.47) | 46.63±16.69 (43.99) | 32.95±7.58 (33.97) | 37.12±13.14 (35.50) | 31.56±14.10 (28.71) | 54.30±10.58 (49.64) | 69.92±22.10 (72.41) | 81.85±13.22 (82.85) |
| RHD (%) | 78.66±3.61 (78.83) | 73.54±1.38 (73.01) | 73.14±2.34 (73.86) | 68.59±2.49 (68.02) | 67.37±1.92 (67.30) | 64.38±2.54 (64.62) | 63.16±1.62 (62.46) | 63.88±1.13 (64.21) | 65.98±3.16 (64.71) | 70.87±3.17 (72.17) | 72.61±3.81 (73.27) | 75.71±1.73 (75.85) |
| n* | 5 | 5 | 8 | 8 | 9 | 8 | 5 | 9 | 7 | 7 | 7 | 6 |
| $PM_{2.5}$ | 23.39±11.65 (26.19) | 21.30±8.37 (20.65) | 16.10±5.31 (14.98) | 15.29±8.34 (11.67) | 8.98±3.78 (6.88) | 11.45±5.61 (9.19) | 16.15±11.12 (14.19) | 14.00±4.27 (13.70) | 17.36±5.60 (17.77) | 12.71±4.81 (13.29) | 16.93±13.09 (12.04) | 30.94±9.51 (24.55) |
| $PM_{10}$ | 24.25±11.99 (27.86) | 22.11±8.50 (21.43) | 17.25±5.28 (16.08) | 18.22±11.08 (13.18) | 12.11±4.43 (14.25) | 14.33±6.98 (10.24) | 19.05±11.57 (16.00) | 16.59±5.52 (16.17) | 19.76±6.87 (20.20) | 15.13±5.81 (16.49) | 18.13±13.58 (13.49) | 32.08±9.44 (26.12) |
| $Cl^-$ | 0.32±0.15 (0.34) | 0.35±0.10 (0.37) | 0.14±0.10 (0.10) | 0.21±0.16 (0.15) | 0.14±0.08 (0.13) | 0.14±0.08 (0.14) | 0.20±0.11 (0.18) | 0.26±0.15 (0.31) | 0.19±0.06 (0.18) | 0.38±0.24 (0.32) | 0.24±0.24 (0.19) | 0.55±0.19 (0.64) |
| $NO_3^-$ | 3.54±1.93 (4.14) | 3.21±1.36 (3.14) | 2.42±1.09 (2.43) | 1.36±0.86 (1.22) | 0.47±0.28 (0.41) | 0.31±0.17 (0.26) | 0.31±0.11 (0.30) | 0.41±0.17 (0.33) | 0.69±0.41 (0.57) | 1.25±0.82 (1.13) | 1.88±1.27 (2.12) | 3.62±1.10 (4.07) |
| $SO_4^{2-}$ | 2.76±1.66 (2.49) | 2.58±0.91 (2.50) | 2.03±0.71 (2.15) | 1.96±0.75 (1.82) | 1.70±0.81 (1.53) | 2.39±1.31 (2.32) | 2.04±0.98 (1.58) | 2.22±0.76 (2.04) | 2.15±0.84 (2.18) | 1.96±1.00 (2.05) | 1.17±0.27 (1.16) | 2.16±0.69 (2.33) |
| $CH_3COO^-$ | 0.51±0.31 (0.45) | 0.79±0.50 (0.63) | 0.30±0.18 (0.29) | 0.84±0.36 (0.82) | 0.64±0.44 0.49 () | 0.58±0.34 (0.57) | 0.82±0.53 (0.48) | 0.69±0.71 (0.57) | 0.70±0.26 (0.60) | 0.80±0.28 (0.89) | 0.46±0.33 (0.43) | 0.59±0.12 (0.54) |
| $HCOO^-$ | 0.07±0.05 (0.06) | 0.06±0.03 (0.06) | 0.09±0.09 (0.05) | 0.15±0.20 (0.07) | 0.04±0.02 (0.05) | 0.05±0.03 (0.04) | 0.12±0.11 (0.07) | 0.08±0.02 (0.08) | 0.09±0.02 (0.10) | 0.09±0.04 (0.08) | 0.03±0.02 (0.04) | 0.07±0.02 (0.07) |
| $C_2O_4^{2-}$ | 0.08±0.06 (0.08) | 0.08±0.04 (0.08) | 0.08±0.04 (0.07) | 0.08±0.05 (0.08) | 0.06±0.04 (0.06) | 0.09±0.06 (0.08) | 0.12±0.09 (0.10) | 0.14±0.05 (0.15) | 0.10±0.08 (0.11) | 0.06±0.06 (0.04) | 0.02±0.02 (0.02) | 0.10±0.04 (0.09) |
| $HCO_3^-$ | 0.39±0.12 (0.43) | 0.48±0.24 (0.39) | 0.39±0.20 (0.36) | 0.65±0.67 (0.39) | 0.32±0.17 (0.38) | 0.64±0.56 (0.45) | 0.51±0.24 (0.56) | 2.23±1.95 (1.73) | 0.63±0.18 (0.60) | 0.25±0.17 (0.24) | 0.25±0.34 (0.10) | 0.35±0.25 (0.42) |
| $Na^+$ | 0.14±0.05 (0.13) | 0.17±0.08 (0.16) | 0.12±0.08 (0.10) | 0.41±0.31 (0.41) | 0.19±0.14 (0.10) | 0.13±0.05 (0.12) | 0.13±0.08 (0.12) | 0.18±0.09 (0.17) | 0.14±0.09 (0.11) | 0.25±0.16 (0.26) | 0.12±0.15 (0.08) | 0.17±0.09 (0.16) |
| $NH_4^+_{total}$ | 2.07±1.08 (2.39) | 1.94±0.70 (1.97) | 1.48±0.58 (1.57) | 0.93±0.25 (0.96) | 0.80±0.29 (0.67) | 0.90±0.36 (0.86) | 0.94±0.40 (0.78) | 0.84±0.16 (0.82) | 0.97±0.23 (0.92) | 1.10±0.48 (1.22) | 1.17±0.57 (1.25) | 2.09±0.49 (2.37) |
| $K^+$ | 0.54±0.25 (0.67) | 0.48±0.20 (0.44) | 0.27±0.12 (0.25) | 0.30±0.13 (0.27) | 0.26±0.16 (0.15) | 0.28±0.16 (0.24) | 0.37±0.21 (0.31) | 0.31±0.11 (0.32) | 0.45±0.14 (0.48) | 0.41±0.13 (0.41) | 0.43±0.32 (0.36) | 0.71±0.18 (0.65) |
| $Mg^{2+}$ | 0.03±0.01 (0.03) | 0.03±0.02 (0.03) | 0.02±0.02 (0.01) | 0.04±0.04 (0.02) | 0.01±0.00 (0.01) | 0.02±0.02 (0.01) | 0.02±0.00 (0.01) | 0.05±0.04 (0.03) | 0.02±0.01 (0.02) | 0.03±0.02 (0.03) | 0.01±0.01 (0.01) | 0.03±0.01 (0.03) |
| $Ca^{2+}$ | 0.21±0.07 (0.22) | 0.27±0.08 (0.26) | 0.22±0.07 (0.19) | 0.28±0.24 (0.21) | 0.15±0.05 (0.17) | 0.30±0.25 (0.22) | 0.25±0.12 (0.23) | 0.92±0.84 (0.65) | 0.30±0.08 (0.31) | 0.15±0.08 (0.15) | 0.16±0.17 (0.10) | 0.20±0.12 (0.20) |
| $pH_{total}$ | 4.20±2.32 (2.88) | 4.17±2.53 (3.27) | 3.94±2.81 (2.51) | 3.88±2.97 (1.80) | 3.95±3.10 (2.82) | 4.14±2.92 (2.27) | 4.58±2.87 (4.26) | 5.06±2.73 (7.14) | 4.24±3.19 (3.55) | 4.25±2.75 (3.60) | 3.86±2.55 (2.54) | 4.12±2.16 (2.87) |

Note: *n represent the number of aerosol sample events collected each month





Table 4: Correlation matrix (Pearson coefficients) for major ionic species (Cl⁻, NO₃⁻, SO₄²⁻, CH₃COO⁻, HCOO⁻, C₂O₄²⁻, HCO₃⁻, Na⁺, NH₄⁺(total), K⁺, Mg²⁺, Ca²⁺) in fine aerosol particles from Iasi, north-eastern Romania, both for cold (a) and warm (b) seasons.

(a)

| PM$_{2.5}$ (cold) | Cl⁻ | NO₃⁻ | SO₄²⁻ | CH₃COO⁻ | HCOO⁻ | C₂O₄²⁻ | HCO₃⁻ | Na⁺ | NH₄⁺(total) | K⁺ | Mg²⁺ | Ca²⁺ |
|---|---|---|---|---|---|---|---|---|---|---|---|---|
| Cl⁻ | 1.00 | **0.75** | 0.59 | 0.56 | 0.57 | **0.71** | 0.22 | **0.49** | **0.73** | **0.80** | 0.35 | 0.04 |
| NO₃⁻ | | 1.00 | **0.91** | 0.37 | **0.74** | **0.87** | 0.13 | 0.04 | **0.98** | **0.85** | 0.00 | 0.06 |
| SO₄²⁻ | | | 1.00 | 0.43 | **0.72** | **0.87** | 0.05 | 0.02 | **0.96** | **0.72** | 0.07 | 0.11 |
| CH₃COO⁻ | | | | 1.00 | 0.56 | **0.63** | 0.13 | 0.17 | 0.45 | 0.47 | 0.01 | 0.03 |
| HCOO⁻ | | | | | 1.00 | **0.85** | 0.04 | 0.01 | **0.76** | **0.65** | 0.12 | 0.09 |
| C₂O₄²⁻ | | | | | | 1.00 | 0.23 | 0.08 | **0.89** | **0.76** | 0.16 | 0.17 |
| HCO₃⁻ | | | | | | | 1.00 | 0.56 | 0.17 | 0.05 | **0.76** | **0.98** |
| Na⁺ | | | | | | | | 1.00 | 0.03 | 0.06 | **0.83** | 0.50 |
| NH₄⁺(total) | | | | | | | | | 1.00 | **0.82** | 0.08 | 0.12 |
| K⁺ | | | | | | | | | | 1.00 | 0.08 | 0.09 |
| Mg²⁺ | | | | | | | | | | | 1.00 | **0.66** |
| Ca²⁺ | | | | | | | | | | | | 1.00 |

(b)

| PM$_{2.5}$ (warm) | Cl⁻ | NO₃⁻ | SO₄²⁻ | CH₃COO⁻ | HCOO⁻ | C₂O₄²⁻ | HCO₃⁻ | Na⁺ | NH₄⁺(total) | K⁺ | Mg²⁺ | Ca²⁺ |
|---|---|---|---|---|---|---|---|---|---|---|---|---|
| Cl⁻ | 1.00 | 0.57 | 0.10 | **0.81** | 0.37 | 0.08 | **0.81** | **0.87** | 0.02 | **0.76** | **0.83** | **0.63** |
| NO₃⁻ | | 1.00 | 0.34 | 0.02 | 0.14 | 0.17 | **0.63** | **0.71** | 0.23 | 0.08 | **0.86** | 0.39 |
| SO₄²⁻ | | | 1.00 | 0.01 | **0.66** | **0.76** | 0.17 | 0.05 | **0.97** | 0.43 | 0.03 | 0.01 |
| CH₃COO⁻ | | | | 1.00 | 0.32 | 0.11 | 0.30 | 0.07 | 0.11 | 0.55 | 0.05 | 0.08 |
| HCOO⁻ | | | | | 1.00 | 0.58 | 0.32 | 0.01 | **0.66** | 0.59 | 0.02 | 0.01 |
| C₂O₄²⁻ | | | | | | 1.00 | 0.58 | 0.04 | **0.71** | 0.46 | 0.01 | 0.15 |
| HCO₃⁻ | | | | | | | 1.00 | 0.45 | 0.06 | 0.13 | 0.12 | **0.99** |
| Na⁺ | | | | | | | | 1.00 | 0.11 | 0.19 | **0.76** | 0.38 |
| NH₄⁺(total) | | | | | | | | | 1.00 | 0.49 | 0.12 | 0.12 |
| K⁺ | | | | | | | | | | 1.00 | 0.02 | 0.03 |
| Mg²⁺ | | | | | | | | | | | 1.00 | **0.81** |
| Ca²⁺ | | | | | | | | | | | | 1.00 |



**Figure captions**

**Figure 1: Sectors contributions identified from classification of 2-day back trajectories of air masses ending at Iasi and representative backward trajectories of long and short range transport, Black Sea influence and African dust (shown trajectories correspond to sampling events).**

**Figure 2: Patterns of the monthly arithmetic mean concentrations and standard deviations in the $PM_{10}$, $PM_{2.5}$ and $PM_{2.5}/PM_{10}$ variables at Iasi, north-eastern Romania.**

**Figure 3: Size distribution histograms of aerosol particles mass concentration gravimetrically determined over both the cold and warm seasons.**

**Figure 4: Distribution of the aerosol pH predicted by ISOROPIA-II (forward mode) vs. the ion balance (a), sensitivity of aerosol pH predicted with the model to small changes in the input aerosol $NH_4^+$ concentration (b) and bar chart distribution in aerosol pH over the warm and cold season both for $NH_4^+$ derived from raw IC data (c) and $NH_4^+$(total) (d).**

**Figure 5: Size distribution of averaged aerosol mass, $NO_3^-$ $SO_4^{2-}$ and $NH_4^+$ concentrations over cold (a) and warm (b) seasons accompanied by the size distribution of ISORROPIA-II estimated pH and $H^+$ mass concentration over both the cold (c) and the warm (d) seasons.**

**Figure 6: Seasonal variations for selected water-soluble ionic components in the $PM_{2.5}$ fraction (a-h) and variation of the mixed layer depth at the investigated site (i). The inset distribution presented within $NO_3^-$ seasonal variation reflects the contribution of the coarse fraction over the warm seasons. The horizontal black line represents the mean, the horizontal colored line – the median, the box – the 25–75% percentiles, the length of the whiskers plot – the 10 and 90% of observed concentrations, circles – outliers).**

**Figure 7: Regression analysis of the $[NH_4^+]$ vs. $(2\times[SO_4^{2-}])$ (a, b), $[NH_4^+]$ vs. $([NO_3^-] + 2\times[SO_4^{2-}])$ (c,d) and $[NH_4^+]$ vs. $([Cl^-] + [NO_3^-] + 2\times[SO_4^{2-}])$ (e,f) dependences specific for $PM_{2.5}$ particles.**

**Figure 8: The relative contributions, as monthly-based averages, of identified and quantified water soluble ions to total detected components in the 0.0276–0.0945 µm (a), 0.155–0.612 µm (b), 0.946–2.39 µm (c) and 3.99–9.94 µm (d) size range grouped fractions.**

**Figure 9: Size distributions of seasonal averaged mass concentrations for $Cl^-$, $NO_3^-$, $SO_4^{2-}$, $NH_4^+$ (a,b) and $K^+$, $Na^+$, $Mg^{2+}$, $Ca^{2+}$ (c,d) ions in atmospheric aerosols from Iasi, both during the cold and, respectively, the warm seasons.**

**Figure 10: Evidences of long range transport contributions from Saharan dust within the size distribution of particulate $Na^+$, $Ca^{2+}$, $Mg^{2+}$, $Cl^-$ ions and aerosol mass (a) and of air masses buoyancy phenomena within the size distribution of particulate $NH_4^+$, $NO_3^-$, $SO_4^{2-}$, $Mg^{2+}$ ions and aerosol mass (b).**





**Figure 1**

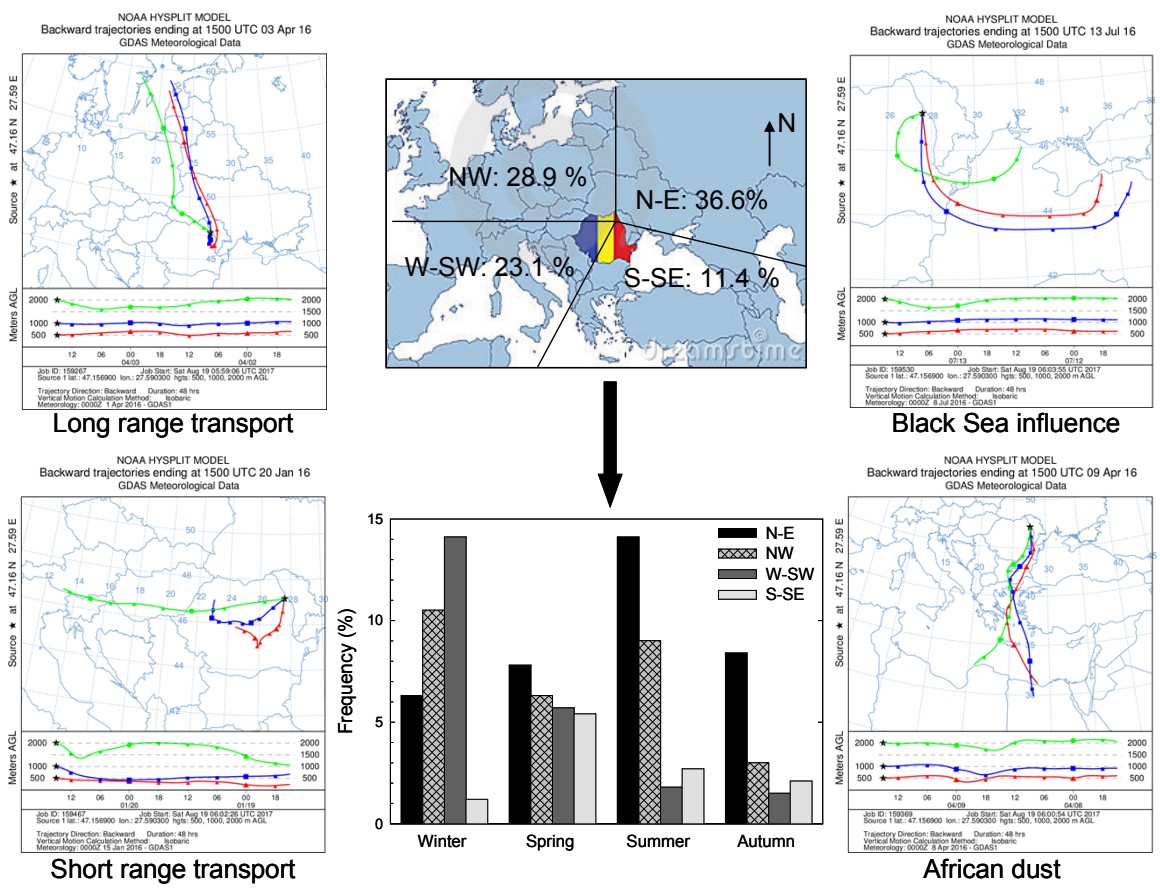

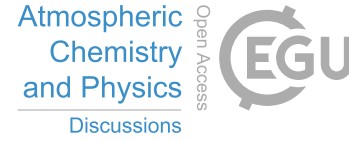

**Figure 2**

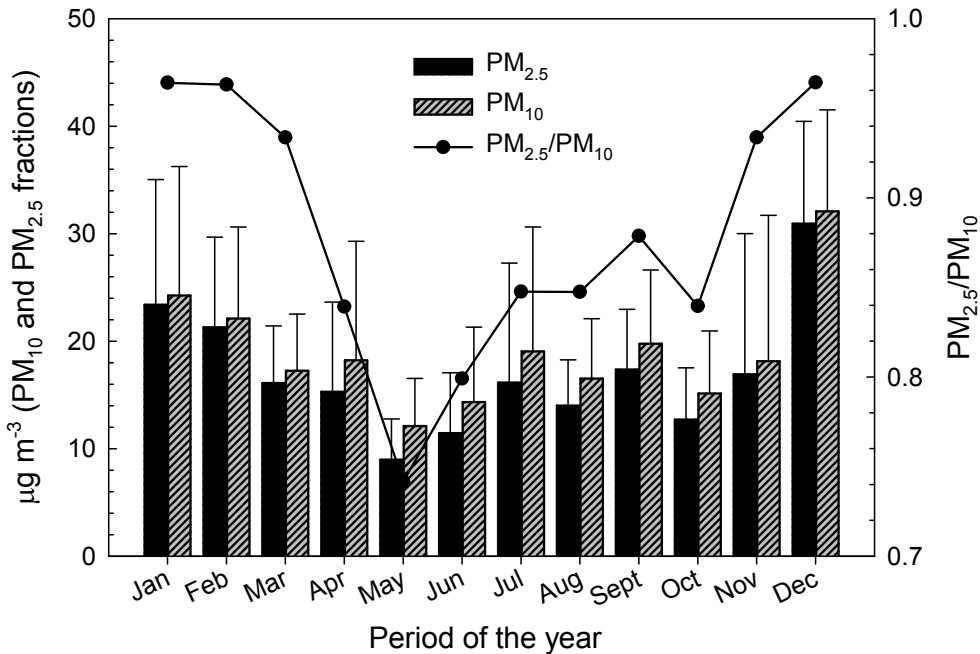





**Figure 3**

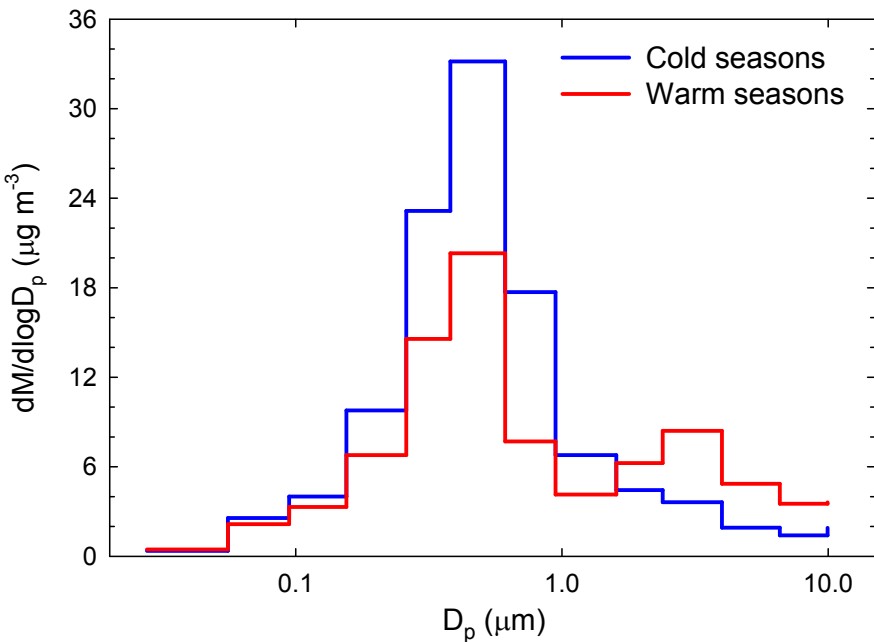





**Figure 4**

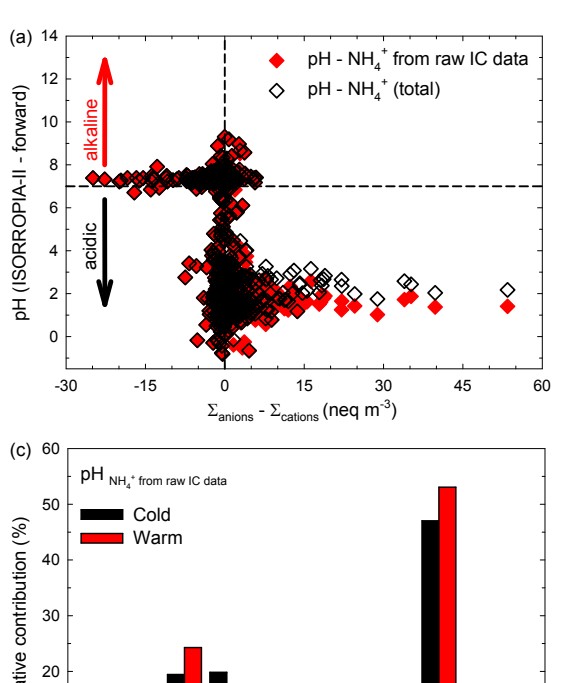

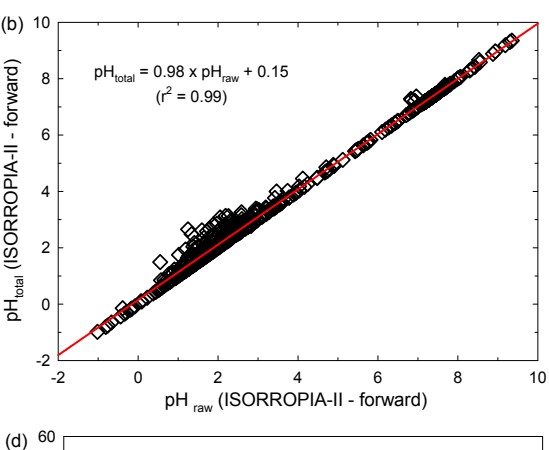

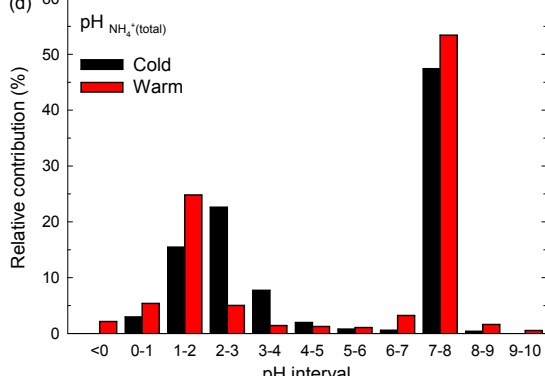



**Figure 5**

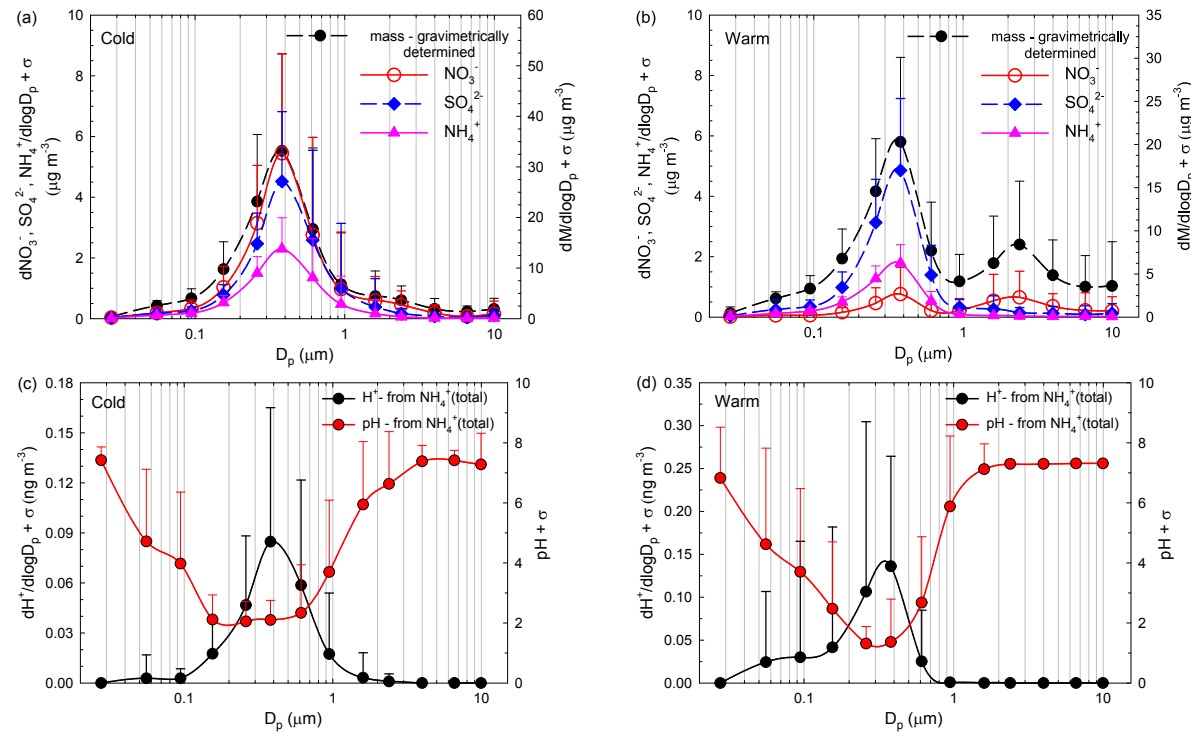





**Figure 6**





**Figure 7**







**Figure 8**

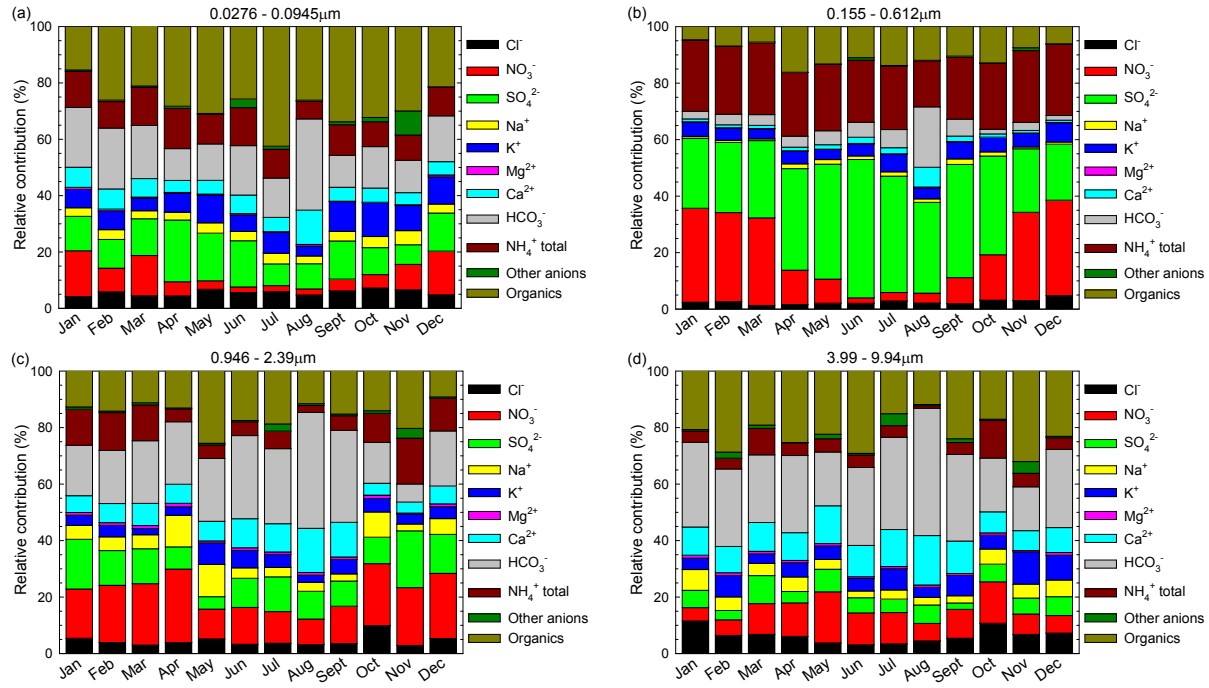





**Figure 9**

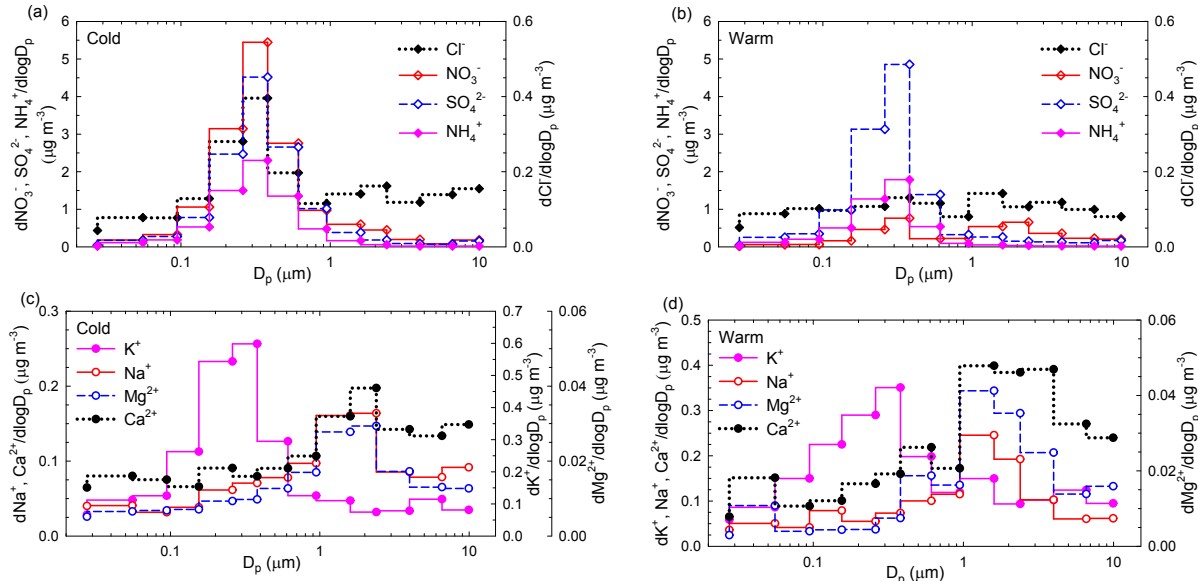





**Figure 10**

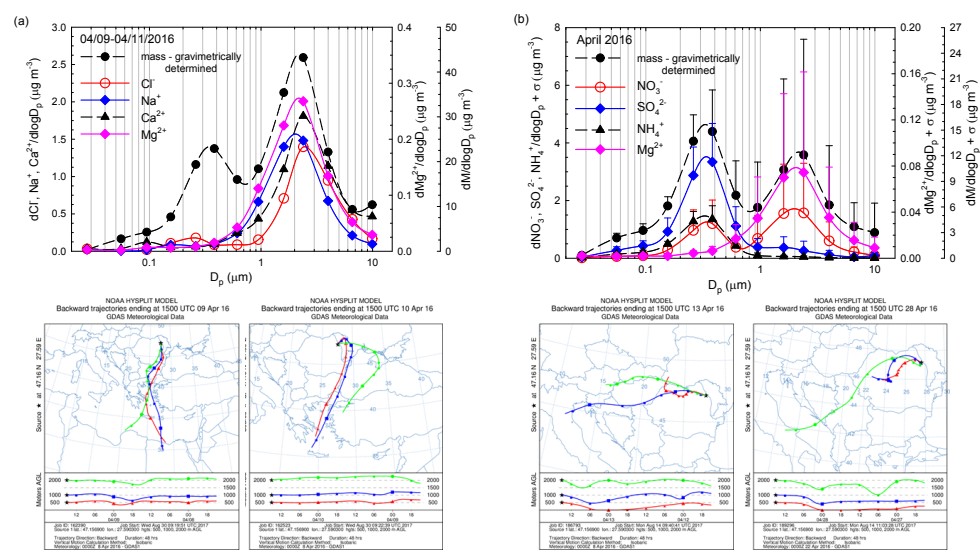