# Peer review of "Chemical characteristics of size resolved atmospheric aerosols in Iasi, north-eastern Romania: Nitrogen-containing inorganic compounds control aerosols chemistry in the area"

_Atmospheric Chemistry and Physics, 2017_

## Referee Comment (RC1) · Anonymous Referee #1 · 18 Dec 2017

This study analyzed the ionic components of size-resolved atmospheric aerosols sampled in northeastern Romania including ion balance, aerosol acidity, formation of ammonium and nitrate, and influence of relative humidity and air mass origins. Measurements of aerosol chemical composition in Eastern Europe are scarce; however, similar measurements at this particular site have been previously reported. The paper is long and I feel the discussions in the paper need to be written more concisely, e.g. especially the discussion of the formation of ammonium and nitrate. The tendency for ammonium

nitrate to form at lower temperatures and higher relative humidity and the strong as-
sociation between ammonium, sulfate and nitrate has been previously reported. The
presence of strongly acidic aerosols is an interesting and potentially important result
that could use more discussion on the acid rain implications. This paper is strongly
focused on the chemical and physical processes driving the formation of secondary
pollutants, but more consideration needs to be given to the gaseous precursors and
where these emissions are coming from.

Specific Comments

P4, lines 31-34: Chemical composition and seasonal variation of water soluble ions at
this site has been previously reported. Could you emphasize the new work that will be
presented in this paper?

P8, lines 16-17: For long range transport, which region is contributing to PM? Can you
specify what the local contributions might be at this site, e.g. commercial/industrial or
vehicular emissions contributing to PM?

P8, lines 22-32: Could the differences in PM concentrations between this study and the
previous study be due to the different particle size cutoff? The previous study reported
PM1.5 and PM(>1.5), whereas this study reported PM2.5 and PM10. Could it also
be due to sampling in different years in which there may be changes in the emission
sources (e.g. emissions reductions)? It seems the difference in altitude of the sampling
site (25m vs. 35m) is too small to account for the difference in PM concentrations. What
was the altitude of the sampling site in Alastuey et al. (2016)?

P9, lines 22-24: "increased frequency of dust and also due to more intense anthro-
pogenic activities in the neighbourhood of the sampling site" This result seems incon-
sistent with earlier results, which stated that PM was likely from long range transport
instead of local contributions.

P9, lines 24-26: Why would dry deposition during the warm seasons be enhanced for

PM2.5 but not PM10?

P9, lines 30-32: I'm not sure how you arrived at the conclusion that the results are not due to emission sources but are likely due to meteorology based on the seasonal trends in the PM size distribution. Please explain how changes in the meteorological conditions and which met parameters affect the seasonal PM size distribution.

P10, lines 6-9 and other places: I suggest moving the detailed statistics in parentheses to a table or the supplement because it makes the rest of the text difficult to follow.

P10, lines 22-28: It's not clear what the missing $NH4+$ is referring to because $NH4+$ measurements are available. Please briefly explain why there is another fraction of $NH4+$ that is not measured by the IC data.

P11, lines 4-23: This text could be condensed or moved to the methods section or the supplement. This section should present the results of the model runs.

P12, lines 20-23: What are the ecosystem or human health implications of strongly acidic aerosols? I suggest providing a brief explanation on the importance of aerosol acidity. What is contributing to the strongly acidic aerosols, pH 0-3? At this aerosol pH, does it contribute significantly to acid rain in this region? Are there any reports of acid rain impacts in this region?

P13 line 29 – P15 line 9: This section examined $NH3/NH4+$ partitioning as a function of relative humidity. Considering that ammonium tends to be associated with sulfate and nitrate, I think that the discussion on $NH3/NH4+$ partitioning needs to consider changes in the sulfate and nitrate concentrations as well as temperature which affect both sulfate and nitrate production. Also, temperature and relative humidity are strongly correlated. Can you confirm that the relationship with relative humidity is not actually due to temperature? Do you still see the strong relationship between $NH3/NH4+$ and relative humidity when you analyze cold and warm seasons separately?

P14, lines 29-31: I think that this conclusion needs to be corroborated by examining

the seasonal trend in the ammonia emissions since ammonia concentrations were not available. Ammonia is typically higher during the warm seasons, but here you suggest it is high enough all year round.

P15, lines 13-16: Are these results expected? For the ions associated with fine particles, are the percentage increases lower than those ions associated with coarse particles?

P15, lines 18-21: Sulfate is usually higher during warm seasons because of increased oxidation of SO2. I'm not sure about the explanation of lower mixing heights or temperature inversions because not all of the pollutants were higher during cold seasons.

P17, line 5: Ammonia is usually higher during warm seasons because of higher temperatures and increased agricultural emissions. What are the sources of ammonia during cold seasons?

P17, lines 16-19: The relative humidity explanation for the high sulfate concentrations is not consistent with the lack of correlation between sulfate and meteorological parameters mentioned earlier (lines 12-13).

P22, lines 13-20: The molar ratios in Fig. 7cdef between cold and warm seasons are not very different and all the R2 values were close to 1. It seems that a complete neutralization of H2SO4 and HNO3 also occurred in the warm seasons. Can you explain the lack of difference in the molar ratios?

P27-28: This section can be improved by showing the inorganic ion concentrations associated with the different air mass origins identified in Fig. 1. While physical and chemical processes are important in the formation of secondary pollutants, it is also important to examine where the gaseous precursors e.g. SO2, NOx and NH3 are coming from. There might be a change in the air masses during different times of the year leading to seasonal variability in the gaseous precursors, which in turn affects the secondary pollutants.

---

## Referee Comment (RC2) · Anonymous Referee #2 · 16 Jan 2018

General This is a comprehensive field and analytical study of aerosol particles sampled under size-resolution in north-eastern Romania. It contains a wealth of data on the aerosol chemical composition in this area where not many data are available for comparison which is generally true for Eastern Europe.

The whole manuscript is a bit lengthy and it might be good if the authors could explore potential for shortening some sections of the paper.

[Figure]

Overall, I think the manuscript requires only minor modification and can then be published in ACP.

Details Page 3. line 3: '... great concern of interest....' - Maybe better '...great concern and of interest....' Please check the English throughout the manuscript again, possibly but a native speaker or an English language editing service.

Section 2.2: Can anything be said how reliable the sampling of the very small size fractions by the DLPI is ? Has this been characterised ?

Table 2: Maybe the names of the sampling locations from the EMEP measurments can be given.

P9, l13: ...Figure... This should be capitalized all over the manuscript.

P15,l21: Is the comparison to the Kanpur measurment helpful here ? I cannot fully understand what CaCO3 has to do with nitrate.

P27,l23: But shouldn't biomass burning have a seasonal pattern ?

P28, l30: Please check sentence: '... such as to....'

---

## Author Comment (AC1) · 12 Feb 2018

Response to reviewers
Manuscript: ACP-2017-1030
Manuscript title: Chemical characteristics of size resolved atmospheric aerosols in Iasi, north-eastern Romania. Nitrogen-containing inorganic compounds controlling aerosols chemistry in the area

The discussion below includes the complete text from the reviewer (italic), along with our responses to the specific comments and the corresponding changes made to the revised manuscript. All line numbers refer to the original manuscript.

Response to **Reviewer #1 Comments**

*This study analyzed the ionic components of size-resolved atmospheric aerosols sampled in northeastern Romania including ion balance, aerosol acidity, formation of ammonium and nitrate, and influence of relative humidity and air mass origins. Measurements of aerosol chemical composition in Eastern Europe are scarce; however, similar measurements at this particular site have been previously reported. The paper is long and I feel the discussions in the paper need to be written more concisely, e.g. especially the discussion of the formation of ammonium and nitrate. The tendency for ammonium nitrate to form at lower temperatures and higher relative humidity and the strong association between ammonium, sulfate and nitrate has been previously reported. The presence of strongly acidic aerosols is an interesting and potentially important result that could use more discussion on the acid rain implications. This paper is strongly focused on the chemical and physical processes driving the formation of secondary pollutants, but more consideration needs to be given to the gaseous precursors and where these emissions are coming from.*

We are very grateful for the reviewer's comments and suggestions and we are highly appreciating the invested time in carefully reviewing the manuscript. Reviewer's concerns and suggestions are addressed here below and we would like also to point out the following:
1) A Supplement Material is accompanying now the manuscript (attached to the present response).
2) The manuscript is now presented in a more concise manner (reduction from 53 pages to 46 pages) (the revised manuscript is attached to the present response).
3) In the revised manuscript amendments related to reviewer #1 concern are highlighted in red colour.
4) The revised manuscript contains 4 additional references.
5) Considering language-related suggestion from one of the reviewers, we have requested a well known English speaking scientist, Prof. Davide Vione from University of Torino, to offer us support in polishing the composition. Language polishing has resulted in a considerably improvement of the manuscript quality.
6) In the revised manuscript all language-related changes are highlighted in blue colour. Blue colour reflects: *i*) corrected grammar errors, *ii*) the place where cuts occurred within former sentences, *iii*) extra-words used in the revised manuscript, *iv*) full sentences addressing more succinctly ideas from the original manuscript.
7) We kindly ask the editorial office to accept our proposal colour-related since working in the track-changes mode significantly affects the readability of the manuscript.

In the revised manuscript the title is proposed in the form
Chemical characteristics of size resolved atmospheric aerosols in Iasi, north-eastern Romania: Nitrogen-containing inorganic compounds control aerosols chemistry in the area

In the Introduction section, P3, lines 26-28 "Sulfate ($SO_4^{2-}$), nitrate ($NO_3^-$) and ammonium ($NH_4^+$) ions are often assigned as significantly contributing inorganic species to the total aerosol mass (Wang et al., 2005; Bressi et al., 2013; Hasheminassab et al., 2014; Voutsa et al., 2014)." we have replaced with the sentence "Sulfate ($SO_4^{2-}$), nitrate ($NO_3^-$) and ammonium ($NH_4^+$) ions, representing major inorganic particle constituents (Wang et al., 2005; Bressi et al., 2013; Hasheminassab et al., 2014; Voutsa et al., 2014), are mainly secondary in nature species formed in the atmosphere by chemical reactions of their precursor gases (photochemical gas-phase, aqueous-phase oxidation, particulate-phase processes) and physical processes (nucleation, condensation, etc.) (Aksoyoglu et al., 2017)."

**Specific Comments**
*Comment (1) P4, lines 31-34: Chemical composition and seasonal variation of water soluble ions at this site has been previously reported. Could you emphasize the new work that will be presented in this paper?*

In the present paper, apart the already emphasized aspects (meant mainly to underline that the work is providing, for the first time to our knowledge, information related to the chemical composition of clearly discriminated size segregated aerosol particles in Iasi urban area, north-eastern Romania) we would like to complete the contribution brought by the present work with the following details:
"As a first attempt to assess particles acidity in the area, the present work highlights the existence of significant aerosol fractions characterized by pH values in the very strong acidity range (0-3 pH units), with potentially important implications on acid rain. Moreover, the potential importance of gaseous precursors (e.g., $NH_3$, $HNO_3$, $HCl$) on secondary inorganic PM is also discussed."

*Comment (2) P8, lines 16-17: For long range transport, which region is contributing to PM? Can you specify what the local contributions might be at this site, e.g. commercial/industrial or vehicular emissions contributing to PM?*

With the sentence "Since not statistically significant differences were observed, the result allowed us suggesting that AMOS site is mainly influenced by long range transport rather than local contributions." the authors wanted mainly to underline that the altitude of the sampling location helped avoiding a potential overwhelming contribution brought by significant sources existent at the ground level in the interest area.

The "local contribution" term was mainly referring to the ground level hot-spot sources including here vehicles transport, sporadic excavation, etc. Although over the last years significant improvement has been observed in Iasi's road-map infrastructure dusty environment seems still to prevail. Although such situations are clearly evident both over the cold and warm seasons, attempts are given by the local administrative sector in settling down soil- or road-resuspended dust especially over the warm seasons by wetting the roads.

The data set from the present work has been investigated through Radar and/or Pie charts in order to identify the potential contributions brought by various sectors long range transport to the atmospheric PM burden in the area (details are given within sections S 9 and S 10 in the supplement material, SM). It should be however emphasized that the PM chemical composition appeared to be mainly driven by the air mass origin and type (e.g., continental, regional), meteorologically-controlled air mass characteristics (buoyancies), geographical context (the Oriental Carpathians chain, with the highest altitude of 1907 m, is facing Iasi ~150 km north-westerly distance), etc. More clearly, in terms of sectors contributions the N-NE up to the S (clockwise round) area seems to bring the most important fraction to both $PM_{2.5}$ and $PM_{10}$ abundances. Higher loaded events in terms of aerosols mass concentration were those associated with long-range continental transport,

Saharan dust-events, regional transport from more arid areas (S-SE) and buoyancy affected air masses.

Regarding potential local anthropogenic contribution to PM atmospheric burden it should be emphasized that in the immediately nearby area neither important commercial areas nor industrial units are operating and traffic should be the most important contributor. Actually in Iasi, in the post-communist period, the industry has almost completely ceased down and presently the Antibiotic company (7 km from the sampling point, western direction), a small brick plant (8 km, SE) and the heat and power plant (5 km, SE) may account as for the most significant local anthropogenic contributors. However, Iasi is known as an important Romanian university centre (about 42000 students) with intense activity over university semesters. Traffic in the nearby area and commercial activity in the top hot-spots area of Iasi get more intense over these periods of the year. As for the rural areas surrounding Iasi, or from the nearby counties, farming activities are mainly related to cereal crops, plantation agriculture, open and closed animal barns (cattle, sheep and goat, pigs, poultry).

In this context, as in Galon et al. manuscript the sentence/paragraph related to the referee's comment is not enough clear in the revised form of the paper we propose "Moreover, the difference in the mean values of the two groups was not high enough to exclude random sampling variability, thereby suggesting that the differences are not statistically significant. Therefore, local anthropogenic activities seem to bring similar contribution to the aerosol burden in the area on both working days and weekends."

*Comment (3) P8, lines 22-32: Could the differences in PM concentrations between this study and the previous study be due to the different particle size cutoff? The previous study reported PM1.5 and PM(>1.5), whereas this study reported PM$_{2.5}$ and PM$_{10}$. Could it also be due to sampling in different years in which there may be changes in the emission sources (e.g. emissions reductions)? It seems the difference in altitude of the sampling site (25m vs. 35m) is too small to account for the difference in PM concentrations. What was the altitude of the sampling site in Alastuey et al. (2016)?*

We are very grateful to the referee comments which appear to be correct.

In the manuscript we propose

"Table 2 presents the annual and/or seasonal arithmetic means of PM$_{10}$ and PM$_{2.5}$ mass concentrations in Iasi, compared to other European sites (mean ± stdev). The annual averages obtained in the present work show differences in comparison with those reported by Arsene et al. (2011) for the same site. Arsene et al. (2011) have used a SFU system consisting of a 8.0 µm pore size, 47-mm diameter Isopore polycarbonate filter mounted in front of a 0.4 µm pore size, 47-mm diameter Isopore filter. However, the values determined in the present work for the fractions PM$_{0.027-1.6}$ (15.6 ± 8.7 µg m$^{-3}$) and PM$_{0.381-1.6}$ (9.1 ± 5.6 µg m$^{-3}$) are much closer to those reported by Arsene et al. (2011) for PM$_{1.5}$ (10.5 ± 11.2 µg m$^{-3}$). The potential influence on the PM levels of particle size cut-off, differences in sampling site altitude, occurrence of precipitation events, long-range transport phenomena, and air masses buoyancy is presented in detail in Section S 1 of the SM."

In the Supplement Material we propose Section S 1 in the form:
"Possible explanations for the differences/similarities between afore mentioned studies might be seen as a result of the contributions brought from various factors. As shown below the differences in the PM mass concentrations reported in Galon et al. and those reported in Arsene et al. (2011) might be only slightly controlled by changes in the particle size cut-off and/or by the difference in sampling site altitude (25 m in Galon et al. vs. 35 m in Arsene et al. (2011)) which is too small to account for the difference in PM concentrations.

In Arsene et al. (2011) study, aerosol sampling has been undertaken by using a stacked filter unit (SFUs) consisting of a 8.0 μm pore size 47-mm diameter Isopore polycarbonate filter mounted in front of a 0.4 μm pore size 47-mm diameter Isopore filter. According to details in Arsene et al. (2011) and references included therein, the 50% cut-point diameter (D50) of the 8.0 μm pore size filter was estimated to be of the order on 1.5 ± 0.2 μm aerodynamic equivalent diameter (AED). Consequently, particles collected on the 8 μm pore size filters, with a diameter larger than 1.5 μm AED, were referring to the aerosol coarse fraction, while the particles collected on the 0.4 μm pore size filters were attributed to particles with a diameter below 1.5 μm AED. In Galon et al. work aerosols samples were collected on 25 mm in diameter ungreased aluminum filters using a cascade Dekati Low-Pressure Impactor (DLPI), operating at a flow rate of 29.85 L min$^{-1}$. Enhanced sampling efficiency of the DLPI unit used in Galon et al. work, with regard to fine and ultrafine particles, has been observed in comparison with the SFU unit reported by Arsene et al. (2011). Parallel DLPI and SFU sampling runs (performed within January–July 2016) showed that the DLPI unit could collect in average with ~ 6 μg m$^{-3}$ more particles than the SFU system and this observation allowed us suggesting that most probably the operational limits of the SFU system with regard to fine and ultrafine particles could suppress to some extend the values reported by Arsene et al. (2011).

The difference observed between the PM$_{10}$ mass concentration, i.e. 18.9 ± 9.3 μg m$^{-3}$, reported by Galon et al. and the PM$_{total}$ mass concentration, i.e. 38.3 ± 25.4 μg m$^{-3}$, reported by Arsene et al. (2011) might actually be the result of sampling in different years with emission sources of various prevalence. Implementation of environmental quality management systems in various sectors with anthropogenic activity may account for that (e.g. attempts given by the local administrative sector in settling down soil- or road-resuspended dust especially over the warm seasons by wetting the roads, protecting areas with intense building activities, etc.).

Regarding other literature available values, in a study from 2016, Alastuey et al. (2016) report for PM$_{10}$ concentrations in Moldova (the closest point to our sampling site), values as high as ~ 25 μg m$^{-3}$ over summer and of ~ 25–30 μg m$^{-3}$ over winter period, yielding an annual averaged value of ~ 27.5 μg m$^{-3}$ which is much higher than the 18.9 μg m$^{-3}$ value reported in the present work, for Iasi, north-eastern Romania. However, the elevated concentrations observed at the eastern sites were attributed to regional or local sources (Alastuey et al., 2016). In Alastuey et al. (2016) the author reports only the altitude above sea level (i.e. 156 m for Moldova's location) and the sampling a.g.l. is not mentioned. It might be that in Alastuey et al. (2016) study if the Moldavian sampling location was assigned as an EMEP site than the sampling a.g.l. was settled in agreement with an EC Directive recommending an a.g.l. of ~ 4 or 8 m. It is believed that the difference observed between Galon et al. site (i.e. 18.9 ± 9.3 μg m$^{-3}$) in comparison with other European sites might actually reflect that sampling altitude would be an important controlling factor to the atmospheric aerosol burden at a site.

In Iasi, north-eastern Romania, mass concentrations in the 5 to 10 μg m$^{-3}$ range, for the PM$_{10}$ and PM$_{2.5}$ fractions were especially observed in samples collected after raining events (i.e., May, June, August, and October). Such behaviour would be expected since particles from the atmosphere might be efficiently removed by precipitation (Arsene et al., 2011). However, during events with strong natural or anthropogenic contributions the PM$_{10}$ and PM$_{2.5}$ mass concentrations exceeded the averages observed at AMOS. For example, an event collected in April 2016, from 9$^{th}$ to 11$^{th}$, with 34.3 μg m$^{-3}$ in PM$_{2.5}$ and 43.9 μg m$^{-3}$ in PM$_{10}$, was actually highly influenced by the long range transport phenomena of African dust and marine aerosols from the Black and Aegean/Mediterranean seas (i.e., event described in detail later in the text). For other events, the exceeding fine fraction mass concentrations are believed to be a result of more variable sources (combustion, biogenic, local mineral dust, meteorological factors, etc.)."

*Comment (4) P9, lines 22-24: "increased frequency of dust and also due to more intense anthropogenic activities in the neighbourhood of the sampling site" This result seems inconsistent*

*with earlier results, which stated that PM was likely from long range transport instead of local contributions.*

We think that this problem is already answered by the revision proposed under reviewer #1, comment 2.

"Moreover, the difference in the mean values of the two groups was not high enough to exclude random sampling variability, thereby suggesting that the differences are not statistically significant. Therefore, local anthropogenic activities seem to bring similar contribution to the aerosol burden in the area on both working days and weekends."

However, taking into account that sampling has been undertaken over a one year period, the problem is really complex if discussed in terms of air masses origin/behavior, geographical location of the sampling point, per event chemical composition, etc. Details given in FIG. 2 highlight that over the warm seasons the contribution of the coarse fraction to $PM_{10}$ is higher than that over the cold seasons. The authors believe that this enhancement would actually reflect the contribution from local anthropogenic activities, not only traffic related (since this is almost equally distributed over the year) but mainly to construction (an activity sector generating mainly coarse particles) corroborated with warm seasons characteristics at the investigated site (cessation in precipitation frequency).

*Comment (5) P9, lines 24-26: Why would dry deposition during the warm seasons be enhanced for $PM_{2.5}$ but not $PM_{10}$?*

Also in this case the referee is right (since the $PM_{10}$ fraction is very susceptible for higher than $PM_{2.5}$ deposition velocities) and, therefore, in the revision of the manuscript we propose "As suggested by Zhang et al. (2001), various land use categories (e.g., grass, crops, mixed farming, shrubs) corroborated with other particle-related characteristics (i.e., particle density, relevant meteorological variables) may enhance the dry deposition of sub-micron particles during the warm seasons and hence their fine/coarse ratio."

Zhang et al. (2001) report about a parameterization study on particle dry deposition by calculating dry deposition velocities ($V_d$) as a function of particle size, density as well as relevant meteorological variables. Within their study the authors have taken into account 15 land use categories (LUC) and 5 seasonal categories (SC). Results related to sensitivity test for dry $V_d$ depending on the selected LUCs for particles with various diameters (in the 0.1 to 5 µm range) highlight that $V_d$ values for particles with larger diameters varies in a wide range. It seems that for deciduous broadleaf trees, crops or shrubs LUCs a strong cessation in the deposition velocity of 5 µm in diameter particles is occurring, with $V_d$ values competing with those specific for particles of 0.1 or 0.5 µm in diameter. Moreover, the authors show that wind speed affects the deposition velocity values for all particles size ranges. For particles in the 0.1–2 µm size range the authors suggest that $V_d$ values can change by almost an order of magnitude when the wind speed changes from 2 to 15 m s$^{-1}$. In Iasi, north-eastern Romania, all-around the 2016 year wind-speed varied in the ~3–7 m s$^{-1}$ range and crops, mixed farming or shrubs related LUCs were expected to increase as importance over the warm seasons.

*Comment (6) P9, lines 30-32: I'm not sure how you arrived at the conclusion that the results are not due to emission sources but are likely due to meteorology based on the seasonal trends in the PM size distribution. Please explain how changes in the meteorological conditions and which met parameters affect the seasonal PM size distribution.*

In the revised manuscript we are going to propose "Again, changes in sources contributions and meteorological conditions could account for the observed differences (details in Section S 2 of the SM)".

In the Supplement Material we propose Section S 2 in the form:

"Beside changes in sources contributions, changes in meteorological conditions (i.e. relative humidity (RH) and wind speed (WS)) might induce distinct behaviour in the seasonal distribution of the PM. While at the investigated site over the warm seasons RH values varied in the range of ~ 30–50% and in the cold seasons in the range of ~ 45–80% we expect that at these RH values particles have proper meteorological conditions to grow. There is suggestion that in the 0 to 98% RH range, the number median diameter (NMD) may increase two-fold (from 0.018 to 0.036 μm) with the range of geometric standard deviation ($\sigma_g$) for the particle size distributions in the 1.53–2.06 range and with the larger number median diameter (NMD) having the smaller $\sigma_g$. Moreover, the authors suggest that the smaller particles grew proportionately more than the larger one (Sinclair et al., 1974).

As presented in FIG. 3 in Galon et al. work, in the 0.0556–0.946 μm particles size range, higher dM/dlogD$_p$ values over the cold seasons toward those specific for the warm seasons might actually reflect a cumulative effect induced by the contribution of the sub-micron growth particles due to higher RH values during the cold seasons. Shifts in particles diameter distribution toward higher values are expected in this case.

The observed 1.60–2.39 μm supermicrone mode is most probably a result of a more significantly contribution brought by large particles mainly associated with dust resuspension due to higher wind speeds. Although over the warm seasons averaged wind speed showed similar value to that specific for the cold seasons, monthly standard deviation toward the mean values suggested much larger dispersion. During the warm seasons, storms and rapid wind ghosts are known as often engaging important quantities of dust carrying especially particles of larger diameter."

*Comment (7) P10, lines 6-9 and other places: I suggest moving the detailed statistics in parentheses to a table or the supplement because it makes the rest of the text difficult to follow.*
As suggested by reviewer comment the detailed statistics is reported in the Supplement Material.

*Comment (8) P10, lines 22-28: It's not clear what the missing $NH_4^+$ is referring to because $NH_4^+$ measurements are available. Please briefly explain why there is another fraction of $NH_4^+$ that is not measured by the IC data.*

In the manuscript we propose now
"In the present work, $HCO_3^-/CO_3^{2-}$ was assigned as the missing anion (details in Section S 3 for $HCO_3^-/CO_3^{2-}$ estimation) while $NH_4^+$ as the main missing cation (details in Section S 4 for the missing $NH_4^+$ assumptions). Detailed statistics of the $\sum_{cations}$ vs. $\sum_{anions}$ dependences, with $HCO_3^-/CO_3^{2-}$ and missing $NH_4^+$ included in the ionic balance, is presented in Table S 3."

In the Supplement Material we propose Section S 3 in the form:
"The missing $HCO_3^-/CO_3^{2-}$ has been estimated as suggested by Arsene et al. (2007) while the rationale previously proposed by Arsene et al. (2011) has been used to estimate the missing $NH_4^+$. Within the $NH_4^+$(total) fraction (defined as the sum between that derived from raw IC data and the part estimated by using the rationale of Arsene et al. (2011)), the correction for the missing $NH_4^+$ accounted for about 21.65 ± 25.70 % for the warm seasons while for the cold seasons the correction accounted for about 46.05 ± 18.43 %. However, when estimated missing $NH_4^+$ has been taken into account a significant improvement in the overall ionic balance ($\sum_{cations}$ vs. $\sum_{anions}$) was observed for both the PM$_{2.5}$ and the PM$_{10}$ fractions (detailed statistics in Table S 3)."

and Section S 4 in the form
"During sampling, perturbations in the gas to particle equilibrium may occur with evaporation of semi-volatile $NH_4NO_3$ and $NH_4Cl$ salts from the fine particles collected on the front filters and

fluctuations during sampling in temperature, relative humidity and/or pressure drop across the filters might highly affect the measurements of these species especially in urban environments (Pathak and Chan, 2005; Ianniello et al., 2011). Moreover, the total concentration of ammonium salts in $PM_{2.5}$ is usually described as a sum of the measurements both for non-volatile (unevolved) and volatile (evolved) fine particulate species (Ianniello et al., 2011). Since in the present work during sampling neither denuders nor backup-filters were used, sampling artefacts of semi-volatile $NH_4NO_3$ and $NH_4Cl$ species are not completely excluded. Although presented data are reliable from the instrumental analysis point of view, limitations related to potential unmeasured species especially due to sampling procedure (during and after) should always be taken into account (i.e. potential $NH_4^+$, $NO_3^-$ and $Cl^-$ evolving from the filters as a result of $NH_4NO_3$ and $NH_4Cl$ dissociations).

In Galon-Negru et al. manuscript the completeness degree of the ionic balance for the identified and quantified species in both the $PM_{10}$ and $PM_{2.5}$ fractions was checked and quite often slopes lower than unity in the $\sum_{cations}$ vs. $\sum_{anions}$ dependences were observed in both cases indicating important cation deficit in the ionic balance. As mentioned in the manuscript, per sampled events either cation or anion deficit has been observed in various stages. For similar behavior in Arsene et al. (2011) a detailed rationale has been proposed in order to explain/estimate either missing anions or cations within the ionic budget (under subsections 3.3 Role of inter-particle and gas-particle interactions in establishing fine and coarse fractions chemical composition and 3.4 The ionic balance in the coarse and fine fractions of Arsene et al. (2011) paper). As given in Arsene et al. (2011), meteorological conditions favourable to generate deliquesced particles in the form of $NH_4NO_{3(aq)}$ and $NH_4Cl_{(aq)}$ may exist in Iasi, north-eastern Romania, with deliquesced particles being formed under conditions of relative humidity at deliquescence (RHD), a parameter which is defined as the relative humidity at which deliquescence is completed.

While in Arsene et al. (2011) high RH values were representative for the warm seasons it seems that in Galon-Negru et al. manuscript high RH values, and hence representative conditions for deliquescence to occur, were mainly prevailing during the cold seasons. As suggested in Arsene et al. (2011) under deliquescence conditions, deliquesced particles may lead to the formation of aqueous droplets by a process favouring formation of internal mixture from externally mixed particles, which actually results in changes in the activities of semi-volatile species. According to Pathak and Chan (2005) in mixed collected particles the gas-particle equilibrium will tend to re-establish by mass exchange between the gas and particulate phases. Moreover, if deliquesced particles are formed under favourable meteorological conditions, then the following reactions can occur during inter-particle or gas-particle interactions:

$$NH_4Cl_{(s)} \xleftarrow{\text{RH>RHD}} NH_4Cl_{(aq)} \leftrightarrow NH_4^+{}_{(aq)} + Cl^-{}_{(aq)} \leftrightarrow NH_{3(g)} + HCl_{(g)}$$
$$(RS1)$$

$$NH_4NO_{3(s)} \xleftarrow{\text{RH>RHD}} NH_4NO_{3(aq)} \leftrightarrow NH_4^+{}_{(aq)} + NO_3^-{}_{(aq)} \leftrightarrow NH_{3(g)} + HNO_{3(g)}$$
$$(RS2)$$

$$H^+{}_{(aq)} + Cl^-{}_{(aq)} \leftrightarrow HCl_{(g)}$$
$$(RS3)$$

$$H^+{}_{(aq)} + NO_3^-{}_{(aq)} \leftrightarrow HNO_{3(g)} \qquad\qquad\qquad (RS4)$$
$$H^+{}_{(aq)} + NH_{3(g)} \leftrightarrow NH_4^+{}_{(aq)} \qquad\qquad\qquad (RS5)$$

with (RS1), (RS2), (RS5) being responsible for artefacting ammonium measured concentrations. While, actually, reactions (RS1) and (RS2) would be responsible for negative ammonium artefacts, reaction (RS5) would involve absorption of ammonia on aqueous droplets with positive artefacts. The competition between (RS1), (RS2) and (RS5) reactions will strongly depend on meteorological conditions directly related to RH, RHD and temperature but also on chemical species abundances. According to Pathak and Chan (2005) in ammonium reach environments, reaction (RS5) would be responsible on highly contributing artefacts in $NH_4^+$ and $H^+$ distribution, especially due to acidity neutralisation by the existent ammonia. In their work, Arsene et al. (2011) proposed a detailed rationale for a potential estimation of the unmeasured $NH_4^+$ fraction induced most probable by the reactions inducing negative artefacts and significantly occurring during sampling. In Galon-Negru

et al. work this unmeasured $NH_4^+$ has been called "missing $NH_4^+$" and for its estimation Arsene et al. (2011) rationale has been used."

*Comment (9) P11, lines 4-23: This text could be condensed or moved to the methods section or the supplement. This section should present the results of the model runs.*

As suggested by the reviewer this part is transferred to the Supplement Material, in the Section S 5.

*Comment (10) P12, lines 20-23: What are the ecosystem or human health implications of strongly acidic aerosols? I suggest providing a brief explanation on the importance of aerosol acidity. What is contributing to the strongly acidic aerosols, pH 0-3? At this aerosol pH, does it contribute significantly to acid rain in this region? Are there any reports of acid rain impacts in this region?*

In the manuscript we have included
"Details on the pH sensitivity tests toward $NH_4^+$ concentrations are given in Section S 6."
and
"Sulfuric, nitric, hydrochloric and formic acids are the most likely contributors to aerosols pH in the 0–3 range. Note that strongly acidic aerosols affect air quality, health of aquatic and terrestrial ecosystems (especially through acid deposition), as well as atmospheric visibility and climate (Dockery et al., 1996; Gwynn et al., 2000; Hennigan et al., 2015). Possible impacts of strongly acidic aerosols are presented in more detail in Section S 7."

[revised manuscript text omitted]

and Section S 7 in the form
"In the Introduction section of Galon-Negru et al. work it has been already stated that "Particles acidity might influence transition metals solubility and enhance aerosols toxicity and atmospheric nutrient delivered through atmospheric deposition in marine areas (Meskhidze et al., 2003; Fang et al., 2017).". Particles pH is known to affect the solubility of trace metals (such as Fe) found in aerosols (Meskhidze et al., 2003; Fang et al., 2017), with lower pH dissolving metal oxides and converting them to soluble metal sulfates (Oakes et al., 2012), conditions which would significantly change the aerosol environmental impacts. As presented in Guo et al., (2016), while at global level metal mobility will mainly affect nutrient distributions with important impacts on productivity, carbon sequestration, and oxygen levels in the ocean, at regional scales soluble transition metals seem mainly to enhance aerosol toxicity or their oxidative potential. Through $SO_4^{2-}$ contributions to aerosols acidity and the historical record of associations between so-called particle "strong acidity" and adverse health effects, Fang et al. (2017) suggest that $SO_4^{2-}$ linkages to health are most probably determined by $SO_4^{2-}$ role in acid dissolution of primary metals commonly found in ambient particles, while Tsagkogeorgas et al. (2017) suggest that on a regional to global scale the acidification of fresh water and forest ecosystems is most probable caused by wet and dry deposition of $SO_2$ and $SO_4^{2-}$ particles.

Epidemiological studies have often reported adverse health outcomes associated with strong aerosols acidity (Dockery et al., 1996; Gwynn et al., 2000; Lelieveld et al., 2015). In "Measuring aerosol damage to the atmosphere" (http://ec.europa.eu/research/infocentre/article_en.cfm?artid=31296, 28 October 2013), "Arsene says the high aerosol levels are probably linked to the high rate of various pulmonary diseases registered at the Clinic of Pulmonary Diseases in Iasi. These include cases like chronic obstructive pulmonary diseases (COPD), pneumonia, asthma (allergy, rhinitis), bronchiectasis (sometimes associated with bacterial infection) and tuberculosis often detected in the area.". The claim is not a result of a systematic work on an epidemiological study in Iasi, north-eastern Romania, but it represents a general observation of Arsene after performing research work with specialists and medical doctors in the field of pulmonary diseases (Cernat et al., 2011).

Moreover, upon our knowledge there is a single report, mentioned by Arsene et al. (2007), referring to indirect evidences of acid rain impacts in Iasi region, north-eastern Romania. In the

performed study, Arsene et al. (2007) calculated high wet deposition fluxes for ions of anthropogenic origin in the $SO_4^{2-} > NO_3^- > NH_4^+$ order. The determined values, although among the highest, were in generally in good agreement with values reported for other north European sites (such as those for Poland, the Czech Republic, etc.). In the analysed raining events, air masses crossing very large continental areas (N and E sectors) brought the highest contribution in terms of fluxes for ionic species such as $Ca^{2+}$ while the NW and W sectors brought more significant contributions for anthropogenic related elements ($SO_4^{2-}$, $NO_3^-$, $NH_4^+$). Arsene et al. (2007) suggested that in the rainwater, higher fluxes of these elements for the NW and W sectors, when compared with other geographical sectors, was most probably also a result of important local influences, as the sampling site was located east side of the Oriental Carpathians chain that could constrain, to some extent, the pollution plume transported from western Europe. Arsene et al. (2007) report that the pH of the analysed rainwater events was 5.92 (volume weighted mean average, VWM) suggesting a sufficient load of alkaline components neutralizing rainwater acidity. Moreover, the authors suggested that on average, 97 % of the acidity in the collected rainwater samples was mostly neutralized by $CaCO_3$ and $NH_3$."

*Comment (11) P13 line 29 – P15 line 9: This section examined $NH_3/NH_4^+$ partitioning as a function of relative humidity. Considering that ammonium tends to be associated with sulfate and nitrate, I think that the discussion on $NH_3/NH_4^+$ partitioning needs to consider changes in the sulfate and nitrate concentrations as well as temperature which affect both sulfate and nitrate production. Also, temperature and relative humidity are strongly correlated. Can you confirm that the relationship with relative humidity is not actually due to temperature? Do you still see the strong relationship between NH3/NH4+ and relative humidity when you analyze cold and warm seasons separately?*

The authors agree with reviewer concerns about $NH_3/NH_4^+$ partition in relation to $NO_3^-$ and $SO_4^{2-}$ behaviour but a suitable answer to this will require more laborious approaches which will increase the length of the manuscript and, moreover, are far beyond of the manuscript aim.

In the manuscript we have included:
"The detailed $NH_3/NH_4^+$ partitioning as a function of RH is presented below, and considerations on the potential effects on the partitioning brought about by changes in the $SO_4^{2-}$ and $NO_3^-$ concentrations, and by temperature affecting both $SO_4^{2-}$ and $NO_3^-$ production, is given in Section S 8 of the SM. The potential role played by temperature on $NH_3/NH_4^+$ partition seems to be minimal, but one should also consider that highly acidic aerosols will affect a variety of processes and definitely the partitioning of $HNO_3$ to the gas phase, producing low nitrate aerosol levels."

In the Supplement Material we propose Section S 8 in the form:
"Regarding meteorological parameters, i.e. RH and temperature, in Galon-Negru et al. work is shown that over the cold season's high RH values and low temperatures were prevailing while over the warm seasons these parameters showed opposite trends. In the work of Ianniello et al. (2011), related to chemical characteristics of inorganic ammonium salts in $PM_{2.5}$ in the atmosphere of Beijing (China), the relationship between partitions of specific nitrogen-containing inorganic species towards various parameters has been investigated. The prevailing conditions over the two investigated time-periods, i.e. 23 January – 14 February 2007 and from 2 to 34 August 2007, showed higher RH values over the summer (high temperatures) and lower RH values over the winter (low temperatures).

Ianniello et al. (2011) treats in a very complex manner the relationship existent between particulate species such as $SO_4^{2-}$, $NO_3^-$, $Cl^-$ and $NH_4^+$, and possible partition routes in the $NH_3/NH_4^+$ system. Gas-to-particle conversion (photochemical processing) or heterogeneous processes are assigned as important sources contributing to particulate $NO_3^-$ abundances in the atmosphere. The authors claim that while $NO_2$ conversion to $NO_3^-$ through photochemical processing during the winter season is expected to contribute to $NO_3^-$ abundance over this period of

the year, the heterogeneous formation (through condensation or absorption of $NO_2$ in moist aerosols) generally relates to relative humidity and the particulate atmospheric loading (Ianniello et al., 2011). The existence of large amounts of particulate $NO_3^-$, observed in summer by Ianniello et al., 2011, was considered as unexpected since $NH_4NO_3$ is semi-volatile and tends to dissociate and remain in the gas phase under high temperatures. Ianniello et al., (2010) report 6 times higher $NH_3$ concentrations in summer than in winter, with temperatures ranging from 1 to 14 $^o$C in winter and 22 to 35 $^o$C in summer periods. Under these circumstances high concentrations of fine particulate $NO_3^-$ in summer period were attributed to the existence of higher concentrations of $NH_3$ in the atmosphere, available to neutralise not only $H_2SO_4$ but also $HNO_3$ from the atmosphere. In addition, at the high RH values (daily mean in the 35 to 90 % range) reported in Ianniello et al. (2011) study significant fraction of $HNO_3$ and $NH_3$ were considered to be dissolved in humid particles, with enhanced distribution of fine particulate $NO_3^-$ and $NH_4^+$ in the atmosphere.

In Galon-Negru et al. work particulate $NH_4^+$ showed higher values during the cold seasons when compared to the warm seasons and its variation coincide with those of fine particulate $NO_3^-$ and $Cl^-$. Although such behaviour would indicate that $NH_4^+$ was largely originating from the neutralization between ammonia and acidic species such as $HNO_3$ and $HCl$, as shown under Subsection 3.2.3 (Stoichiometry of $(NH_4)_2SO_4$, $NH_4NO_3$ and $NH_4Cl$), the most important contribution was most probably brought by $H_2SO_4$ species. However, the very similar pattern observed for $NO_3^-$, $Cl^-$ and $NH_4^+$ suggest that these species were representative for likely internally mixed particles and came most probably from similar gas-to-particle processes (Huang et al., 2010). Galon et al. report for $SO_4^{2-}$ comparable concentrations during both the summer and winter seasons, and Backes et al. (2016a) suggest that the formation of $SO_4^{2-}$ particles is not limited by $NH_3$ in any season. During the warm seasons, higher temperatures and solar radiation intensity, enhancing the photochemical activity and the atmospheric oxidation potential, is also enhancing the oxidation rate of $SO_2$ to particulate $SO_4^{2-}$, but during the cold seasons wood burning might become an important source of $SO_4^{2-}$.

At high RH values, and especially for deliquescent particles, there is suggestion that most of the fine particulate $NO_3^-$ exists as an internal mixture with $SO_4^{2-}$, so that $HNO_3$ can be easily absorbed into the droplets, a process considerably reducing the thermodynamic dissociation constant for $NH_4NO_3$ (Zhang et al., 2000; Ianniello et al., 2011). Under these circumstances it is supposed that fine particulate $NO_3^-$ can be formed from $HNO_3$ and $NH_3$ through heterogeneous reactions on fully neutralised fine particulate $SO_4^{2-}$ (a process which has been taken into account also in Galon et al. work, under the Subsection 3.2.3). In Galon et al. work significant correlation has been observed between $SO_4^{2-}$ and $NO_3^-$. These two measured parameters showed significant correlation with the RH, with high concentrations in $SO_4^{2-}$ or $NO_3^-$ being formed at high RH values. Such behaviour would allow us suggesting that $NO_3^-$ is being produced on preexisting $SO_4^{2-}$ aerosols, which could provide sufficient area and aerosol water content for the heterogeneous reaction to occur should be available. Markovic et al. (2011) show that at high RH, the amount of the gaseous precursors, such as $NH_3$ and $HNO_3$, have relatively little influence on the formation of fine particulate nitrate. Ianniello et al. (2011) concluded also that in summer period, with high prevailing RH values, almost on all days the meteorological conditions were favourable for the formation of $NH_4NO_3$ at Beijing site.

In Galon et al. work, regarding gaseous $NH_3$ values such as those derived from ISORROPIA, it should be pointed out that its estimated concentration was much higher than that of $HNO_3$ and $HCl$ in gas phase. During the cold seasons the $[NH_3]/([HNO_3] + [HCl])$ ratio is as high as $2.0 \pm 0.6$ for RH values < 40 %, $3.6 \pm 2.0$ for RH values in the 40–60 % range, and $3.0 \pm 1.4$ for RH values > 60 % RH. During the warm seasons, for similar RH groups, the ratio takes the $4.9 \pm 1.9$, $4.6 \pm 1.4$ and $9.1.0 \pm 6.2$ values. Ianniello et al. (2011) report a $[NH_3]/([HNO_3] + [HCl])$ ratio of $27.90 \pm 12.70$ for winter season and of $54.06 \pm 20.60$ for the summer period. The authors are finally claiming that the atmosphere of their interest location was ammonia-rich in gas phase over both investigated seasons. Moreover, for the specific conditions referring to Galon et al. work, the authors have observed that by analysing the events for similar RH groups, but for cold and warm seasons separately, the $NH_3/NH_4^+$ partition didn't enhance significantly. During the cold seasons

over the three RH investigated groups, the fractions of the $NH_3$, HCl and $HNO_3$ present in the gaseous form have taken in average the $71.4 \pm 4.7$, $4.4 \pm 0.3$ and $24.0 \pm 4.8\%$ values. Over the warm seasons for the previously mention species the fractions present in the gaseous form have taken in average the $83.1 \pm 3.4$, $6.6 \pm 0.6$ and $10.3 \pm 3.4\%$ values."

*Comment (12) P14, lines 29-31: I think that this conclusion needs to be corroborated by examining the seasonal trend in the ammonia emissions since ammonia concentrations were not available. Ammonia is typically higher during the warm seasons, but here you suggest it is high enough all year round.*

In the manuscript we have included:
"The seasonal trends in the $NH_3$ concentrations derived from ISORROPIA runs for Iasi are reported in FIG. S 1 (Section S 9). The same section reports considerations on possible interrelated emission factors governing the distribution in the $NH_3$ concentration levels in Iasi."

In the Supplement Material we propose Section S 9 in the form:
"Backes et al. (2016a,b) present in some of their publication seasonal distributions in $NH_3$ emissions under static- and dynamic-time profile (STP and DTP) scenarios. While the STP scenario lacks dynamic, meteorology dependent or specific differences in policies or intensity of animal husbandry, the DTP scenario takes into account potential meteorological variables (such as wind speed and surface temperature) influence on the distribution of the annual emissions. However, while under the STP scenario, the annual time series of $NH_3$ showed one annual peak in spring (March), in the DTP scenario $NH_3$ concentrations show two annual peaks, one in spring (May) and one in autumn (September). The authors underline that implementation of the DTP resulted in a shift from seasonal average winter $NH_3$ emissions to summer emissions. According to Backes et al. (2016a), high $NH_3$ concentrations are known to appear in proximity to emission sources, due to its low atmospheric lifetime which results in $NH_3$ concentration levels in the atmosphere which closely follow the seasonal emission trend.

Moreover, Sutton et al. (2013) claim that together with increased anthropogenic activity, global $NH_3$ emissions may increase from 65 (48–85) Tg N in 2008 to about 132 (89–179) Tg N by 2100. Most $NH_3$ emissions are known to result from agricultural productions strongly influenced by climatic interactions. There is, however, recognition about the fact that most of the up to now used approaches failed in recognising that a warm dry-year would tend to give larger $NH_3$ emissions than a cold-wet year. Estimates in the global $NH_3$ emission made by Sutton et al., (2013) indicate agricultural soils and crops, including emissions from grazing and land application of animal manure, as a first ammonia ranked source with 28.3 Tg N $yr^{-1}$ contribution. This is immediately followed by excreta from domestic animals (8.7 Tg N $yr^{-1}$), oceans and volcanoes (8.6 Tg N $yr^{-1}$), biomass burning (5.5 Tg N $yr^{-1}$), waste composting and processing (4.4 Tg N $yr^{-1}$), and by other sources with contributions of < 3.3 Tg N $yr^{-1}$.

[Figure]

(a)

[Figure]

(b)

**Figure S 1**: Time series in $NH_3$ concentrations derived from ISORROPIA-II runs for the 2016 data-base, both for $PM_{2.5}$ and $PM_{10}$ scenarios in Iasi, north-eastern Romania.

Details presented in FIG. S 1 clearly show that for Iasi, north-eastern Romania, yearly $NH_3$ distributions as derived from ISORROPIA-II model present clear maxima in February and December in a behaviour suggesting important local sources contributions over the winter season. It might be that this distribution follows actually a trend mainly controlled by the agricultural practices in rural areas surrounding Iasi, mainly related to the timing of manure spreading on agricultural soils. Moreover, in Iasi and in the nearby counties, the contribution brought by open (sheep and goat) barns, operating over the cold season mainly in a "hot-spot" mode, might increase as importance. Manure storage from local and regional closed barns (pigs and poultry) might also bring important contributions over the winter. Although biomass burning might bring important contribution in terms of global ammonia emissions it is believed that at the interest location this source is overwhelmed by agricultural practices in the area."

*Comment (13) P15, lines 13-16: Are these results expected? For the ions associated with fine particles, are the percentage increases lower than those ions associated with coarse particles?*

The increase, as given in the manuscript, was estimated on the base of the formula:

$$increase = \frac{\left(C_{A_{PM_{10}}} - C_{A_{PM_{2.5}}}\right)}{C_{A_{PM_{2.5}}}} \times 100(\%)$$

The behaviour presented in the manuscript is going to be expected since these % increases reflect the increase in analyte concentration in the $PM_{10}$ fraction toward analyte concentration in the $PM_{2.5}$ fraction.

For example $Ca^{2+}$ and $HCO_3^-$ are ions preponderantly found in the coarse fraction, with very little contribution in the fine fraction. Estimating % increase in the $PM_{10}$ fraction toward the $PM_{2.5}$ fraction will generate very high % values (in average 61 % for $Ca^{2+}$, 63 % for $HCO_3^-$). Opposite behaviour is observed for the ionic species residing mainly in the fine fraction with little contribution to the coarse fraction (17 % for $SO_4^{2-}$, 12% for $NH_4^+$, etc.).

*Comment (14) P15, lines 18-21: Sulfate is usually higher during warm seasons because of increased oxidation of SO2. I'm not sure about the explanation of lower mixing heights or temperature inversions because not all of the pollutants were higher during cold seasons.*

In the manuscript we have now included:
"Although lowering mixing heights over the cold seasons might increase pollutant concentration in the atmosphere, for some species additional phenomena should be taken into account in order to explain their distribution. For particulate $SO_4^{2-}$, high concentrations can be observed during winter and autumn but also in summer, and in the latter case they can be due to higher temperature and solar radiation that enhance photochemical reactions and the atmospheric oxidation potential, because of the elevated occurrence of oxidant species such as ozone, hydroxyl and nitrate radicals. These conditions favour the oxidation of $SO_2$ to particulate $SO_4^{2-}$. Also particulate $C_2O_4^{2-}$ was maximum in summer, possibly due to enhanced photochemical processing. Moreover, the maxima observed for $SO_4^{2-}$ during the cold seasons might be due to the intensification of coal burning for heating purposes."

*Comment (15) P17, line 5: Ammonia is usually higher during warm seasons because of higher temperatures and increased agricultural emissions. What are the sources of ammonia during cold seasons?*

Since the discussion on this specific comment is related also to #12 comment, in the manuscript we have included:

"Reactive nitrogen species are emitted to the atmosphere mainly in the forms of $NO_x$ (from transport or power generation) and $NH_3$ (agriculture). In Iasi, the animal husbandry sector (open and closed barns, manure storage/spreading) is most likely an important $NH_3$ source."

*Comment (16) P17, lines 16-19: The relative humidity explanation for the high sulfate concentrations is not consistent with the lack of correlation between sulfate and meteorological parameters mentioned earlier (lines 12-13).*

We agree with reviewers' observation and therefore in the text we now have
"Particulate $SO_4^{2-}$ maxima are observable during both cold and warm seasons. However, the particulate $SO_4^{2-}$ mass concentrations showed statistically significant correlation with the measured meteorological parameters only at a 68 % confidence level (detailed statistics in Table S 4)."

*Comment (17) P22, lines 13-20: The molar ratios in Fig. 7cdef between cold and warm seasons are not very different and all the R2 values were close to 1. It seems that a complete neutralization of H2SO4 and HNO3 also occurred in the warm seasons. Can you explain the lack of difference in the molar ratios?*

In Galon et al. manuscript there is a very important part describing in detail chemical and physical processes potential to govern the stoichiometry of species such as $(NH_4)_2SO_4$, $NH_4NO_3$ and $NH_4Cl$ as a result of a complete neutralization of $H_2SO_4$ and $HNO_3$ occurring in both during the cold and the warm seasons. Although FIGS. 7ef are actually of little importance authors thoughts were to include also these scenarios in order to underline the potential impact induced by the $Cl^-$ ion. The lack of difference in the molar ratios observed from FIGS. 7ef toward FIGS. 7cd is most probably induced by the fact that HCl neutralisation by ammonia would have very little contribution in establishing the atmospheric burden of specific particulate form. Moreover, ISORROPIA runs allowed us identify that in the investigated events HCl contribution in the gaseous form was over the entire year almost insignificant in comparison to that of $HNO_3$. We consider that all other possible explanations are already presented in the manuscript.

*Comment (18) P27-28: This section can be improved by showing the inorganic ion concentrations associated with the different air mass origins identified in Fig. 1. While physical and chemical processes are important in the formation of secondary pollutants, it is also important to examine where the gaseous precursors e.g. SO2, NOx and NH3 are coming from. There might be a change in the air masses during different times of the year leading to seasonal variability in the gaseous precursors, which in turn affects the secondary pollutants.*

Although reviewer comment is important also in this case a suitable answer would increase even more paper's length. However, in order to supply details on the requested information in the Supplement Material we have included representative charts both for the ionic chemical constituents and also for gaseous species as derived from ISORROPIA runs. The charts are associated to specific selected sampled event such as clearly to highlight the contribution of the different air mass origins identified in FIG. 1 of the manuscript. Although we do not have yet gaseous precursors (e.g. $SO_2$, $NO_x$ and $NH_3$) measurements the distributions in the given Radar or Pie charts will clearly show air mass sector contributions toward species as $NH_3$, $HNO_3$, and HCl (as derived by ISORROPIA-II).

In the manuscript we have included:

"Annual averaged sector contributions, in terms of PM long-range transport, are shown in FIG. S 2 while the seasonal contributions of PM and particulate inorganic/organic ions associated with different air mass origins are reported as Radar charts in FIG. S 3 (within Section S 10). FIG. S 4 (within Section S 10) highlights the % distributions for the identified-quantified ions and the gaseous concentrations of $NH_3$, $HNO_3$ and $HCl$ for selected investigated events, predicted by ISORROPIA-II."

In the Supplement Material we propose Section S 10 in the form:
"The data set from the present work has been investigated through Radar and/or Pie charts in order to identify the potential contributions brought by various sectors long-range transport to the atmospheric PM burden in the area. It should be however emphasized that the PM chemical composition appeared to be mainly driven by the air mass origin and type (e.g., continental, regional), meteorologically-controlled air mass characteristics (buoyancies), geographical context (the Oriental Carpathians chain, with the highest altitude of 1907 m, is facing Iasi ~ 150 km north-westerly distance), etc. As presented in FIG. S 2a, in overall, in terms of sectors contributions the N-NE up to the S (clockwise round) area seems to bring the most important fraction to both the $PM_{2.5}$ and $PM_{10}$ abundances. Higher loaded events in terms of aerosols mass concentration were those associated with long-range continental transport, Saharan dust-events, regional transport from more arid areas (S-SE) and buoyancy affected air masses.

Regarding potential local anthropogenic contribution to PM atmospheric burden it should be emphasized that in the immediately nearby area of the sampling point neither important commercial areas nor industrial units are operating and traffic should be the most important contributor. Actually in Iasi, in the post-communist period, the industry has almost completely ceased down and presently the Antibiotic company (7 km from the sampling point, western, W, direction), a small brick plant (8 km, SE) and the heat and power plant (5 km, SE) may account as for the most significant local anthropogenic contributors. However, Iasi is known as an important Romanian university centre (> 42000 students by 2015) with intense activity over university semesters. Traffic in the nearby area and commercial activity in the top hot-spots area of Iasi get more intense over these periods of the year. As for the rural areas surrounding Iasi, or from the nearby counties, farming activities are mainly related to cereal crops, plantation agriculture, open and closed animal barns (cattle, sheep and goat, pigs, poultry).

[Figure]

(S 2a)                                    (S 2b)

**Figure S 2**: Averaged annual sectors contributions both in the $PM_{2.5}$ (S 2a) and $PM_{10}$ (S 2b) fractions.

**Figure S 3**: Particulate inorganic and organic ions seasonal contributions (in the PM$_{2.5}$ fraction) associated with the different identified air mass. Note: NH$_4^+$$_{total}$.

[Figure]

[Figure]

[Figure]

[Figure]

[Figure]

[Figure]

[Figure]

[Figure]

**Figure S 4**: Trajectory air mass associated percentual contributions for the identified/quantified ions and ISORROPIA predicted gaseous $NH_3$, $HNO_3$ and HCl both in $PM_{2.5}$ (top) and $PM_{10}$ (bottom) fractions. Selected investigated events are presented. Note: Fo – formate, Ac – acetate, Ox – oxalate, $NH_4^+$ reflects $NH_4^+{}_{total}$.

It should be also underlined that for ionic species specific for the coarse fraction, i.e. $Na^+$ and $Ca^{2+}$, the maxima seems to be mainly induced either by the prevalent direction of the air masses (spring maxima of $Na^+$ ion most probably induced by the fact that March and April appeared as months with prevalent southern air masses crossing large continental areas with more salty soil) or by the drought (the summer maxima of $Ca^{2+}$ was mainly influenced by the ionic chemical composition of aerosol samples collected in August 2016, 7 events, a month of the 2016 year without any single raining events). Actually, in August 2016, 1 event was strongly affected by atmospheric driven buoyancy (18.2 % fine particulate $Ca^{2+}$ and 42.1 % fine particulate $HCO_3^-$), 1 event was affected by turbulent local air masses (10.7 % fine particulate $Ca^{2+}$ and 24.8 % fine particulate $HCO_3^-$) and other 3 events showed contributions brought by air masses crossing larger continental areas (9.8 ± 1.7 % fine particulate $Ca^{2+}$ and 25.3 ± 3.9 % fine particulate $HCO_3^-$). Other 2 events sampled at the beginning of August 2016 (of N-NW direction both) in terms of $Ca^{2+}$ and $HCO_3^-$ brought contributions as high as (4.3 ± 0.6 % fine particulate $Ca^{2+}$ and 9.7 ± 2.3 % fine particulate $HCO_3^-$). High percentual contributions within the identified and quantified species brought by $Ca^{2+}$ and $HCO_3^-$ ions were also observed in various events collected all year round which were affected by long-range transport air masses accompanied by atmospheric driven

buoyancy or by long-range transport air masses coming from arid areas (e.g. 31 January 2016, 14 February 2016, 20 March 2016, 06 April 2016, 09 April 2016, etc). It has been observed that in average the contributions brought by such events were strongly affecting the distribution of the ionic chemical composition in sampled aerosols."

[revised manuscript text omitted]

**Supplement material (SM) for the manuscript https://doi.org/10.5194/acp-2017-1030**

**Chemical characteristics of size resolved atmospheric aerosols in Iasi, north-eastern Romania. Nitrogen-containing inorganic compounds controlling aerosols chemistry in the area**

Alina Giorgiana Galon-Negru[1], Romeo Iulian Olariu[1,2], Cecilia Arsene[1,2]

[1]"Alexandru Ioan Cuza" University of Iasi, Faculty of Chemistry, Department of Chemistry, 11 Carol I, 700506 Iasi, Romania
[2]"Alexandru Ioan Cuza" University of Iasi, Integrated Centre of Environmental Science Studies in the North Eastern Region, 11 Carol I, Iasi 700506, Romania

*Correspondence to*: Cecilia Arsene (carsene@uaic.ro); Phone number: +40-232-201354; Fax: +40-232-201313; Postal address: Cecilia Arsene, "Alexandru Ioan Cuza" University of Iasi, Faculty of Chemistry, Department of Chemistry, 11 Carol I, 700506 Iasi, Romania

**Table S 1**: Detailed statistics for the $PM_{10}$ and $PM_{2.5}$ fractions mass concentrations determined over the investigated period (n = 84 sampling events) in Iasi, north-eastern Romania.

| Applied test | $PM_{2.5}$ | | $PM_{10}$ | |
|---|---|---|---|---|
| | p-value | Confidence level (%) | p-value | Confidence level (%) |
| Shapiro-Wilk normality test | 0.546 | 95 | 0.682 | 95 |
| t-test | 0.987 | 95 | 0.998 | 95 |

**Section S 1**

Possible explanations for the differences/similarities between afore mentioned studies might be seen as a result of the contributions brought from various factors. As shown below the differences in the PM mass concentrations reported in Galon et al. and those reported in Arsene et al. (2011) might be only slightly controlled by changes in the particle size cut-off and/or by the difference in sampling site altitude (25 m in Galon et al. vs. 35 m in Arsene et al. (2011)) which is too small to account for the difference in PM concentrations.

In Arsene et al. (2011) study, aerosol sampling has been undertaken by using a stacked filter unit (SFUs) consisting of a 8.0 μm pore size 47-mm diameter Isopore polycarbonate filter mounted in front of a 0.4 μm pore size 47-mm diameter Isopore filter. According to details in Arsene et al. (2011) and references included therein, the 50% cut-point diameter (D50) of the 8.0 μm pore size filter was estimated to be of the order on 1.5 ± 0.2 μm aerodynamic equivalent diameter (AED). Consequently, particles collected on the 8 μm pore size filters, with a diameter larger than 1.5 μm AED, were referring to the aerosol coarse fraction, while the particles collected on the 0.4 μm pore size filters were attributed to particles with a diameter below 1.5 μm AED. In Galon et al. work aerosols samples were collected on 25 mm in diameter ungreased aluminum filters using a cascade Dekati Low-Pressure Impactor (DLPI), operating at a flow rate of 29.85 L min$^{-1}$. Enhanced

sampling efficiency of the DLPI unit used in Galon et al. work, with regard to fine and ultrafine particles, has been observed in comparison with the SFU unit reported by Arsene et al. (2011). Parallel DLPI and SFU sampling runs (performed within January–July 2016) showed that the DLPI unit could collect in average with ~ 6 µg m$^{-3}$ more particles than the SFU system and this observation allowed us suggesting that most probably the operational limits of the SFU system with regard to fine and ultrafine particles could suppress to some extend the values reported by Arsene et al. (2011).

The difference observed between the PM$_{10}$ mass concentration, i.e. 18.9 ± 9.3 µg m$^{-3}$, reported by Galon et al. and the PM$_{total}$ mass concentration, i.e. 38.3 ± 25.4 µg m$^{-3}$, reported by Arsene et al. (2011) might actually be the result of sampling in different years with emission sources of various prevalence. Implementation of environmental quality management systems in various sectors with anthropogenic activity may account for that (e.g. attempts given by the local administrative sector in settling down soil- or road-resuspended dust especially over the warm seasons by wetting the roads, protecting areas with intense building activities, etc.).

Regarding other literature available values, in a study from 2016, Alastuey et al. (2016) report for PM$_{10}$ concentrations in Moldova (the closest point to our sampling site), values as high as ~ 25 µg m$^{-3}$ over summer and of ~ 25–30 µg m$^{-3}$ over winter period, yielding an annual averaged value of ~ 27.5 µg m$^{-3}$ which is much higher than the 18.9 µg m$^{-3}$ value reported in the present work, for Iasi, north-eastern Romania. However, the elevated concentrations observed at the eastern sites were attributed to regional or local sources (Alastuey et al., 2016). In Alastuey et al. (2016) the author reports only the altitude above sea level (i.e. 156 m for Moldova's location) and the sampling a.g.l. is not mentioned. It might be that in Alastuey et al. (2016) study if the Moldavian sampling location was assigned as an EMEP site than the sampling a.g.l. was settled in agreement with an EC Directive recommending an a.g.l. of ~ 4 or 8 m. It is believed that the difference observed between Galon et al. site (i.e. 18.9 ± 9.3 µg m$^{-3}$) in comparison with other European sites might actually reflect that sampling altitude would be an important controlling factor to the atmospheric aerosol burden at a site.

In Iasi, north-eastern Romania, mass concentrations in the 5 to 10 µg m$^{-3}$ range, for the PM$_{10}$ and PM$_{2.5}$ fractions were especially observed in samples collected after raining events (i.e., May, June, August, and October). Such behaviour would be expected since particles from the atmosphere might be efficiently removed by precipitation (Arsene et al., 2011). However, during events with strong natural or anthropogenic contributions the PM$_{10}$ and PM$_{2.5}$ mass concentrations exceeded the averages observed at AMOS. For example, an event collected in April 2016, from 9[th] to 11[th], with 34.3 µg m$^{-3}$ in PM$_{2.5}$ and 43.9 µg m$^{-3}$ in PM$_{10}$, was actually highly influenced by the long range transport phenomena of African dust and marine aerosols from the Black and Aegean/Mediterranean seas (i.e., event described in detail later in the text). For other events, the exceeding fine fraction mass concentrations are believed to be a result of more variable sources (combustion, biogenic, local mineral dust, meteorological factors, etc.).

**Section S 2**

Beside changes in sources contributions, changes in meteorological conditions (i.e. relative humidity (RH) and wind speed (WS)) might induce distinct behaviour in the seasonal distribution of the PM. While at the investigated site over the warm

seasons RH values varied in the range of ~ 30–50% and in the cold seasons in the range of ~ 45–80% we expect that at these RH values particles have proper meteorological conditions to grow. There is suggestion that in the 0 to 98% RH range, the number median diameter (NMD) may increase two-fold (from 0.018 to 0.036 μm) with the range of geometric standard deviation ($\sigma_g$) for the particle size distributions in the 1.53–2.06 range and with the larger number median diameter (NMD) having the smaller $\sigma_g$. Moreover, the authors suggest that the smaller particles grew proportionately more than the larger one (Sinclair et al., 1974).

As presented in FIG. 3 in Galon et al. work, in the 0.0556–0.946 μm particles size range, higher dM/dlogD$_p$ values over the cold seasons toward those specific for the warm seasons might actually reflect a cumulative effect induced by the contribution of the sub-micron growth particles due to higher RH values during the cold seasons. Shifts in particles diameter distribution toward higher values are expected in this case.

The observed 1.60–2.39 μm supermicrone mode is most probably a result of a more significantly contribution brought by large particles mainly associated with dust resuspension due to higher wind speeds. Although over the warm seasons averaged wind speed showed similar value to that specific for the cold seasons, monthly standard deviation toward the mean values suggested much larger dispersion. During the warm seasons, storms and rapid wind ghosts are known as often engaging important quantities of dust carrying especially particles of larger diameter.

**Table S 2**: Detailed statistics of linear regression analysis for $\sum_{cations}$ and $\sum_{anions}$ determined for raw ion chromatography data in the PM$_{2.5}$ and PM$_{10}$ fractions over the total investigated period, cold and warm seasons, in Iasi, north-eastern Romania.

| | PM$_{2.5}$ | | | | PM$_{10}$ | | | |
|---|---|---|---|---|---|---|---|---|
| | $\sum_{cations}/\sum_{anions}$ ratio | r | p-value | Confidence level (%) | $\sum_{cations}/\sum_{anions}$ ratio | r | p-value | Confidence level (%) |
| Total period | 0.69 | 0.94 | < 0.001 | 99.9 | 0.70 | 0.94 | < 0.001 | 99.9 |
| Cold seasons | 0.67 | 0.98 | < 0.001 | 99.9 | 0.68 | 0.98 | < 0.001 | 99.9 |
| Warm seasons | 0.84 | 0.87 | < 0.001 | 99.9 | 0.86 | 0.86 | < 0.001 | 99.9 |

Note: r–Pearson coefficient

**Section S 3**

The missing $HCO_3^-/CO_3^{2-}$ has been estimated as suggested by Arsene et al. (2007) while the rationale previously proposed by Arsene et al. (2011) has been used to estimate the missing $NH_4^+$. Within the $NH_4^+$(total) fraction (defined as the sum between that derived from raw IC data and the part estimated by using the rationale of Arsene et al. (2011)), the correction for the missing $NH_4^+$ accounted for about 21.65 ± 25.70 % for the warm seasons while for the cold seasons the correction accounted for about 46.05 ± 18.43 %. However, when estimated missing $NH_4^+$ has been taken into account a significant improvement in the overall ionic balance ($\sum_{cations}$ vs. $\sum_{anions}$) was observed for both the PM$_{2.5}$ and the PM$_{10}$ fractions (detailed statistics in Table S 3).

**Section S 4**

During sampling, perturbations in the gas to particle equilibrium may occur with evaporation of semi-volatile $NH_4NO_3$ and $NH_4Cl$ salts from the fine particles collected on the front filters and fluctuations during sampling in temperature, relative humidity and/or pressure drop across the filters might highly affect the measurements of these species especially in urban environments (Pathak and Chan, 2005; Ianniello et al., 2011). Moreover, the total concentration of ammonium salts in $PM_{2.5}$ is usually described as a sum of the measurements both for non-volatile (unevolved) and volatile (evolved) fine particulate species (Ianniello et al., 2011). Since in the present work during sampling neither denuders nor backup-filters were used, sampling artefacts of semi-volatile $NH_4NO_3$ and $NH_4Cl$ species are not completely excluded. Although presented data are reliable from the instrumental analysis point of view, limitations related to potential unmeasured species especially due to sampling procedure (during and after) should always be taken into account (i.e. potential $NH_4^+$, $NO_3^-$ and $Cl^-$ evolving from the filters as a result of $NH_4NO_3$ and $NH_4Cl$ dissociations).

In Galon-Negru et al. manuscript the completeness degree of the ionic balance for the identified and quantified species in both the $PM_{10}$ and $PM_{2.5}$ fractions was checked and quite often slopes lower than unity in the $\sum_{cations}$ vs. $\sum_{anions}$ dependences were observed in both cases indicating important cation deficit in the ionic balance. As mentioned in the manuscript, per sampled events either cation or anion deficit has been observed in various stages. For similar behavior in Arsene et al. (2011) a detailed rationale has been proposed in order to explain/estimate either missing anions or cations within the ionic budget (under subsections 3.3 Role of inter-particle and gas-particle interactions in establishing fine and coarse fractions chemical composition and 3.4 The ionic balance in the coarse and fine fractions of Arsene et al. (2011) paper). As given in Arsene et al. (2011), meteorological conditions favourable to generate deliquesced particles in the form of $NH_4NO_{3(aq)}$ and $NH_4Cl_{(aq)}$ may exist in Iasi, north-eastern Romania, with deliquesced particles being formed under conditions of relative humidity at deliquescence (RHD), a parameter which is defined as the relative humidity at which deliquescence is completed.

While in Arsene et al. (2011) high RH values were representative for the warm seasons it seems that in Galon-Negru et al. manuscript high RH values, and hence representative conditions for deliquescence to occur, were mainly prevailing during the cold seasons. As suggested in Arsene et al. (2011) under deliquescence conditions, deliquesced particles may lead to the formation of aqueous droplets by a process favouring formation of internal mixture from externally mixed particles, which actually results in changes in the activities of semi-volatile species. According to Pathak and Chan (2005) in mixed collected particles the gas-particle equilibrium will tend to re-establish by mass exchange between the gas and particulate phases. Moreover, if deliquesced particles are formed under favourable meteorological conditions, then the following reactions can occur during inter-particle or gas-particle interactions:

$$NH_4Cl_{(s)} \xleftrightarrow{\;RH>RHD\;} NH_4Cl_{(aq)} \leftrightarrow NH_4^+{}_{(aq)} + Cl^-{}_{(aq)} \leftrightarrow NH_{3(g)} + HCl_{(g)} \tag{RS1}$$

$$NH_4NO_{3(s)} \xleftrightarrow{\;RH>RHD\;} NH_4NO_{3(aq)} \leftrightarrow NH_4^+{}_{(aq)} + NO_3^-{}_{(aq)} \leftrightarrow NH_{3(g)} + HNO_{3(g)} \tag{RS2}$$

$$H^+{}_{(aq)} + Cl^-{}_{(aq)} \leftrightarrow HCl_{(g)} \tag{RS3}$$

$$H^+{}_{(aq)} + NO_3^-{}_{(aq)} \leftrightarrow HNO_{3(g)} \tag{RS4}$$

$H^+_{(aq)} + NH_{3(g)} \leftrightarrow NH_4^+_{(aq)}$ (RS5)

with (RS1), (RS2), (RS5) being responsible for artefacting ammonium measured concentrations. While, actually, reactions (RS1) and (RS2) would be responsible for negative ammonium artefacts, reaction (RS5) would involve absorption of ammonia on aqueous droplets with positive artefacts. The competition between (RS1), (RS2) and (RS5) reactions will strongly depend on meteorological conditions directly related to RH, RHD and temperature but also on chemical species abundances. According to Pathak and Chan (2005) in ammonium reach environments, reaction (RS5) would be responsible on highly contributing artefacts in $NH_4^+$ and $H^+$ distribution, especially due to acidity neutralisation by the existent ammonia. In their work, Arsene et al. (2011) proposed a detailed rationale for a potential estimation of the unmeasured $NH_4^+$ fraction induced most probable by the reactions inducing negative artefacts and significantly occurring during sampling. In Galon-Negru et al. work this unmeasured $NH_4^+$ has been called "missing $NH_4^+$" and for its estimation Arsene et al. (2011) rationale has been used.

**Table S 3**: Detailed statistics of linear regression analysis for $\sum_{cations}$ and $\sum_{anions}$ determined when estimated missing $NH_4^+$ has been taken into account, in the $PM_{2.5}$ and $PM_{10}$ fractions over the total investigated period, cold and warm seasons, in Iasi, north-eastern Romania.

| | PM$_{2.5}$ | | | | PM$_{10}$ | | | |
|---|---|---|---|---|---|---|---|---|
| | $\sum_{cations}/\sum_{anions}$ ratio | r | p-value | Confidence level (%) | $\sum_{cations}/\sum_{anions}$ ratio | r | p-value | Confidence level (%) |
| Total period | 0.95 | 0.98 | < 0.001 | 99.9 | 0.95 | 0.98 | < 0.001 | 99.9 |
| Cold seasons | 0.97 | 0.99 | < 0.001 | 99.9 | 0.98 | 0.99 | < 0.001 | 99.9 |
| Warm seasons | 0.87 | 0.96 | < 0.001 | 99.9 | 0.86 | 0.96 | < 0.001 | 99.9 |

Note: r–Pearson coefficient

**Section S 5**

**Additional information about ISORROPIA-II runs performed in the present work**

[revised manuscript text omitted]

**Section S 7**

In the Introduction section of Galon-Negru et al. work it has been already stated that "Particles acidity might influence transition metals solubility and enhance aerosols toxicity and atmospheric nutrient delivered through atmospheric deposition in marine areas (Meskhidze et al., 2003; Fang et al., 2017).". Particles pH is known to affect the solubility of trace metals

(such as Fe) found in aerosols (Meskhidze et al., 2003; Fang et al., 2017), with lower pH dissolving metal oxides and converting them to soluble metal sulfates (Oakes et al., 2012), conditions which would significantly change the aerosol environmental impacts. As presented in Guo et al., (2016), while at global level metal mobility will mainly affect nutrient distributions with important impacts on productivity, carbon sequestration, and oxygen levels in the ocean, at regional scales soluble transition metals seem mainly to enhance aerosol toxicity or their oxidative potential. Through $SO_4^{2-}$ contributions to aerosols acidity and the historical record of associations between so-called particle "strong acidity" and adverse health effects, Fang et al. (2017) suggest that $SO_4^{2-}$ linkages to health are most probably determined by $SO_4^{2-}$ role in acid dissolution of primary metals commonly found in ambient particles, while Tsagkogeorgas et al. (2017) suggest that on a regional to global scale the acidification of fresh water and forest ecosystems is most probable caused by wet and dry deposition of $SO_2$ and $SO_4^{2-}$ particles.

Epidemiological studies have often reported adverse health outcomes associated with strong aerosols acidity (Dockery et al., 1996; Gwynn et al., 2000; Lelieveld et al., 2015). In "Measuring aerosol damage to the atmosphere" (http://ec.europa.eu/research/infocentre/article_en.cfm?artid=31296, 28 October 2013), "Arsene says the high aerosol levels are probably linked to the high rate of various pulmonary diseases registered at the Clinic of Pulmonary Diseases in Iasi. These include cases like chronic obstructive pulmonary diseases (COPD), pneumonia, asthma (allergy, rhinitis), bronchiectasis (sometimes associated with bacterial infection) and tuberculosis often detected in the area.". The claim is not a result of a systematic work on an epidemiological study in Iasi, north-eastern Romania, but it represents a general observation of Arsene after performing research work with specialists and medical doctors in the field of pulmonary diseases (Cernat et al., 2011).

Moreover, upon our knowledge there is a single report, mentioned by Arsene et al. (2007), referring to indirect evidences of acid rain impacts in Iasi region, north-eastern Romania. In the performed study, Arsene et al. (2007) calculated high wet deposition fluxes for ions of anthropogenic origin in the $SO_4^{2-} > NO_3^- > NH_4^+$ order. The determined values, although among the highest, were in generally in good agreement with values reported for other north European sites (such as those for Poland, the Czech Republic, etc.). In the analysed raining events, air masses crossing very large continental areas (N and E sectors) brought the highest contribution in terms of fluxes for ionic species such as $Ca^{2+}$ while the NW and W sectors brought more significant contributions for anthropogenic related elements ($SO_4^{2-}$, $NO_3^-$, $NH_4^+$). Arsene et al. (2007) suggested that in the rainwater, higher fluxes of these elements for the NW and W sectors, when compared with other geographical sectors, was most probably also a result of important local influences, as the sampling site was located east side of the Oriental Carpathians chain that could constrain, to some extent, the pollution plume transported from western Europe. Arsene et al. (2007) report that the pH of the analysed rainwater events was 5.92 (volume weighted mean average, VWM) suggesting a sufficient load of alkaline components neutralizing rainwater acidity. Moreover, the authors suggested that on average, 97 % of the acidity in the collected rainwater samples was mostly neutralized by $CaCO_3$ and $NH_3$.

**Section S 8**

Regarding meteorological parameters, i.e. RH and temperature, in Galon-Negru et al. work is shown that over the cold season's high RH values and low temperatures were prevailing while over the warm seasons these parameters showed opposite trends. In the work of Ianniello et al. (2011), related to chemical characteristics of inorganic ammonium salts in $PM_{2.5}$ in the atmosphere of Beijing (China), the relationship between partitions of specific nitrogen-containing inorganic species towards various parameters has been investigated. The prevailing conditions over the two investigated time-periods, i.e. 23 January – 14 February 2007 and from 2 to 34 August 2007, showed higher RH values over the summer (high temperatures) and lower RH values over the winter (low temperatures).

Ianniello et al. (2011) treats in a very complex manner the relationship existent between particulate species such as $SO_4^{2-}$, $NO_3^-$, $Cl^-$ and $NH_4^+$, and possible partition routes in the $NH_3/NH_4^+$ system. Gas-to-particle conversion (photochemical processing) or heterogeneous processes are assigned as important sources contributing to particulate $NO_3^-$ abundances in the atmosphere. The authors claim that while $NO_2$ conversion to $NO_3^-$ through photochemical processing during the winter season is expected to contribute to $NO_3^-$ abundance over this period of the year, the heterogeneous formation (through condensation or absorption of $NO_2$ in moist aerosols) generally relates to relative humidity and the particulate atmospheric loading (Ianniello et al., 2011). The existence of large amounts of particulate $NO_3^-$, observed in summer by Ianniello et al., 2011, was considered as unexpected since $NH_4NO_3$ is semi-volatile and tends to dissociate and remain in the gas phase under high temperatures. Ianniello et al., (2010) report 6 times higher $NH_3$ concentrations in summer than in winter, with temperatures ranging from 1 to 14 $^oC$ in winter and 22 to 35 $^oC$ in summer periods. Under these circumstances high concentrations of fine particulate $NO_3^-$ in summer period were attributed to the existence of higher concentrations of $NH_3$ in the atmosphere, available to neutralise not only $H_2SO_4$ but also $HNO_3$ from the atmosphere. In addition, at the high RH values (daily mean in the 35 to 90 % range) reported in Ianniello et al. (2011) study significant fraction of $HNO_3$ and $NH_3$ were considered to be dissolved in humid particles, with enhanced distribution of fine particulate $NO_3^-$ and $NH_4^+$ in the atmosphere.

In Galon-Negru et al. work particulate $NH_4^+$ showed higher values during the cold seasons when compared to the warm seasons and its variation coincide with those of fine particulate $NO_3^-$ and $Cl^-$. Although such behaviour would indicate that $NH_4^+$ was largely originating from the neutralization between ammonia and acidic species such as $HNO_3$ and $HCl$, as shown under Subsection 3.2.3 (Stoichiometry of $(NH_4)_2SO_4$, $NH_4NO_3$ and $NH_4Cl$), the most important contribution was most probably brought by $H_2SO_4$ species. However, the very similar pattern observed for $NO_3^-$, $Cl^-$ and $NH_4^+$ suggest that these species were representative for likely internally mixed particles and came most probably from similar gas-to-particle processes (Huang et al., 2010). Galon et al. report for $SO_4^{2-}$ comparable concentrations during both the summer and winter seasons, and Backes et al. (2016a) suggest that the formation of $SO_4^{2-}$ particles is not limited by $NH_3$ in any season. During the warm seasons, higher temperatures and solar radiation intensity, enhancing the photochemical activity and the atmospheric oxidation potential, is also enhancing the oxidation rate of $SO_2$ to particulate $SO_4^{2-}$, but during the cold seasons wood burning might become an important source of $SO_4^{2-}$.

At high RH values, and especially for deliquescent particles, there is suggestion that most of the fine particulate $NO_3^-$ exists as an internal mixture with $SO_4^{2-}$, so that $HNO_3$ can be easily absorbed into the droplets, a process considerably reducing the thermodynamic dissociation constant for $NH_4NO_3$ (Zhang et al., 2000; Ianniello et al., 2011). Under these circumstances it is supposed that fine particulate $NO_3^-$ can be formed from $HNO_3$ and $NH_3$ through heterogeneous reactions on fully neutralised fine particulate $SO_4^{2-}$ (a process which has been taken into account also in Galon et al. work, under the Subsection 3.2.3). In Galon et al. work significant correlation has been observed between $SO_4^{2-}$ and $NO_3^-$. These two measured parameters showed significant correlation with the RH, with high concentrations in $SO_4^{2-}$ or $NO_3^-$ being formed at high RH values. Such behaviour would allow us suggesting that $NO_3^-$ is being produced on preexisting $SO_4^{2-}$ aerosols, which could provide sufficient area and aerosol water content for the heterogeneous reaction to occur should be available.

Markovic et al. (2011) show that at high RH, the amount of the gaseous precursors, such as $NH_3$ and $HNO_3$, have relatively little influence on the formation of fine particulate nitrate. Ianniello et al. (2011) concluded also that in summer period, with high prevailing RH values, almost on all days the meteorological conditions were favourable for the formation of $NH_4NO_3$ at Beijing site.

In Galon et al. work, regarding gaseous $NH_3$ values such as those derived from ISORROPIA, it should be pointed out that its estimated concentration was much higher than that of $HNO_3$ and $HCl$ in gas phase. During the cold seasons the $[NH_3]/([HNO_3] + [HCl])$ ratio is as high as $2.0 \pm 0.6$ for RH values < 40 %, $3.6 \pm 2.0$ for RH values in the 40–60 % range, and $3.0\pm1.4$ for RH values > 60 % RH. During the warm seasons, for similar RH groups, the ratio takes the $4.9 \pm 1.9$, $4.6 \pm 1.4$ and $9.1.0 \pm 6.2$ values. Ianniello et al. (2011) report a $[NH_3]/([HNO_3] + [HCl])$ ratio of $27.90 \pm 12.70$ for winter season and of $54.06 \pm 20.60$ for the summer period. The authors are finally claiming that the atmosphere of their interest location was ammonia-rich in gas phase over both investigated seasons.

Moreover, for the specific conditions referring to Galon et al. work, the authors have observed that by analysing the events for similar RH groups, but for cold and warm seasons separately, the $NH_3/NH_4^+$ partition didn't enhance significantly. During the cold seasons over the three RH investigated groups, the fractions of the $NH_3$, $HCl$ and $HNO_3$ present in the gaseous form have taken in average the $71.4 \pm 4.7$, $4.4 \pm 0.3$ and $24.0 \pm 4.8\%$ values. Over the warm seasons for the previously mention species the fractions present in the gaseous form have taken in average the $83.1 \pm 3.4$, $6.6 \pm 0.6$ and $10.3 \pm 3.4\%$ values.

**Section S 9**

Backes et al. (2016a,b) present in some of their publication seasonal distributions in $NH_3$ emissions under static- and dynamic-time profile (STP and DTP) scenarios. While the STP scenario lacks dynamic, meteorology dependent or specific differences in policies or intensity of animal husbandry, the DTP scenario takes into account potential meteorological variables (such as wind speed and surface temperature) influence on the distribution of the annual emissions. However, while under the STP scenario, the annual time series of $NH_3$ showed one annual peak in spring (March), in the DTP scenario $NH_3$ concentrations show two annual peaks, one in spring (May) and one in autumn (September). The authors underline that

implementation of the DTP resulted in a shift from seasonal average winter NH₃ emissions to summer emissions. According to Backes et al. (2016a), high NH₃ concentrations are known to appear in proximity to emission sources, due to its low atmospheric lifetime which results in NH₃ concentration levels in the atmosphere which closely follow the seasonal emission trend.

[Figure]

(a)                                                               (b)

**Figure S 1**: Time series in NH₃ concentrations derived from ISORROPIA-II runs for the 2016 data-base, both for $PM_{2.5}$ and $PM_{10}$ scenarios in Iasi, north-eastern Romania.

Moreover, Sutton et al. (2013) claim that together with increased anthropogenic activity, global NH₃ emissions may increase from 65 (48–85) Tg N in 2008 to about 132 (89–179) Tg N by 2100. Most NH₃ emissions are known to result from agricultural productions strongly influenced by climatic interactions. There is, however, recognition about the fact that most of the up to now used approaches failed in recognising that a warm dry-year would tend to give larger NH₃ emissions than a cold-wet year. Estimates in the global NH₃ emission made by Sutton et al., (2013) indicate agricultural soils and crops, including emissions from grazing and land application of animal manure, as a first ammonia ranked source with 28.3 Tg N $yr^{-1}$ contribution. This is immediately followed by excreta from domestic animals (8.7 Tg N $yr^{-1}$), oceans and volcanoes (8.6 Tg N $yr^{-1}$), biomass burning (5.5 Tg N $yr^{-1}$), waste composting and processing (4.4 Tg N $yr^{-1}$), and by other sources with contributions of < 3.3 Tg N $yr^{-1}$.

Details presented in FIG. S 1 clearly show that for Iasi, north-eastern Romania, yearly NH₃ distributions as derived from ISORROPIA-II model present clear maxima in February and December in a behaviour suggesting important local sources contributions over the winter season. It might be that this distribution follows actually a trend mainly controlled by the agricultural practices in rural areas surrounding Iasi, mainly related to the timing of manure spreading on agricultural soils. Moreover, in Iasi and in the nearby counties, the contribution brought by open (sheep and goat) barns, operating over the cold season mainly in a "hot-spot" mode, might increase as importance. Manure storage from local and regional closed

barns (pigs and poultry) might also bring important contributions over the winter. Although biomass burning might bring important contribution in terms of global ammonia emissions it is believed that at the interest location this source is overwhelmed by agricultural practices in the area.

**Table S 4**: Detailed statistics of linear regression analysis for major water soluble ions determined in the $PM_{2.5}$ and $PM_{10}$ fractions over the total investigated period in Iasi, north-eastern Romania.

| | PM fraction | Slope | Pearson coefficient (r) | p-value | Confidence level (%) |
|---|---|---|---|---|---|
| $Cl^-$ vs. RH | $PM_{2.5}$; $PM_{10}$ | + | > 0.71 | 0.010 | 95.4 |
| $Cl^-$ vs. T | $PM_{10}$ | - | > 0.59 | 0.045 | 95.4 |
| $Cl^-$ vs. PM | $PM_{2.5}$; $PM_{10}$ | + | > 0.70 | 0.038 | 95.4 |
| $Cl^-$ vs. MLD | $PM_{2.5}$; $PM_{10}$ | - | > 0.69 | 0.012 | 95.4 |
| $NO_3^-$ vs. RH | $PM_{2.5}$; $PM_{10}$ | + | > 0.84 | < 0.001 | 99.9 |
| $NO_3^-$ vs. T | $PM_{2.5}$; $PM_{10}$ | - | > 0.94 | < 0.001 | 99.9 |
| $NO_3^-$ vs. MLD | $PM_{2.5}$; $PM_{10}$ | - | > 0.84 | < 0.001 | 99.9 |
| $NO_3^-$ vs. PM | $PM_{2.5}$; $PM_{10}$ | + | > 0.70 | 0.021 | 95.4 |
| $SO_4^{2-}$ vs. RH | $PM_{2.5}$; $PM_{10}$ | + | > 0.44 | 0.177 | 68 |
| $SO_4^{2-}$ vs. T | $PM_{2.5}$; $PM_{10}$ | - | > 0.41 | 0.214 | 68 |
| $SO_4^{2-}$ vs. PM | $PM_{2.5}$; $PM_{10}$ | + | > 0.45 | 0.162 | 68 |
| $NH_4^+$(total) vs. RH | $PM_{2.5}$; $PM_{10}$ | + | > 0.80 | 0.001 | 95.4 |
| $NH_4^+$(total) vs. T | $PM_{2.5}$; $PM_{10}$ | - | > 0.94 | < 0.001 | 99.9 |
| $NH_4^+$(total) vs. MLD | $PM_{2.5}$; $PM_{10}$ | - | > 0.80 | 0.001 | 95.4 |
| $NH_4^+$(total) vs. PM | $PM_{2.5}$; $PM_{10}$ | + | > 0.70 | 0.023 | 95.4 |

Note: RH–relative humidity; T–temperature; MLD–mixing layer depth; PM–particle loading;

**Table S 5**: Detailed statistics of linear regression analysis for molar concentrations of particulate $NH_4^+$ and $SO_4^{2-}$ in $PM_{10}$ fraction over the cold and warm seasons in Iasi, north-eastern Romania.

| | $NH_4^+$ from raw IC data | | | $NH_4^+$(total) | | |
|---|---|---|---|---|---|---|
| | Pearson coefficient (r) | p-value | Confidence level (%) | Pearson coefficient (r) | p-value | Confidence level (%) |
| Cold seasons | 0.97 | < 0.001 | 99.9 | 0.96 | < 0.001 | 99.9 |
| Warm seasons | 0.98 | < 0.001 | 99.9 | 0.97 | < 0.001 | 99.9 |

**Section S 10**

The data set from the present work has been investigated through Radar and/or Pie charts in order to identify the potential contributions brought by various sectors long-range transport to the atmospheric PM burden in the area. It should be however emphasized that the PM chemical composition appeared to be mainly driven by the air mass origin and type (e.g., continental, regional), meteorologically-controlled air mass characteristics (buoyancies), geographical context (the Oriental Carpathians chain, with the highest altitude of 1907 m, is facing Iasi ~ 150 km north-westerly distance), etc. As presented in FIG. S 2a, in overall, in terms of sectors contributions the N-NE up to the S (clockwise round) area seems to bring the most important fraction to both the $PM_{2.5}$ and $PM_{10}$ abundances. Higher loaded events in terms of aerosols mass concentration

were those associated with long-range continental transport, Saharan dust-events, regional transport from more arid areas (S-SE) and buoyancy affected air masses.

Regarding potential local anthropogenic contribution to PM atmospheric burden it should be emphasized that in the immediately nearby area of the sampling point neither important commercial areas nor industrial units are operating and traffic should be the most important contributor. Actually in Iasi, in the post-communist period, the industry has almost completely ceased down and presently the Antibiotic company (7 km from the sampling point, western, W, direction), a small brick plant (8 km, SE) and the heat and power plant (5 km, SE) may account as for the most significant local anthropogenic contributors. However, Iasi is known as an important Romanian university centre (> 42000 students by 2015) with intense activity over university semesters. Traffic in the nearby area and commercial activity in the top hot-spots area of Iasi get more intense over these periods of the year. As for the rural areas surrounding Iasi, or from the nearby counties, farming activities are mainly related to cereal crops, plantation agriculture, open and closed animal barns (cattle, sheep and goat, pigs, poultry).

[Figure]

(S 2a)                                                    (S 2b)

**Figure S 2**: Averaged annual sectors contributions both in the $PM_{2.5}$ (S 2a) and $PM_{10}$ (S 2b) fractions.

| | Winter | Spring | Summer | Autumn |
|---|---|---|---|---|
| PM$_{2.5}$ | | | | |
| Fo | | | | |
| Cl$^-$ | | | | |
| HCO$_3^-$ | | | | |
| Ac | | | | |
| NO$_3^-$ | | | | |
| SO$_4^{2-}$ | | | | |
| Ox | | | | |
| NH$_4^+$ | | | | |
| Na$^+$ | | | | |

[Figure]

**Figure S 3**: Particulate inorganic and organic ions seasonal contributions (in the PM$_{2.5}$ fraction) associated with the different identified air mass.

[Figure]

PM$_{2.5}$

PM$_{2.5}$

PM$_{10}$

PM$_{10}$

PM$_{2.5}$

PM$_{2.5}$

PM$_{10}$

PM$_{10}$

[Figure]

[Figure]

[Figure]

[Figure]

[Figure]

[Figure]

[Figure]

[Figure]

PM$_{2.5}$

PM$_{2.5}$

PM$_{10}$

PM$_{10}$

PM$_{2.5}$

PM$_{2.5}$

PM$_{10}$

PM$_{10}$

[Figure]

[Figure]

[Figure]

**Figure S 4**: Trajectory air mass associated percentual contributions for the identified-quantified ions and ISORROPIA-II predicted gaseous $NH_3$, $HNO_3$ and $HCl$ both in $PM_{2.5}$ (top) and $PM_{10}$ (bottom) fractions. Selected investigated events are presented. Note: Fo – formate, Ac – acetate, Ox – oxalate, $NH_4^+$ reflects $NH_4^+_{total}$.

It should be also underlined that for ionic species specific for the coarse fraction, i.e. $Na^+$ and $Ca^{2+}$, the maxima seems to be mainly induced either by the prevalent direction of the air masses (spring maxima of $Na^+$ ion most probably induced by the fact that March and April appeared as months with prevalent southern air masses crossing large continental areas with more salty soil) or by the drought (the summer maxima of $Ca^{2+}$ was mainly influenced by the ionic chemical composition of aerosol samples collected in August 2016, 7 events, a month of the 2016 year without any single raining events). Actually, in August 2016, 1 event was strongly affected by atmospheric driven buoyancy (18.2 % fine particulate $Ca^{2+}$ and 42.1 % fine particulate $HCO_3^-$), 1 event was affected by turbulent local air masses (10.7 % fine particulate $Ca^{2+}$ and 24.8 % fine particulate $HCO_3^-$) and other 3 events showed contributions brought by air masses crossing larger continental areas ($9.8 \pm 1.7$ % fine particulate $Ca^{2+}$ and $25.3 \pm 3.9$ % fine particulate $HCO_3^-$). Other 2 events sampled at the beginning of August 2016 (of N-NW direction both) in terms of $Ca^{2+}$ and $HCO_3^-$ brought contributions as high as ($4.3 \pm 0.6$ % fine particulate $Ca^{2+}$ and $9.7 \pm 2.3$ % fine particulate $HCO_3^-$). High percentual contributions within the identified and quantified species brought by $Ca^{2+}$ and $HCO_3^-$ ions were also observed in various events collected all year round which were affected by long-range transport air masses accompanied by atmospheric driven buoyancy or by long-range transport air masses coming from arid areas (e.g. 31 January 2016, 14 February 2016, 20 March 2016, 06 April 2016, 09 April 2016, etc). It has been observed that in average the contributions brought by such events were strongly affecting the distribution of the ionic chemical composition in sampled aerosols.

---

## Author Comment (AC2)

Response to reviewers
Manuscript: ACP-2017-1030
Manuscript title: Chemical characteristics of size resolved atmospheric aerosols in Iasi, north-eastern Romania. Nitrogen-containing inorganic compounds controlling aerosols chemistry in the area

The discussion below includes the complete text from the reviewer (italic), along with our responses to the specific comments and the corresponding changes made to the revised manuscript. All line numbers refer to the original manuscript.

Response to **Reviewer #2 Comments**
*General*
*This is a comprehensive field and analytical study of aerosol particles sampled under size-resolution in north-eastern Romania. It contains a wealth of data on the aerosol chemical composition in this area where not many data are available for comparison which is generally true for Eastern Europe. The whole manuscript is a bit lengthy and it might be good if the authors could explore potential for shortening some sections of the paper.*

*Overall, I think the manuscript requires only minor modification and can then be published in ACP.*

We are grateful for the reviewer's constructive comments and suggestions and the manuscript has been revised accordingly. Reviewer's concerns and suggestions are addressed here below and we would like also to point out the following:
1) A Supplement Material is accompanying now the manuscript (attached to the present response).
2) The manuscript is now presented in a more concise manner (reduction from 53 pages to 46 pages) (the revised manuscript is attached to the present response).
3) In the revised manuscript amendments related to reviewer #2 concern are highlighted in green colour.
4) The revised manuscript contains 4 additional references.
5) Considering language-related suggestion from one of the reviewers, we have requested a well known English speaking scientist, Prof. Davide Vione from University of Torino, to offer us support in polishing the composition. Language polishing has resulted in a considerably improvement of the manuscript quality.
6) In the revised manuscript all language-related changes are highlighted in blue colour. Blue colour reflects: *i*) corrected grammar errors, *ii*) the place where cuts occurred within former sentences, *iii*) extra-words used in the revised manuscript, *iv*) full sentences addressing more succinctly ideas from the original manuscript.
7) We kindly ask the editorial office to accept our proposal colour-related since working in the track-changes mode significantly affects the readability of the manuscript.

In the revised manuscript the title is proposed in the form
Chemical characteristics of size resolved atmospheric aerosols in Iasi, north-eastern Romania: Nitrogen-containing inorganic compounds control aerosols chemistry in the area

*Details*
*Page 3. line 3: '... great concern of interest....' - Maybe better '...great concern and of interest....'*
*Please check the English throughout the manuscript again, possibly but a native speaker or an English language editing service.*

As raised under point 5 of the present answer, in the revised form of the manuscript, language problems have been checked. Reviewers suggestion have been also taken into account. In the revised manuscript text composition is significantly improved, mainly as a result of sentences shortening with "sharper" meaning.

*Section 2.2: Can anything be said how reliable the sampling of the very small size fractions by the DLPI is? Has this been characterised?*

In the manuscript under Section 2.2 after revision we have
"The DLPI unit performs aerosol size classification in 13 specific fractions (with size cuts at 0.0276, 0.0556, 0.0945, 0.155, 0.260, 0.381, 0.612, 0.946, 1.60, 2.39, 3.99, 6.58 and 9.94 µm at 50 % calibrated aerodynamic cut-point diameters and at 21.7 $^{o}$C, inlet pressure 1013.3 mbar, outlet pressure 100 mbar and 29.85 L min$^{-1}$ flow rate)."

However, these aspects should be corroborated with details presented now in Section S1 of the supplement material (SM) where we give:

Section S 1
"Possible explanations for the differences/similarities between afore mentioned studies might be seen as a result of the contributions brought from various factors. As shown below the differences in the PM mass concentrations reported in Galon et al. and those reported in Arsene et al. (2011) might be only slightly controlled by changes in the particle size cut-off and/or by the difference in sampling site altitude (25 m in Galon et al. vs. 35 m in Arsene et al. (2011)) which is too small to account for the difference in PM concentrations.

In Arsene et al. (2011) study, aerosol sampling has been undertaken by using a stacked filter unit (SFUs) consisting of a 8.0 µm pore size 47-mm diameter Isopore polycarbonate filter mounted in front of a 0.4 µm pore size 47-mm diameter Isopore filter. According to details in Arsene et al. (2011) and references included therein, the 50% cut-point diameter (D50) of the 8.0 µm pore size filter was estimated to be of the order on $1.5 \pm 0.2$ µm aerodynamic equivalent diameter (AED). Consequently, particles collected on the 8 µm pore size filters, with a diameter larger than 1.5 µm AED, were referring to the aerosol coarse fraction, while the particles collected on the 0.4 µm pore size filters were attributed to particles with a diameter below 1.5 µm AED. In Galon et al. work aerosols samples were collected on 25 mm in diameter ungreased aluminum filters using a cascade Dekati Low-Pressure Impactor (DLPI), operating at a flow rate of 29.85 L min$^{-1}$. Enhanced sampling efficiency of the DLPI unit used in Galon et al. work, with regard to fine and ultrafine particles, has been observed in comparison with the SFU unit reported by Arsene et al. (2011). Parallel DLPI and SFU sampling runs (performed within January–July 2016) showed that the DLPI unit could collect in average with ~ 6 µg m$^{-3}$ more particles than the SFU system and this observation allowed us suggesting that most probably the operational limits of the SFU system with regard to fine and ultrafine particles could suppress to some extend the values reported by Arsene et al. (2011).

The difference observed between the PM$_{10}$ mass concentration, i.e. $18.9 \pm 9.3$ µg m$^{-3}$, reported by Galon et al. and the PM$_{total}$ mass concentration, i.e. $38.3 \pm 25.4$ µg m$^{-3}$, reported by Arsene et al. (2011) might actually be the result of sampling in different years with emission sources of various prevalence. Implementation of environmental quality management systems in various sectors with anthropogenic activity may account for that (e.g. attempts given by the local administrative sector in settling down soil- or road-resuspended dust especially over the warm seasons by wetting the roads, protecting areas with intense building activities, etc.).

Regarding other literature available values, in a study from 2016, Alastuey et al. (2016) report for PM$_{10}$ concentrations in Moldova (the closest point to our sampling site), values as high as ~ 25

µg m$^{-3}$ over summer and of ~ 25–30 µg m$^{-3}$ over winter period, yielding an annual averaged value of ~ 27.5 µg m$^{-3}$ which is much higher than the 18.9 µg m$^{-3}$ value reported in the present work, for Iasi, north-eastern Romania. However, the elevated concentrations observed at the eastern sites were attributed to regional or local sources (Alastuey et al., 2016). In Alastuey et al. (2016) the author reports only the altitude above sea level (i.e. 156 m for Moldova's location) and the sampling a.g.l. is not mentioned. It might be that in Alastuey et al. (2016) study if the Moldavian sampling location was assigned as an EMEP site than the sampling a.g.l. was settled in agreement with an EC Directive recommending an a.g.l. of ~ 4 or 8 m. It is believed that the difference observed between Galon et al. site (i.e. 18.9 ± 9.3 µg m$^{-3}$) in comparison with other European sites might actually reflect that sampling altitude would be an important controlling factor to the atmospheric aerosol burden at a site.

In Iasi, north-eastern Romania, mass concentrations in the 5 to 10 µg m$^{-3}$ range, for the PM$_{10}$ and PM$_{2.5}$ fractions were especially observed in samples collected after raining events (i.e., May, June, August, and October). Such behaviour would be expected since particles from the atmosphere might be efficiently removed by precipitation (Arsene et al., 2011). However, during events with strong natural or anthropogenic contributions the PM$_{10}$ and PM$_{2.5}$ mass concentrations exceeded the averages observed at AMOS. For example, an event collected in April 2016, from 9$^{th}$ to 11$^{th}$, with 34.3 µg m$^{-3}$ in PM$_{2.5}$ and 43.9 µg m$^{-3}$ in PM$_{10}$, was actually highly influenced by the long range transport phenomena of African dust and marine aerosols from the Black and Aegean/Mediterranean seas (i.e., event described in detail later in the text). For other events, the exceeding fine fraction mass concentrations are believed to be a result of more variable sources (combustion, biogenic, local mineral dust, meteorological factors, etc.)."

*Table 2: Maybe the names of the sampling locations from the EMEP measurements can be given.*

The names of the sampling locations from the EMEP measurements (Alastuey et al., 2016) are given as a Note under Table 2.
SE12 – Aspvreten (Sweden)
IE321 – Mace Head (Ireland)
FR09 – Revin (France)
DE44 – Melpitz (Germany)
SK06 – Starina (Slovak Republic)
HU02 – K-Puszta (Hungary)
MD13 – Leova II (Moldavian Republic)
ES22 – Montsec (Spain)
IT01 – Montelibretti (Italy)
GR02 – Finokalia (Greece)

*P9, l13: ...Figure... This should be capitalized all over the manuscript.*

 "Figure…" or "Fig." has been replaced with "FIGURE" or "FIG." in the entire manuscript but the final decision is to be taken by the editorial office.

*P15, l21: Is the comparison to the Kanpur measurement helpful here? I cannot fully understand what CaCO3 has to do with nitrate.*

In order to answer referee's question we have included in the revised manuscript:
"Higher abundances of particulate NO$_3^-$, SO$_4^{2-}$, NH$_4^+$ and K$^+$ in winter compared to summer are reported for other European (Schwarz et al., 2012; Voutsa et al., 2014) and non-European sites as

well (Sharma et al., 2007). Sharma et al. (2007) also suggest a potential role of $CaCO_3$ in controlling particulate $NO_3^-$ abundance in Kanpur, India."

This aspect is actually completed by the information given in the manuscript in regard with the potential effect played by dust in the distribution of particulate $NO_3^-$ with a clear shift from the fine- to the coarse-mode. For Iasi, north-eastern Romania we have actually identified that $MgCO_3$ and not $CaCO_3$ will be involved in the formation of the coarse mode $NO_3^-$.

*P27, l23: But shouldn't biomass burning have a seasonal pattern?*

In order to answer referee's question we have reformulated this aspect in the revised manuscript:
"The size distributions of particulate $K^+$ reflect the occurrence of one dominant fine mode (with maxima at 381 nm) during both the cold and the warm seasons, and of a second less important mode during the warm seasons (with maxima in the 0.946–1.6 µm range). Such behaviour most likely reflects contributions from biomass burning all over the year (Schmidl et al., 2008; Pachon et al., 2013)."

However, potential implications of the biomass burning process towards $K^+$ seasonal pattern (i.e. $K^+$ as a tracer of the biomass burning process) has been addressed in more details within the discussion on FIG. 6e.

*P28, l30: Please check sentence: '... such as to.....*

Language editing problems are entirely addressed within the revised manuscript.

**References**

[revised manuscript text omitted]

**Supplement material (SM) for the manuscript https://doi.org/10.5194/acp-2017-1030**

**Chemical characteristics of size resolved atmospheric aerosols in Iasi, north-eastern Romania. Nitrogen-containing inorganic compounds controlling aerosols chemistry in the area**

Alina Giorgiana Galon-Negru[1], Romeo Iulian Olariu[1,2], Cecilia Arsene[1,2]

[1]"Alexandru Ioan Cuza" University of Iasi, Faculty of Chemistry, Department of Chemistry, 11 Carol I, 700506 Iasi, Romania
[2]"Alexandru Ioan Cuza" University of Iasi, Integrated Centre of Environmental Science Studies in the North Eastern Region, 11 Carol I, Iasi 700506, Romania

*Correspondence to*: Cecilia Arsene (carsene@uaic.ro); Phone number: +40-232-201354; Fax: +40-232-201313; Postal address: Cecilia Arsene, "Alexandru Ioan Cuza" University of Iasi, Faculty of Chemistry, Department of Chemistry, 11 Carol I, 700506 Iasi, Romania

**Table S 1**: Detailed statistics for the $PM_{10}$ and $PM_{2.5}$ fractions mass concentrations determined over the investigated period (n = 84 sampling events) in Iasi, north-eastern Romania.

| Applied test | $PM_{2.5}$ | | $PM_{10}$ | |
|---|---|---|---|---|
| | p-value | Confidence level (%) | p-value | Confidence level (%) |
| Shapiro-Wilk normality test | 0.546 | 95 | 0.682 | 95 |
| t-test | 0.987 | 95 | 0.998 | 95 |

**Section S 1**

Possible explanations for the differences/similarities between afore mentioned studies might be seen as a result of the contributions brought from various factors. As shown below the differences in the PM mass concentrations reported in Galon et al. and those reported in Arsene et al. (2011) might be only slightly controlled by changes in the particle size cut-off and/or by the difference in sampling site altitude (25 m in Galon et al. vs. 35 m in Arsene et al. (2011)) which is too small to account for the difference in PM concentrations.

In Arsene et al. (2011) study, aerosol sampling has been undertaken by using a stacked filter unit (SFUs) consisting of a 8.0 μm pore size 47-mm diameter Isopore polycarbonate filter mounted in front of a 0.4 μm pore size 47-mm diameter Isopore filter. According to details in Arsene et al. (2011) and references included therein, the 50% cut-point diameter (D50) of the 8.0 μm pore size filter was estimated to be of the order on 1.5 ± 0.2 μm aerodynamic equivalent diameter (AED). Consequently, particles collected on the 8 μm pore size filters, with a diameter larger than 1.5 μm AED, were referring to the aerosol coarse fraction, while the particles collected on the 0.4 μm pore size filters were attributed to particles with a diameter below 1.5 μm AED. In Galon et al. work aerosols samples were collected on 25 mm in diameter ungreased aluminum filters using a cascade Dekati Low-Pressure Impactor (DLPI), operating at a flow rate of 29.85 L min$^{-1}$. Enhanced

sampling efficiency of the DLPI unit used in Galon et al. work, with regard to fine and ultrafine particles, has been observed in comparison with the SFU unit reported by Arsene et al. (2011). Parallel DLPI and SFU sampling runs (performed within January–July 2016) showed that the DLPI unit could collect in average with ~ 6 µg m$^{-3}$ more particles than the SFU system and this observation allowed us suggesting that most probably the operational limits of the SFU system with regard to fine and ultrafine particles could suppress to some extend the values reported by Arsene et al. (2011).

The difference observed between the PM$_{10}$ mass concentration, i.e. 18.9 ± 9.3 µg m$^{-3}$, reported by Galon et al. and the PM$_{total}$ mass concentration, i.e. 38.3 ± 25.4 µg m$^{-3}$, reported by Arsene et al. (2011) might actually be the result of sampling in different years with emission sources of various prevalence. Implementation of environmental quality management systems in various sectors with anthropogenic activity may account for that (e.g. attempts given by the local administrative sector in settling down soil- or road-resuspended dust especially over the warm seasons by wetting the roads, protecting areas with intense building activities, etc.).

Regarding other literature available values, in a study from 2016, Alastuey et al. (2016) report for PM$_{10}$ concentrations in Moldova (the closest point to our sampling site), values as high as ~ 25 µg m$^{-3}$ over summer and of ~ 25–30 µg m$^{-3}$ over winter period, yielding an annual averaged value of ~ 27.5 µg m$^{-3}$ which is much higher than the 18.9 µg m$^{-3}$ value reported in the present work, for Iasi, north-eastern Romania. However, the elevated concentrations observed at the eastern sites were attributed to regional or local sources (Alastuey et al., 2016). In Alastuey et al. (2016) the author reports only the altitude above sea level (i.e. 156 m for Moldova's location) and the sampling a.g.l. is not mentioned. It might be that in Alastuey et al. (2016) study if the Moldavian sampling location was assigned as an EMEP site than the sampling a.g.l. was settled in agreement with an EC Directive recommending an a.g.l. of ~ 4 or 8 m. It is believed that the difference observed between Galon et al. site (i.e. 18.9 ± 9.3 µg m$^{-3}$) in comparison with other European sites might actually reflect that sampling altitude would be an important controlling factor to the atmospheric aerosol burden at a site.

In Iasi, north-eastern Romania, mass concentrations in the 5 to 10 µg m$^{-3}$ range, for the PM$_{10}$ and PM$_{2.5}$ fractions were especially observed in samples collected after raining events (i.e., May, June, August, and October). Such behaviour would be expected since particles from the atmosphere might be efficiently removed by precipitation (Arsene et al., 2011). However, during events with strong natural or anthropogenic contributions the PM$_{10}$ and PM$_{2.5}$ mass concentrations exceeded the averages observed at AMOS. For example, an event collected in April 2016, from 9[th] to 11[th], with 34.3 µg m$^{-3}$ in PM$_{2.5}$ and 43.9 µg m$^{-3}$ in PM$_{10}$, was actually highly influenced by the long range transport phenomena of African dust and marine aerosols from the Black and Aegean/Mediterranean seas (i.e., event described in detail later in the text). For other events, the exceeding fine fraction mass concentrations are believed to be a result of more variable sources (combustion, biogenic, local mineral dust, meteorological factors, etc.).

**Section S 2**

Beside changes in sources contributions, changes in meteorological conditions (i.e. relative humidity (RH) and wind speed (WS)) might induce distinct behaviour in the seasonal distribution of the PM. While at the investigated site over the warm

seasons RH values varied in the range of ~ 30–50% and in the cold seasons in the range of ~ 45–80% we expect that at these RH values particles have proper meteorological conditions to grow. There is suggestion that in the 0 to 98% RH range, the number median diameter (NMD) may increase two-fold (from 0.018 to 0.036 μm) with the range of geometric standard deviation ($\sigma_g$) for the particle size distributions in the 1.53–2.06 range and with the larger number median diameter (NMD) having the smaller $\sigma_g$. Moreover, the authors suggest that the smaller particles grew proportionately more than the larger one (Sinclair et al., 1974).

As presented in FIG. 3 in Galon et al. work, in the 0.0556–0.946 μm particles size range, higher dM/dlogD$_p$ values over the cold seasons toward those specific for the warm seasons might actually reflect a cumulative effect induced by the contribution of the sub-micron growth particles due to higher RH values during the cold seasons. Shifts in particles diameter distribution toward higher values are expected in this case.

The observed 1.60–2.39 μm supermicrone mode is most probably a result of a more significantly contribution brought by large particles mainly associated with dust resuspension due to higher wind speeds. Although over the warm seasons averaged wind speed showed similar value to that specific for the cold seasons, monthly standard deviation toward the mean values suggested much larger dispersion. During the warm seasons, storms and rapid wind ghosts are known as often engaging important quantities of dust carrying especially particles of larger diameter.

**Table S 2**: Detailed statistics of linear regression analysis for $\sum_{cations}$ and $\sum_{anions}$ determined for raw ion chromatography data in the PM$_{2.5}$ and PM$_{10}$ fractions over the total investigated period, cold and warm seasons, in Iasi, north-eastern Romania.

| | PM$_{2.5}$ | | | | PM$_{10}$ | | | |
|---|---|---|---|---|---|---|---|---|
| | $\sum_{cations}/\sum_{anions}$ ratio | r | p-value | Confidence level (%) | $\sum_{cations}/\sum_{anions}$ ratio | r | p-value | Confidence level (%) |
| Total period | 0.69 | 0.94 | < 0.001 | 99.9 | 0.70 | 0.94 | < 0.001 | 99.9 |
| Cold seasons | 0.67 | 0.98 | < 0.001 | 99.9 | 0.68 | 0.98 | < 0.001 | 99.9 |
| Warm seasons | 0.84 | 0.87 | < 0.001 | 99.9 | 0.86 | 0.86 | < 0.001 | 99.9 |

Note: r–Pearson coefficient

**Section S 3**

The missing $HCO_3^-/CO_3^{2-}$ has been estimated as suggested by Arsene et al. (2007) while the rationale previously proposed by Arsene et al. (2011) has been used to estimate the missing $NH_4^+$. Within the $NH_4^+$(total) fraction (defined as the sum between that derived from raw IC data and the part estimated by using the rationale of Arsene et al. (2011)), the correction for the missing $NH_4^+$ accounted for about 21.65 ± 25.70 % for the warm seasons while for the cold seasons the correction accounted for about 46.05 ± 18.43 %. However, when estimated missing $NH_4^+$ has been taken into account a significant improvement in the overall ionic balance ($\sum_{cations}$ vs. $\sum_{anions}$) was observed for both the PM$_{2.5}$ and the PM$_{10}$ fractions (detailed statistics in Table S 3).

**Section S 4**

During sampling, perturbations in the gas to particle equilibrium may occur with evaporation of semi-volatile $NH_4NO_3$ and $NH_4Cl$ salts from the fine particles collected on the front filters and fluctuations during sampling in temperature, relative humidity and/or pressure drop across the filters might highly affect the measurements of these species especially in urban environments (Pathak and Chan, 2005; Ianniello et al., 2011). Moreover, the total concentration of ammonium salts in $PM_{2.5}$ is usually described as a sum of the measurements both for non-volatile (unevolved) and volatile (evolved) fine particulate species (Ianniello et al., 2011). Since in the present work during sampling neither denuders nor backup-filters were used, sampling artefacts of semi-volatile $NH_4NO_3$ and $NH_4Cl$ species are not completely excluded. Although presented data are reliable from the instrumental analysis point of view, limitations related to potential unmeasured species especially due to sampling procedure (during and after) should always be taken into account (i.e. potential $NH_4^+$, $NO_3^-$ and $Cl^-$ evolving from the filters as a result of $NH_4NO_3$ and $NH_4Cl$ dissociations).

In Galon-Negru et al. manuscript the completeness degree of the ionic balance for the identified and quantified species in both the $PM_{10}$ and $PM_{2.5}$ fractions was checked and quite often slopes lower than unity in the $\sum_{cations}$ vs. $\sum_{anions}$ dependences were observed in both cases indicating important cation deficit in the ionic balance. As mentioned in the manuscript, per sampled events either cation or anion deficit has been observed in various stages. For similar behavior in Arsene et al. (2011) a detailed rationale has been proposed in order to explain/estimate either missing anions or cations within the ionic budget (under subsections 3.3 Role of inter-particle and gas-particle interactions in establishing fine and coarse fractions chemical composition and 3.4 The ionic balance in the coarse and fine fractions of Arsene et al. (2011) paper). As given in Arsene et al. (2011), meteorological conditions favourable to generate deliquesced particles in the form of $NH_4NO_{3(aq)}$ and $NH_4Cl_{(aq)}$ may exist in Iasi, north-eastern Romania, with deliquesced particles being formed under conditions of relative humidity at deliquescence (RHD), a parameter which is defined as the relative humidity at which deliquescence is completed.

While in Arsene et al. (2011) high RH values were representative for the warm seasons it seems that in Galon-Negru et al. manuscript high RH values, and hence representative conditions for deliquescence to occur, were mainly prevailing during the cold seasons. As suggested in Arsene et al. (2011) under deliquescence conditions, deliquesced particles may lead to the formation of aqueous droplets by a process favouring formation of internal mixture from externally mixed particles, which actually results in changes in the activities of semi-volatile species. According to Pathak and Chan (2005) in mixed collected particles the gas-particle equilibrium will tend to re-establish by mass exchange between the gas and particulate phases. Moreover, if deliquesced particles are formed under favourable meteorological conditions, then the following reactions can occur during inter-particle or gas-particle interactions:

$$NH_4Cl_{(s)} \xleftrightarrow{\ RH>RHD\ } NH_4Cl_{(aq)} \leftrightarrow NH_4^+{}_{(aq)} + Cl^-{}_{(aq)} \leftrightarrow NH_{3(g)} + HCl_{(g)} \qquad (RS1)$$

$$NH_4NO_{3(s)} \xleftrightarrow{\ RH>RHD\ } NH_4NO_{3(aq)} \leftrightarrow NH_4^+{}_{(aq)} + NO_3^-{}_{(aq)} \leftrightarrow NH_{3(g)} + HNO_{3(g)} \qquad (RS2)$$

$$H^+{}_{(aq)} + Cl^-{}_{(aq)} \leftrightarrow HCl_{(g)} \qquad (RS3)$$

$$H^+{}_{(aq)} + NO_3^-{}_{(aq)} \leftrightarrow HNO_{3(g)} \qquad (RS4)$$

$$H^{+}_{(aq)} + NH_{3(g)} \leftrightarrow NH_4^{+}_{(aq)} \tag{RS5}$$

with (RS1), (RS2), (RS5) being responsible for artefacting ammonium measured concentrations. While, actually, reactions (RS1) and (RS2) would be responsible for negative ammonium artefacts, reaction (RS5) would involve absorption of ammonia on aqueous droplets with positive artefacts. The competition between (RS1), (RS2) and (RS5) reactions will strongly depend on meteorological conditions directly related to RH, RHD and temperature but also on chemical species abundances. According to Pathak and Chan (2005) in ammonium reach environments, reaction (RS5) would be responsible on highly contributing artefacts in $NH_4^{+}$ and $H^{+}$ distribution, especially due to acidity neutralisation by the existent ammonia. In their work, Arsene et al. (2011) proposed a detailed rationale for a potential estimation of the unmeasured $NH_4^{+}$ fraction induced most probable by the reactions inducing negative artefacts and significantly occurring during sampling. In Galon-Negru et al. work this unmeasured $NH_4^{+}$ has been called "missing $NH_4^{+}$" and for its estimation Arsene et al. (2011) rationale has been used.

**Table S 3**: Detailed statistics of linear regression analysis for $\sum_{cations}$ and $\sum_{anions}$ determined when estimated missing $NH_4^{+}$ has been taken into account, in the $PM_{2.5}$ and $PM_{10}$ fractions over the total investigated period, cold and warm seasons, in Iasi, north-eastern Romania.

| | $PM_{2.5}$ | | | | $PM_{10}$ | | | |
|---|---|---|---|---|---|---|---|---|
| | $\sum_{cations}/\sum_{anions}$ ratio | r | p-value | Confidence level (%) | $\sum_{cations}/\sum_{anions}$ ratio | r | p-value | Confidence level (%) |
| Total period | 0.95 | 0.98 | < 0.001 | 99.9 | 0.95 | 0.98 | < 0.001 | 99.9 |
| Cold seasons | 0.97 | 0.99 | < 0.001 | 99.9 | 0.98 | 0.99 | < 0.001 | 99.9 |
| Warm seasons | 0.87 | 0.96 | < 0.001 | 99.9 | 0.86 | 0.96 | < 0.001 | 99.9 |

Note: r–Pearson coefficient

**Section S 5**

**Additional information about ISORROPIA-II runs performed in the present work**

[revised manuscript text omitted]

**Section S 7**

In the Introduction section of Galon-Negru et al. work it has been already stated that "Particles acidity might influence transition metals solubility and enhance aerosols toxicity and atmospheric nutrient delivered through atmospheric deposition in marine areas (Meskhidze et al., 2003; Fang et al., 2017).". Particles pH is known to affect the solubility of trace metals

(such as Fe) found in aerosols (Meskhidze et al., 2003; Fang et al., 2017), with lower pH dissolving metal oxides and converting them to soluble metal sulfates (Oakes et al., 2012), conditions which would significantly change the aerosol environmental impacts. As presented in Guo et al., (2016), while at global level metal mobility will mainly affect nutrient distributions with important impacts on productivity, carbon sequestration, and oxygen levels in the ocean, at regional scales soluble transition metals seem mainly to enhance aerosol toxicity or their oxidative potential. Through $SO_4^{2-}$ contributions to aerosols acidity and the historical record of associations between so-called particle "strong acidity" and adverse health effects, Fang et al. (2017) suggest that $SO_4^{2-}$ linkages to health are most probably determined by $SO_4^{2-}$ role in acid dissolution of primary metals commonly found in ambient particles, while Tsagkogeorgas et al. (2017) suggest that on a regional to global scale the acidification of fresh water and forest ecosystems is most probable caused by wet and dry deposition of $SO_2$ and $SO_4^{2-}$ particles.

Epidemiological studies have often reported adverse health outcomes associated with strong aerosols acidity (Dockery et al., 1996; Gwynn et al., 2000; Lelieveld et al., 2015). In "Measuring aerosol damage to the atmosphere" (http://ec.europa.eu/research/infocentre/article_en.cfm?artid=31296, 28 October 2013), "Arsene says the high aerosol levels are probably linked to the high rate of various pulmonary diseases registered at the Clinic of Pulmonary Diseases in Iasi. These include cases like chronic obstructive pulmonary diseases (COPD), pneumonia, asthma (allergy, rhinitis), bronchiectasis (sometimes associated with bacterial infection) and tuberculosis often detected in the area.". The claim is not a result of a systematic work on an epidemiological study in Iasi, north-eastern Romania, but it represents a general observation of Arsene after performing research work with specialists and medical doctors in the field of pulmonary diseases (Cernat et al., 2011).

Moreover, upon our knowledge there is a single report, mentioned by Arsene et al. (2007), referring to indirect evidences of acid rain impacts in Iasi region, north-eastern Romania. In the performed study, Arsene et al. (2007) calculated high wet deposition fluxes for ions of anthropogenic origin in the $SO_4^{2-} > NO_3^- > NH_4^+$ order. The determined values, although among the highest, were in generally in good agreement with values reported for other north European sites (such as those for Poland, the Czech Republic, etc.). In the analysed raining events, air masses crossing very large continental areas (N and E sectors) brought the highest contribution in terms of fluxes for ionic species such as $Ca^{2+}$ while the NW and W sectors brought more significant contributions for anthropogenic related elements ($SO_4^{2-}$, $NO_3^-$, $NH_4^+$). Arsene et al. (2007) suggested that in the rainwater, higher fluxes of these elements for the NW and W sectors, when compared with other geographical sectors, was most probably also a result of important local influences, as the sampling site was located east side of the Oriental Carpathians chain that could constrain, to some extent, the pollution plume transported from western Europe. Arsene et al. (2007) report that the pH of the analysed rainwater events was 5.92 (volume weighted mean average, VWM) suggesting a sufficient load of alkaline components neutralizing rainwater acidity. Moreover, the authors suggested that on average, 97 % of the acidity in the collected rainwater samples was mostly neutralized by $CaCO_3$ and $NH_3$.

**Section S 8**

Regarding meteorological parameters, i.e. RH and temperature, in Galon-Negru et al. work is shown that over the cold season's high RH values and low temperatures were prevailing while over the warm seasons these parameters showed opposite trends. In the work of Ianniello et al. (2011), related to chemical characteristics of inorganic ammonium salts in $PM_{2.5}$ in the atmosphere of Beijing (China), the relationship between partitions of specific nitrogen-containing inorganic species towards various parameters has been investigated. The prevailing conditions over the two investigated time-periods, i.e. 23 January – 14 February 2007 and from 2 to 34 August 2007, showed higher RH values over the summer (high temperatures) and lower RH values over the winter (low temperatures).

Ianniello et al. (2011) treats in a very complex manner the relationship existent between particulate species such as $SO_4^{2-}$, $NO_3^-$, $Cl^-$ and $NH_4^+$, and possible partition routes in the $NH_3/NH_4^+$ system. Gas-to-particle conversion (photochemical processing) or heterogeneous processes are assigned as important sources contributing to particulate $NO_3^-$ abundances in the atmosphere. The authors claim that while $NO_2$ conversion to $NO_3^-$ through photochemical processing during the winter season is expected to contribute to $NO_3^-$ abundance over this period of the year, the heterogeneous formation (through condensation or absorption of $NO_2$ in moist aerosols) generally relates to relative humidity and the particulate atmospheric loading (Ianniello et al., 2011). The existence of large amounts of particulate $NO_3^-$, observed in summer by Ianniello et al., 2011, was considered as unexpected since $NH_4NO_3$ is semi-volatile and tends to dissociate and remain in the gas phase under high temperatures. Ianniello et al., (2010) report 6 times higher $NH_3$ concentrations in summer than in winter, with temperatures ranging from 1 to 14 $^o$C in winter and 22 to 35 $^o$C in summer periods. Under these circumstances high concentrations of fine particulate $NO_3^-$ in summer period were attributed to the existence of higher concentrations of $NH_3$ in the atmosphere, available to neutralise not only $H_2SO_4$ but also $HNO_3$ from the atmosphere. In addition, at the high RH values (daily mean in the 35 to 90 % range) reported in Ianniello et al. (2011) study significant fraction of $HNO_3$ and $NH_3$ were considered to be dissolved in humid particles, with enhanced distribution of fine particulate $NO_3^-$ and $NH_4^+$ in the atmosphere.

In Galon-Negru et al. work particulate $NH_4^+$ showed higher values during the cold seasons when compared to the warm seasons and its variation coincide with those of fine particulate $NO_3^-$ and $Cl^-$. Although such behaviour would indicate that $NH_4^+$ was largely originating from the neutralization between ammonia and acidic species such as $HNO_3$ and $HCl$, as shown under Subsection 3.2.3 (Stoichiometry of $(NH_4)_2SO_4$, $NH_4NO_3$ and $NH_4Cl$), the most important contribution was most probably brought by $H_2SO_4$ species. However, the very similar pattern observed for $NO_3^-$, $Cl^-$ and $NH_4^+$ suggest that these species were representative for likely internally mixed particles and came most probably from similar gas-to-particle processes (Huang et al., 2010). Galon et al. report for $SO_4^{2-}$ comparable concentrations during both the summer and winter seasons, and Backes et al. (2016a) suggest that the formation of $SO_4^{2-}$ particles is not limited by $NH_3$ in any season. During the warm seasons, higher temperatures and solar radiation intensity, enhancing the photochemical activity and the atmospheric oxidation potential, is also enhancing the oxidation rate of $SO_2$ to particulate $SO_4^{2-}$, but during the cold seasons wood burning might become an important source of $SO_4^{2-}$.

At high RH values, and especially for deliquescent particles, there is suggestion that most of the fine particulate $NO_3^-$ exists as an internal mixture with $SO_4^{2-}$, so that $HNO_3$ can be easily absorbed into the droplets, a process considerably reducing the thermodynamic dissociation constant for $NH_4NO_3$ (Zhang et al., 2000; Ianniello et al., 2011). Under these circumstances it is supposed that fine particulate $NO_3^-$ can be formed from $HNO_3$ and $NH_3$ through heterogeneous reactions on fully neutralised fine particulate $SO_4^{2-}$ (a process which has been taken into account also in Galon et al. work, under the Subsection 3.2.3). In Galon et al. work significant correlation has been observed between $SO_4^{2-}$ and $NO_3^-$. These two measured parameters showed significant correlation with the RH, with high concentrations in $SO_4^{2-}$ or $NO_3^-$ being formed at high RH values. Such behaviour would allow us suggesting that $NO_3^-$ is being produced on preexisting $SO_4^{2-}$ aerosols, which could provide sufficient area and aerosol water content for the heterogeneous reaction to occur should be available.

Markovic et al. (2011) show that at high RH, the amount of the gaseous precursors, such as $NH_3$ and $HNO_3$, have relatively little influence on the formation of fine particulate nitrate. Ianniello et al. (2011) concluded also that in summer period, with high prevailing RH values, almost on all days the meteorological conditions were favourable for the formation of $NH_4NO_3$ at Beijing site.

In Galon et al. work, regarding gaseous $NH_3$ values such as those derived from ISORROPIA, it should be pointed out that its estimated concentration was much higher than that of $HNO_3$ and $HCl$ in gas phase. During the cold seasons the $[NH_3]/([HNO_3] + [HCl])$ ratio is as high as $2.0 \pm 0.6$ for RH values < 40 %, $3.6 \pm 2.0$ for RH values in the 40–60 % range, and $3.0\pm1.4$ for RH values > 60 % RH. During the warm seasons, for similar RH groups, the ratio takes the $4.9 \pm 1.9$, $4.6 \pm 1.4$ and $9.1.0 \pm 6.2$ values. Ianniello et al. (2011) report a $[NH_3]/([HNO_3] + [HCl])$ ratio of $27.90 \pm 12.70$ for winter season and of $54.06 \pm 20.60$ for the summer period. The authors are finally claiming that the atmosphere of their interest location was ammonia-rich in gas phase over both investigated seasons.

Moreover, for the specific conditions referring to Galon et al. work, the authors have observed that by analysing the events for similar RH groups, but for cold and warm seasons separately, the $NH_3/NH_4^+$ partition didn't enhance significantly. During the cold seasons over the three RH investigated groups, the fractions of the $NH_3$, $HCl$ and $HNO_3$ present in the gaseous form have taken in average the $71.4 \pm 4.7$, $4.4 \pm 0.3$ and $24.0 \pm 4.8\%$ values. Over the warm seasons for the previously mention species the fractions present in the gaseous form have taken in average the $83.1 \pm 3.4$, $6.6 \pm 0.6$ and $10.3 \pm 3.4\%$ values.

**Section S 9**

Backes et al. (2016a,b) present in some of their publication seasonal distributions in $NH_3$ emissions under static- and dynamic-time profile (STP and DTP) scenarios. While the STP scenario lacks dynamic, meteorology dependent or specific differences in policies or intensity of animal husbandry, the DTP scenario takes into account potential meteorological variables (such as wind speed and surface temperature) influence on the distribution of the annual emissions. However, while under the STP scenario, the annual time series of $NH_3$ showed one annual peak in spring (March), in the DTP scenario $NH_3$ concentrations show two annual peaks, one in spring (May) and one in autumn (September). The authors underline that

implementation of the DTP resulted in a shift from seasonal average winter NH$_3$ emissions to summer emissions. According to Backes et al. (2016a), high NH$_3$ concentrations are known to appear in proximity to emission sources, due to its low atmospheric lifetime which results in NH$_3$ concentration levels in the atmosphere which closely follow the seasonal emission trend.

[Figure]

(a)                                                              (b)

**Figure S 1**: Time series in NH$_3$ concentrations derived from ISORROPIA-II runs for the 2016 data-base, both for PM$_{2.5}$ and PM$_{10}$ scenarios in Iasi, north-eastern Romania.

Moreover, Sutton et al. (2013) claim that together with increased anthropogenic activity, global NH$_3$ emissions may increase from 65 (48–85) Tg N in 2008 to about 132 (89–179) Tg N by 2100. Most NH$_3$ emissions are known to result from agricultural productions strongly influenced by climatic interactions. There is, however, recognition about the fact that most of the up to now used approaches failed in recognising that a warm dry-year would tend to give larger NH$_3$ emissions than a cold-wet year. Estimates in the global NH$_3$ emission made by Sutton et al., (2013) indicate agricultural soils and crops, including emissions from grazing and land application of animal manure, as a first ammonia ranked source with 28.3 Tg N yr$^{-1}$ contribution. This is immediately followed by excreta from domestic animals (8.7 Tg N yr$^{-1}$), oceans and volcanoes (8.6 Tg N yr$^{-1}$), biomass burning (5.5 Tg N yr$^{-1}$), waste composting and processing (4.4 Tg N yr$^{-1}$), and by other sources with contributions of < 3.3 Tg N yr$^{-1}$.

Details presented in FIG. S 1 clearly show that for Iasi, north-eastern Romania, yearly NH$_3$ distributions as derived from ISORROPIA-II model present clear maxima in February and December in a behaviour suggesting important local sources contributions over the winter season. It might be that this distribution follows actually a trend mainly controlled by the agricultural practices in rural areas surrounding Iasi, mainly related to the timing of manure spreading on agricultural soils. Moreover, in Iasi and in the nearby counties, the contribution brought by open (sheep and goat) barns, operating over the cold season mainly in a "hot-spot" mode, might increase as importance. Manure storage from local and regional closed

barns (pigs and poultry) might also bring important contributions over the winter. Although biomass burning might bring important contribution in terms of global ammonia emissions it is believed that at the interest location this source is overwhelmed by agricultural practices in the area.

**Table S 4**: Detailed statistics of linear regression analysis for major water soluble ions determined in the $PM_{2.5}$ and $PM_{10}$ fractions over the total investigated period in Iasi, north-eastern Romania.

| | PM fraction | Slope | Pearson coefficient (r) | p-value | Confidence level (%) |
|---|---|---|---|---|---|
| $Cl^-$ vs. RH | $PM_{2.5}$; $PM_{10}$ | + | > 0.71 | 0.010 | 95.4 |
| $Cl^-$ vs. T | $PM_{10}$ | - | > 0.59 | 0.045 | 95.4 |
| $Cl^-$ vs. PM | $PM_{2.5}$; $PM_{10}$ | + | > 0.70 | 0.038 | 95.4 |
| $Cl^-$ vs. MLD | $PM_{2.5}$; $PM_{10}$ | - | > 0.69 | 0.012 | 95.4 |
| $NO_3^-$ vs. RH | $PM_{2.5}$; $PM_{10}$ | + | > 0.84 | < 0.001 | 99.9 |
| $NO_3^-$ vs. T | $PM_{2.5}$; $PM_{10}$ | - | > 0.94 | < 0.001 | 99.9 |
| $NO_3^-$ vs. MLD | $PM_{2.5}$; $PM_{10}$ | - | > 0.84 | < 0.001 | 99.9 |
| $NO_3^-$ vs. PM | $PM_{2.5}$; $PM_{10}$ | + | > 0.70 | 0.021 | 95.4 |
| $SO_4^{2-}$ vs. RH | $PM_{2.5}$; $PM_{10}$ | + | > 0.44 | 0.177 | 68 |
| $SO_4^{2-}$ vs. T | $PM_{2.5}$; $PM_{10}$ | - | > 0.41 | 0.214 | 68 |
| $SO_4^{2-}$ vs. PM | $PM_{2.5}$; $PM_{10}$ | + | > 0.45 | 0.162 | 68 |
| $NH_4^+$(total) vs. RH | $PM_{2.5}$; $PM_{10}$ | + | > 0.80 | 0.001 | 95.4 |
| $NH_4^+$(total) vs. T | $PM_{2.5}$; $PM_{10}$ | - | > 0.94 | < 0.001 | 99.9 |
| $NH_4^+$(total) vs. MLD | $PM_{2.5}$; $PM_{10}$ | - | > 0.80 | 0.001 | 95.4 |
| $NH_4^+$(total) vs. PM | $PM_{2.5}$; $PM_{10}$ | + | > 0.70 | 0.023 | 95.4 |

Note: RH–relative humidity; T–temperature; MLD–mixing layer depth; PM–particle loading;

**Table S 5**: Detailed statistics of linear regression analysis for molar concentrations of particulate $NH_4^+$ and $SO_4^{2-}$ in $PM_{10}$ fraction over the cold and warm seasons in Iasi, north-eastern Romania.

| | $NH_4^+$ from raw IC data | | | $NH_4^+$(total) | | |
|---|---|---|---|---|---|---|
| | Pearson coefficient (r) | p-value | Confidence level (%) | Pearson coefficient (r) | p-value | Confidence level (%) |
| Cold seasons | 0.97 | < 0.001 | 99.9 | 0.96 | < 0.001 | 99.9 |
| Warm seasons | 0.98 | < 0.001 | 99.9 | 0.97 | < 0.001 | 99.9 |

**Section S 10**

The data set from the present work has been investigated through Radar and/or Pie charts in order to identify the potential contributions brought by various sectors long-range transport to the atmospheric PM burden in the area. It should be however emphasized that the PM chemical composition appeared to be mainly driven by the air mass origin and type (e.g., continental, regional), meteorologically-controlled air mass characteristics (buoyancies), geographical context (the Oriental Carpathians chain, with the highest altitude of 1907 m, is facing Iasi ~ 150 km north-westerly distance), etc. As presented in FIG. S 2a, in overall, in terms of sectors contributions the N-NE up to the S (clockwise round) area seems to bring the most important fraction to both the $PM_{2.5}$ and $PM_{10}$ abundances. Higher loaded events in terms of aerosols mass concentration

were those associated with long-range continental transport, Saharan dust-events, regional transport from more arid areas (S-SE) and buoyancy affected air masses.

Regarding potential local anthropogenic contribution to PM atmospheric burden it should be emphasized that in the immediately nearby area of the sampling point neither important commercial areas nor industrial units are operating and traffic should be the most important contributor. Actually in Iasi, in the post-communist period, the industry has almost completely ceased down and presently the Antibiotic company (7 km from the sampling point, western, W, direction), a small brick plant (8 km, SE) and the heat and power plant (5 km, SE) may account as for the most significant local anthropogenic contributors. However, Iasi is known as an important Romanian university centre (> 42000 students by 2015) with intense activity over university semesters. Traffic in the nearby area and commercial activity in the top hot-spots area of Iasi get more intense over these periods of the year. As for the rural areas surrounding Iasi, or from the nearby counties, farming activities are mainly related to cereal crops, plantation agriculture, open and closed animal barns (cattle, sheep and goat, pigs, poultry).

[Figure]

(S 2a)                              (S 2b)

**Figure S 2**: Averaged annual sectors contributions both in the $PM_{2.5}$ (S 2a) and $PM_{10}$ (S 2b) fractions.

| | Winter | Spring | Summer | Autumn |
|---|---|---|---|---|
| PM$_{2.5}$ | | | | |
| Fo | | | | |
| Cl$^-$ | | | | |
| HCO$_3^-$ | | | | |
| Ac | | | | |
| NO$_3^-$ | | | | |
| SO$_4^{2-}$ | | | | |
| Ox | | | | |
| NH$_4^+$ | | | | |
| Na$^+$ | | | | |

[Figure]

**Figure S 3**: Particulate inorganic and organic ions seasonal contributions (in the PM$_{2.5}$ fraction) associated with the different identified air mass.

[Figure]

PM$_{2.5}$

PM$_{2.5}$

PM$_{10}$

PM$_{10}$

PM$_{2.5}$

PM$_{2.5}$

PM$_{10}$

PM$_{10}$

[Figure]

[Figure]

[Figure]

[Figure]

[Figure]

[Figure]

[Figure]

[Figure]

[Figure]

[Figure]

[Figure]

**Figure S 4**: Trajectory air mass associated percentual contributions for the identified-quantified ions and ISORROPIA-II predicted gaseous NH$_3$, HNO$_3$ and HCl both in PM$_{2.5}$ (top) and PM$_{10}$ (bottom) fractions. Selected investigated events are presented. Note: Fo – formate, Ac – acetate, Ox – oxalate, NH$_4^+$ reflects NH$_4^+$$_{total}$.

It should be also underlined that for ionic species specific for the coarse fraction, i.e. Na$^+$ and Ca$^{2+}$, the maxima seems to be mainly induced either by the prevalent direction of the air masses (spring maxima of Na$^+$ ion most probably induced by the fact that March and April appeared as months with prevalent southern air masses crossing large continental areas with more salty soil) or by the drought (the summer maxima of Ca$^{2+}$ was mainly influenced by the ionic chemical composition of aerosol samples collected in August 2016, 7 events, a month of the 2016 year without any single raining events). Actually, in August 2016, 1 event was strongly affected by atmospheric driven buoyancy (18.2 % fine particulate Ca$^{2+}$ and 42.1 % fine particulate HCO$_3^-$), 1 event was affected by turbulent local air masses (10.7 % fine particulate Ca$^{2+}$ and 24.8 % fine particulate HCO$_3^-$) and other 3 events showed contributions brought by air masses crossing larger continental areas (9.8 ± 1.7 % fine particulate Ca$^{2+}$ and 25.3 ± 3.9 % fine particulate HCO$_3^-$). Other 2 events sampled at the beginning of August 2016 (of N-NW direction both) in terms of Ca$^{2+}$ and HCO$_3^-$ brought contributions as high as (4.3 ± 0.6 % fine particulate Ca$^{2+}$ and 9.7 ± 2.3 % fine particulate HCO$_3^-$). High percentual contributions within the identified and quantified species brought by Ca$^{2+}$ and HCO$_3^-$ ions were also observed in various events collected all year round which were affected by long-range transport air masses accompanied by atmospheric driven buoyancy or by long-range transport air masses coming from arid areas (e.g. 31 January 2016, 14 February 2016, 20 March 2016, 06 April 2016, 09 April 2016, etc). It has been observed that in average the contributions brought by such events were strongly affecting the distribution of the ionic chemical composition in sampled aerosols.